# LiME: Lightweight Mixture of Experts for Efficient Multimodal Multi-task Learning

**Md Kowsher** [1]  **Haris Mansoor** [2]  **Nusrat Jahan Prottasha** [1]  **Ozlem Garibay** [1]  **Victor Zhu** [3]  **Zhengping Ji** [3]
**Chen Chen** [1]

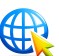 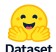 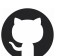

## Abstract

MoE-PEFT methods combine Mixture of Experts with parameter-efficient fine-tuning for multi-task adaptation, but require separate adapters per expert—causing trainable parameters to scale linearly with expert count and limiting applicability to adapter-based architectures. We propose **LiME** (**Li**ghtweight **M**ixture of **E**xperts), which achieves expert specialization through lightweight modulation rather than adapter replication. Instead of separate adapters, LiME uses a single shared PEFT module and modulates its output with lightweight expert vectors, reducing expert parameters while generalizing to any PEFT method. Notably, LiME introduces zero-parameter routing by leveraging existing frozen and adapted representations—eliminating learned router parameters typically required per layer. Theoretically, we prove that (i) more experts preserve more task-relevant information and (ii) modulation approximates full expert-specific PEFT with bounded error. LiME further incorporates n-gram windowed routing and adaptive expert selection (Auto Top-K) based on routing confidence. Experiments on MMT-47, a multimodal multi-task benchmark with 47 tasks spanning text, image, and video, demonstrate that LiME achieves competitive or superior performance while using up to $4\times$ fewer trainable parameters and up to 29% faster training compared to corresponding MoE-PEFT baselines.

[1]University of Central Florida [2]Coventry University [3]Axon. Correspondence to: Md Kowsher <ga.kowsher@gmail.com>.

*Proceedings of the 43rd International Conference on Machine Learning*, Seoul, South Korea. PMLR 306, 2026. Copyright 2026 by the author(s).

## 1. Introduction

Parameter-efficient fine-tuning (PEFT) has emerged as the dominant paradigm for adapting large pre-trained models to downstream tasks (Hu et al., 2022; Prottasha et al., 2024). By updating only a small fraction of parameters while keeping the backbone frozen, PEFT methods dramatically reduce computational and memory requirements compared to full fine-tuning. However, a fundamental limitation persists: current PEFT methods apply the same adaptation uniformly across all inputs, ignoring the inherent diversity in real-world data (Ning et al., 2025).

Mixture of Experts (MoE) offers a natural solution by routing different inputs to specialized sub-networks (Cai et al., 2025). This raises a natural question: is adding more experts actually beneficial?

---
**Theorem 1: Adding Experts Is Information-Preserving**

Let $X \in \mathbb{R}^d$ be an input and $Y$ the target. For an $n$-expert MoE with output $Z_n$ and an $(n-1)$-expert MoE with output $Z_{n-1}$,
$$I(Y; Z_n) \geq I(Y; Z_{n-1}).$$
(Full proof in Appendix A.)

---

Theorem 1 suggests that scaling the number of experts is beneficial in principle, more experts allow finer input partitioning without losing task-relevant information. However, realizing this benefit with existing MoE-PEFT methods is costly. Recent work has combined MoE with PEFT, using separate adapters per expert to achieve input-specific adaptation (Wu et al., 2024b; Dou et al., 2024). While effective, this approach introduces three inefficiencies: (i) **parameter explosion**—with $E$ experts, methods like MoLA (Gao et al., 2024) or LoRAMoE (Dou et al., 2024) replicate adapters, causing parameters to grow with $E$; (ii) **router overhead**—a learned router adds $d \times E$ parameters per layer, where $d$ is the hidden dimension; and (iii) **architecture dependence**—existing MoE-PEFT methods are largely restricted to LoRA-style adapters (Li et al., 2024b; Dou et al., 2024; Luo et al., 2024), excluding non-adapter PEFT methods such as Prompt Tuning (Lester et al., 2021), Slice-Fine (Kowsher et al., 2025a), and LayerNorm Tuning (Zhao et al., 2023).

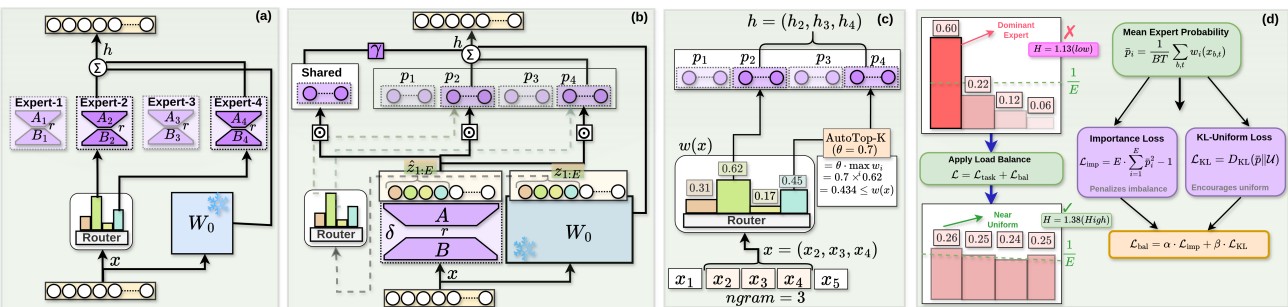

*Figure 1.* LiME is compatible with any PEFT method; we use LoRA only as an example. (a) MoE-LoRA replicates LoRA adapters $(A_i, B_i)$ for each expert and uses a learned router, requiring $E \times |\phi|$ adapter parameters plus $d_i \times E$ router parameters. (b) LiME shares a single PEFT module (LoRA here) and uses lightweight expert modulators $\mathbf{p}_i \in \mathbb{R}^{d_o}$, reducing trainable MoE parameters to $|\phi| + Ed_o$. *Router reuse:* routing is computed directly from representations already produced in the forward pass—an $E$-dimensional slice of the frozen output $z_{1:E}$ and the PEFT-modified output $\hat{z}_{1:E}$—so no separate router weights are introduced (dashed router). This PEFT block can be replaced by other PEFT strategies (e.g., DoRA, Prompt Tuning, SliceFine) without changing LiME. (c) N-gram routing shares one routing decision within each window (e.g., $n$=3), and Auto Top-K selects experts with $w_i \geq \theta \times \max_j w_j$. (d) Load balancing losses prevent expert collapse and encourage more uniform utilization (Details in §3).

These costs conflict with the core objective of PEFT. Existing MoE-PEFT scales by replicating adapters and adding routers, making trainable parameters grow quickly with $E$. This raises a question: *can we scale to a large number of experts while keeping tuning overhead minimal, enabling expert specialization for any PEFT method with zero routing parameters?* We show that the answer is yes, by re-examining and relaxing two common assumptions underlying current MoE-PEFT designs.

**Assumption 1:** *"Separate adapters are necessary for expert specialization."* We revisit this assumption. Our view is that expert specialization does not require replicating full PEFT modules; instead, experts can share a common adaptation and differ through lightweight *feature-wise* rescaling vectors (§3). This is motivated by the fact that large pretrained models already encode substantial knowledge, so effective adaptation for fine-tuning often amounts to small perturbations rather than architectural changes (Wu et al., 2024c). Empirically, element-wise rescaling of pretrained activations can match more complex adapters using only scaling vectors (Liu et al., 2022; Hyeon-Woo et al., 2021; Lian et al., 2022). Similarly, partial parameter sharing in LoRA (e.g., a shared $A$ with expert-specific $B$) remains effective (Tian et al., 2024). These results support our design choice: expert-specific behavior can emerge by rescaling the output of a shared PEFT update rather than replicating separate expert modules, and Theorem 2 formalizes this by showing that LiME's modulation can approximate expert-specific PEFT with bounded error.

**Assumption 2:** *"A learned router is necessary for routing."* Standard MoE learns a router to compute routing weights, increasing computational and parameter costs. However, recent work demonstrates that internal representations of pretrained models contain rich semantic information, en-

abling tasks like clustering without external models (Lee et al., 2025; Aharoni & Goldberg, 2020). Similarly, PEFT adaptation learns to encode task-relevant features—inputs requiring similar adaptations naturally cluster together (Gou et al., 2023). *Together, the frozen pretrained representations and the PEFT adaptation provide sufficient signal for routing, eliminating the need for additional routing parameters (§E).*

Building on these insights, we present **LiME** (illustrated in Figure 1) to address these limitations through two key components: **(i) Lightweight experts**, a scaling-based approach that mitigates *parameter explosion* and *architecture dependence*. Instead of replicating full adapters for each expert, LiME applies expert-specific scaling vectors to rescale a shared PEFT output element-wise. This design substantially reduces expert-specific parameters and is compatible with any PEFT method, rather than being restricted to LoRA-style adapters. **(ii) Zero-parameter routing**, which eliminates *router overhead*. Instead of a learned router, LiME uses the frozen layer output and the PEFT output to compute expert selection probabilities. We take an $E$-dimensional slice from these representations (with $E \ll d$) and map it to a distribution over the $E$ experts, without introducing any additional routing parameters.

Beyond its core design, LiME incorporates mechanisms that address common MoE training challenges: **Auto Top-K** adaptively selects experts based on routing confidence, activating fewer experts when confident and more when uncertain—this improves over fixed top-k selection by avoiding both wasted computation on irrelevant experts and loss of useful expert combinations. **N-gram windowed routing** groups adjacent tokens within a fixed window to share routing decisions, encouraging local coherence in expert assignments based on the observation that neighboring tokens

*Table 1.* Comparison of MoE-based PEFT methods for a layer $\mathbf{W}_0 \in \mathbb{R}^{d_o \times d_i}$. LiME uniquely combines: (1) **zero router parameters**, (2) **adaptive expert selection** (Auto Top-K), (3) **n-gram routing granularity**, (4) **any PEFT compatibility** (not restricted to LoRA), and (5) **shared PEFT module** (trainable). $d_i$: input dimension, $d_o$: output dimension, $E$: number of experts, $|\phi|$: PEFT adapter parameters (e.g., $r(d_i + d_o)$ for LoRA, $r$: rank).

| Method | Expert Selection | Routing Granularity | Router Params | Any PEFT | Shared Expert | Shared PEFT | Expert Type | Expert Params | Total Params | Load Balance |
|---|---|---|---|---|---|---|---|---|---|---|
| MoCLE (Gou et al., 2023) | Top-1 | Token | $d_i \times E$ | ✗ | ✓ | ✗ | LoRA | $Er(d_i + d_o)$ | $d_iE + Er(d_i + d_o)$ | ✗ |
| LoRAMoE (Dou et al., 2024) | Soft | Token | $d_i \times E$ | ✗ | ✗ | ✗ | LoRA | $Er(d_i + d_o)$ | $d_iE + Er(d_i + d_o)$ | ✓ |
| MoELoRA (Luo et al., 2024) | Top-$K$ | Token | $d_i \times E$ | ✗ | ✗ | ✗ | LoRA | $Er(d_i + d_o)$ | $d_iE + Er(d_i + d_o)$ | ✓ |
| MoRAL (Yang et al., 2024) | Top-$K$ | Token | $d_i \times E$ | ✗ | ✗ | ✗ | LoRA | $Er(d_i + d_o)$ | $d_iE + Er(d_i + d_o)$ | ✗ |
| MixLoRA (Li et al., 2024b) | Top-$K$ | Token | $d_i \times E$ | ✗ | ✗ | ✗ | LoRA | $Er(d_i + d_o)$ | $d_iE + Er(d_i + d_o)$ | ✗ |
| HydraLoRA (Tian et al., 2024) | Soft | Token | $d_i \times E$ | ✗ | ✗ | ✓ | LoRA | $r(d_i + Ed_o)$ | $d_iE + r(d_i + Ed_o)$ | ✗ |
| MOLA (Gao et al., 2025) | Top-$K$ | Token | $d_i \times E$ | ✗ | ✗ | ✗ | LoRA | $Er(d_i + d_o)$ | $d_iE + Er(d_i + d_o)$ | ✓ |
| LoRACoE (Li et al., 2025b) | Soft | Token | $d_i \times E$ | ✗ | ✓ | ✗ | LoRA | $Er(d_i + d_o)$ | $d_iE + Er(d_i + d_o)$ | ✗ |
| DynMoLE (Li et al., 2025a) | Hybrid | Token | $d_i \times E$ | ✗ | ✗ | ✗ | LoRA | $Er(d_i + d_o)$ | $d_iE + Er(d_i + d_o)$ | ✓ |
| LD-MoLE (Zhuang et al., 2025) | Sparse | Token | $d_i \times E$ | ✗ | ✗ | ✗ | LoRA | $Er(d_i + d_o)$ | $d_iE + Er(d_i + d_o)$ | ✓ |
| MaLoRA (Wang et al., 2025) | Top-K | Token | $d_i \times E$ | ✗ | ✓ | ✗ | LoRA | $Er(d_i + d_o)$ | $d_iE + Er(d_i + d_o)$ | ✗ |
| HMoRA (Liao et al., 2025) | Top-K | Token and Task | $d_i \times E$ | ✗ | ✗ | ✗ | LoRA | $Er(d_i + d_o)$ | $d_iE + Er(d_i + d_o)$ | ✗ |
| MoSLD (Zhao et al., 2025) | Top-K | Token | $d_i \times E$ | ✗ | ✗ | ✓ | LoRA | $r(d_i + Ed_o)$ | $d_iE + r(d_i + Ed_o)$ | ✓ |
| **LiME (Ours)** | **Auto-$K$** | **N-gram** | **0** | ✓ | ✓ | ✓ | **Vector** | $\mathbf{Ed_o}$ | $|\phi| + \mathbf{Ed_o}$ | ✓ |

often share semantic context (Zaheer et al., 2020; Beltagy et al., 2020). **Load balancing losses** encourage uniform expert utilization, preventing the expert collapse problem where routing concentrates on few experts while others remain underutilized (Shazeer et al., 2017; Fedus et al., 2022).

To evaluate multimodal multi-task performance, we compile **MMT-47**, a unified suite of 47 existing benchmarks spanning text understanding, commonsense reasoning, video understanding, and image understanding. MMT-47 combines these tasks into a single training mixture with task-specific evaluations (details in Appendix L).

**Contributions:** (i) **LiME**, a framework that achieves expert specialization via element-wise rescaling on top of any PEFT method, with zero routing parameters; (ii) **Practical mechanisms** including Auto Top-K for adaptive expert selection, n-gram routing for local semantic coherence, and load balancing to prevent expert collapse; (iii) **Theoretical foundations** that motivate LiME, including an information-preservation result for scaling experts and an approximation guarantee showing that modulation can match expert-specific PEFT; and (iv) **Extensive evaluation** across 47 tasks, showing competitive or superior performance with up to 4× fewer parameters and 29% faster training compared to corresponding MoE-PEFT baselines.

**Related Work.** We briefly discuss related work, comparing LiME with existing MoE-PEFT methods in Table 1, and provide a detailed literature review in Appendix I. Broadly, this work connects to PEFT methods (Hu et al., 2022; Liu et al., 2024) and MoE-based PEFT (Zhuang et al., 2025; Dou et al., 2024; Li et al., 2024b). Existing MoE-PEFT approaches replicate adapters per expert, scaling parameters, requiring learned routers, and restricting applicability to adapter-based methods. LiME instead modulates a shared PEFT output with lightweight expert vectors and zero-parameter routing, enabling compatibility with any PEFT method.

## 2. Preliminaries

We consider adapting large pre-trained models to multiple downstream tasks across different modalities. Let $f_{\Theta} : \mathcal{X} \to \mathcal{Y}$ denote a pre-trained model with frozen parameters $\Theta$. Given $K$ tasks spanning $M$ modalities, our goal is to efficiently adapt $f_{\Theta}$ while minimizing trainable parameters.

**Parameter-Efficient Fine-Tuning.** Rather than updating all parameters, PEFT methods introduce a small set of trainable parameters $\phi$ while keeping $\Theta$ frozen. The adapted output becomes $h = f_{\Theta}(x) + g_{\phi}(x)$, where $g_{\phi}(\cdot)$ is a lightweight module—encompassing methods like LoRA, adapters, and prompt tuning. For a linear layer $\mathbf{W}_0 \in \mathbb{R}^{d_o \times d_i}$, this simplifies to $h = \mathbf{W}_0 x + \delta(x)$, where $\delta(x) \in \mathbb{R}^{d_o}$ is the adaptation term. The key benefit is $|\phi| \ll |\Theta|$. However, standard PEFT applies the same adaptation to all inputs, ignoring the diversity of inputs in multi-task settings.

**Mixture of Experts.** MoE addresses this limitation by routing different inputs to specialized experts. Given $E$ experts $\{\mathcal{E}_1, \ldots, \mathcal{E}_E\}$ and routing weights $w(x) \in \Delta^{E-1}$, the output is $y = \sum_{i=1}^{E} w_i(x) \cdot \mathcal{E}_i(x)$. However, traditional MoE in PEFT requires separate parameters per expert, leading to linear parameter growth and increased data requirements—each expert sees fewer samples as $E$ increases, making training less efficient. This motivates our lightweight expert modulation approach.

**Notation.** We denote frozen parameters by $\mathbf{W}_0$ and trainable parameters by $\phi$. We use $\odot$ for element-wise multiplication, $\|\cdot\|$ for $\ell_2$ norm, $\|\cdot\|_\infty$ for the max norm, $[E]$ for the set $\{1, \ldots, E\}$, and $\Delta^{E-1} = \{w \in \mathbb{R}^E : w_i \geq 0, \sum_i w_i = 1\}$

for the probability simplex.

# 3. LiME

> **Core Idea:** *Instead of replicating PEFT modules for each expert, use a single shared PEFT module and modulate its output with lightweight expert-specific vectors.*

We present LiME in four parts: *(i)* expert modulation that achieves specialization through lightweight vectors, *(ii)* zero-parameter routing derived from existing computations, *(iii)* practical mechanisms including n-gram windowed routing and adaptive expert selection, and *(iv)* training considerations including load balancing. The theoretical results in this section formalize the intuitions behind each design choice and are supported empirically in §4. **Lightweight Experts.** Let $z = \mathbf{W}_0 x \in \mathbb{R}^{d_o}$ be the frozen output and $\hat{z} = \delta(x) \in \mathbb{R}^{d_o}$ be the PEFT output. LiME rescales the PEFT output with expert-specific scaling vectors. Let $\mathbf{p}_i \in \mathbb{R}^{d_o}$ for $i \in [E]$ be the $E$ expert scaling vectors, and let $w(x) \in \mathbb{R}^E$ be routing weights (defined below). The scaled output is:

$$h = z + \hat{z} \odot \mathcal{P}(x) \quad (1)$$

where $\mathcal{P}(x) = \sum_{i=1}^{E} w_i(x) \cdot \mathbf{p}_i$ combines expert scaling vectors weighted by routing scores. Optionally, we include a shared scaling vector $\mathbf{p}_s \in \mathbb{R}^{d_o}$ with a learnable scalar $\gamma$:

$$h = z + \hat{z} \odot \mathcal{P}(x) + \gamma \cdot (\hat{z} \odot \mathbf{p}_s) \quad (2)$$

A natural question arises: can this lightweight modulation match full expert-specific PEFT, where each expert has its own adapter? We provide a theoretical guarantee:

> **Theorem 2: LiME Approximates Expert-Specific PEFT**
>
> Let $Z_{\text{MoE}}$ be the output of expert-specific PEFT (separate $\phi_e$ per expert) and $Z_{\text{LiME}}$ be LiME output (shared $\phi$ with modulators $\mathbf{p}_e$). If the approximation error is bounded by $\bar{\varepsilon}$, then:
> $$\left| \mathcal{R}^*(Z_{\text{LiME}}) - \mathcal{R}^*(Z_{\text{MoE}}) \right| \leq \mathcal{O}(\bar{\varepsilon}),$$
> where $\mathcal{R}^*$ is the optimal risk (proof in Appendix B).

This means LiME can closely approximate full expert-specific PEFT while using fewer parameters: $|\phi| + E d_o$ instead of $E \times |\phi|$. Since pretrained representations are highly redundant, full expert-specific adapters are often over-parameterized (Tian et al., 2025): much of the adaptation can be expressed by selectively scaling a small subset of useful feature dimensions. Supporting this view, Luo et al. (2023) show that using only $\sim 1\%$ of the most important feature dimensions can recover the performance achieved by the full representation. Theorem 1 motivates this efficient design: by keeping parameter costs low, LiME can scale to more experts without significant parameter growth.

**Zero-Parameter Routing.** Standard MoE layers learn a router $\mathbf{W}_r \in \mathbb{R}^{d \times E}$ to produce routing weights, adding $d \times E$ parameters per layer. LiME removes this cost by computing routing weights directly from representations that are already available in the forward pass.

> **Key Insight:** *We reuse existing representations as routing features. The frozen output $z$ provides general semantic information, while the PEFT output $\hat{z}$ captures task- and input-dependent corrections. Combining them yields effective routing with **zero** learned router parameters.*

This design is motivated by two observations. *(i) Forward-pass representations are informative.* The frozen layer output $z$ captures general semantic information, while the PEFT output $\hat{z}$ captures task-specific changes induced by fine-tuning. Inputs that require similar expert processing tend to produce similar patterns in these representations (§F.11), making them useful signals for expert selection. *(ii) A low-dimensional slice is sufficient.* Since $E \ll d$, routing only needs a small number of degrees of freedom to choose among $E$ experts, and a small slice of these already-computed representations provides enough signal in practice (Appendix E, §4.1).

Formally, we compute routing scores from $E$ dimensions of both representations; the specific dimensions are not critical, and we use the first $E$ for simplicity (Figure 4a):

$$w(x) = \text{softmax}\left( \frac{(1 - \gamma_r) \cdot \tilde{z}_{1:E} + \gamma_r \cdot \tilde{\hat{z}}_{1:E}}{\tau} \right). \quad (3)$$

Here $\tilde{z}_{1:E} = z_{1:E}/\|z_{1:E}\|_\infty$ and $\tilde{\hat{z}}_{1:E} = \hat{z}_{1:E}/\|\hat{z}_{1:E}\|_\infty$ are normalized slices, $\gamma_r \in [0, 1]$ balances frozen and PEFT-modified signals (§4.1), and $\tau > 0$ is the temperature (§F.6). The normalization stabilizes routing across layers and inputs.

**N-gram Windowed Routing.** LiME supports different routing granularities. In token-level routing, each token chooses experts independently, resulting in $T$ routing decisions for a length-$T$ sequence. In sequence-level routing, a single pooled representation is used to make one routing decision for the entire sequence, but this can miss within-sequence variation. We adopt n-gram windowed routing as a middle ground: we partition the sequence into windows of size $n$, and all tokens in a window share one routing decision (i.e., $T/n$ decisions per sequence). This reduces sensitivity to token-level noise and encourages locally consistent expert assignments, since neighboring tokens often form coherent semantic units (§F.2).

For causal models, we use the last token of each window as the routing representative. This choice is theoretically motivated:

> **Theorem 3: Informativeness in Causal N-gram Windows**
>
> Let $h_1, \ldots, h_n \in \mathbb{R}^d$ be hidden states within an n-gram window under causal attention, and let $Y$ denote the task objective. Then the routing informativeness is maximized at the window's final position:
> $$I(Y; h_n) \geq I(Y; h_{n-1}) \geq \cdots \geq I(Y; h_1)$$
> (Full proof in Appendix C).

Since later positions aggregate more context via causal attention, the last token of each n-gram window captures the

most task-relevant information, making it the optimal choice for routing decisions (Kowsher et al., 2025c).

**Auto Top-K.** After computing routing weights $w(x)$, we select which experts to activate. The standard approach—fixed top-$k$—always selects exactly $k$ experts regardless of the score distribution (§F.3). This can be inefficient: when one expert clearly dominates (e.g., 0.52 vs 0.12), selecting additional experts wastes computation; when multiple experts have similar scores (e.g., 0.35 vs 0.34 vs 0.33), restricting to $k = 2$ can discard useful expert combinations. We therefore use an adaptive policy that activates fewer experts when routing is confident and more when it is uncertain. This is consistent with Theorem 1, which suggests that retaining more experts is beneficial when multiple experts carry comparable task-relevant information. Auto Top-K uses a relative threshold:

$$\mathcal{S}_\theta(x) = \left\{ i : w_i(x) \geq \theta \cdot \max_j w_j(x) \right\}, \qquad (4)$$

where $\theta \in (0, 1]$ controls selection aggressiveness. For example, with $\theta = 0.5$, we keep all experts scoring at least half as high as the best. Selected experts are renormalized as $\tilde{w}_i(x) = w_i(x) / \sum_{j \in \mathcal{S}_\theta(x)} w_j(x)$. See Appendix D for comparisons with other strategies.

**Layer Formulation.** LiME applies to any layer equipped with PEFT (§H). For a frozen linear layer $\mathbf{W}_0$, the forward pass is: *(1)* compute the base output $z = \mathbf{W}_0 x$; *(2)* compute the PEFT update $\hat{z} \in \mathbb{R}^{d_o}$; *(3)* compute routing weights $w(x)$ from $z$ and $\hat{z}$, then select experts $\mathcal{S}_\theta(x)$ and renormalize weights to $\tilde{w}(x)$; *(4)* form the expert modulator $\mathcal{P}(x) = \sum_{i \in \mathcal{S}_\theta(x)} \tilde{w}_i(x) \cdot \mathbf{p}_i$; and *(5)* produce the final output: $h = z + \hat{z} \odot \mathcal{P}(x) + \gamma \cdot (\hat{z} \odot \mathbf{p}_s)$. Routing granularity (token or window) is chosen based on the task (§F.2).

**Initialization.** Expert modulators are initialized near unity: $\mathbf{p}_i \sim \mathcal{U}(1 - \sigma, 1 + \sigma)$ with $\sigma = 0.1$, so LiME starts close to standard PEFT and specialization emerges during training, following Kowsher et al. (2025b) (§F.8).

**Parameter Count.** For a model with $L$ LiME layers and $E$ experts, let $|\phi|$ denote the PEFT parameters per layer. The total number of trainable parameters is:

$$|\phi_{\text{LiME}}| = L \cdot \Big( \underbrace{|\phi|}_{\text{shared PEFT}} + \underbrace{E \cdot d_o}_{\text{expert modulators}} + \underbrace{d_o + 1}_{\text{shared modulator} +\gamma} \Big).$$

In contrast, traditional MoE-PEFT requires $L \cdot E \cdot |\phi|$ parameters, which scales linearly with the number of experts. LiME adds only $E \cdot d_o$ parameters per layer for expert modulators, independent of the PEFT method.

**Load Balancing.** MoE models can suffer from *expert collapse*, where routing concentrates on a few experts while others are underutilized. We add auxiliary losses to encourage balanced utilization. Let $\bar{p}_i = \frac{1}{BT} \sum_{b,t} w_i(x_{b,t})$ be the mean routing probability of expert $i$ over a batch of $B$ sequences with $T$ tokens (so $\sum_i \bar{p}_i = 1$). We use two comple-

mentary losses: *(i) Importance Loss* $\mathcal{L}_{\text{imp}} = E \sum_{i=1}^E \bar{p}_i^2 - 1$, and *(ii) KL-Uniform Loss* $\mathcal{L}_{\text{KL}} = D_{\text{KL}}(\bar{p} \| \mathcal{U})$, where $\mathcal{U}$ is uniform. The total objective is:

$$\mathcal{L} = \mathcal{L}_{\text{task}} + \alpha \mathcal{L}_{\text{imp}} + \beta \mathcal{L}_{\text{KL}}, \qquad (5)$$

where $\alpha$ and $\beta$ control the balancing strength (§4.1, §F.7).

# 4. Experiments

We evaluate LiME on multimodal multi-task dataset to test whether it works well across different modalities and tasks. LiME can be combined with any trainable PEFT method. To validate this, We instantiate LiME with adapter-based PEFT (LoRA (Hu et al., 2022), DoRA (Liu et al., 2024), LoRA-FA (Zhang et al., 2023)) and non-adapter PEFT (Slice-Fine (Kowsher et al., 2025a), Prompt Tuning (Lester et al., 2021)), denoted as LiMELoRA, LiMEDoRA, LiMELoRA-FA, LiMESliceFine, and LiMEPromptTuning.

We compare against (1) standard PEFT baselines (LoRA, DoRA, LoRA-FA, SliceFine, Prompt Tuning) and (2) MoE-PEFT baselines (MoCLE (Lee et al., 2025), MoELoRA (Luo et al., 2024), MixLoRA (Li et al., 2024b), HydraLoRA (Tian et al., 2024), MoLA (Gao et al., 2025), MoRe (Zhang et al., 2025)), and we also implement MoE-DoRA and MoELoRA-FA following standard MoE-PEFT procedures. We use LLaVA-OneVision-Qwen2-7B (Li et al., 2024a) as the base model (and LLaVA-OneVision-Qwen2-0.5B for ablations). Experiments are conducted on MMT-47, which we compile from existing benchmarks into a unified training mixture (158K samples) with 47 test sets spanning text, commonsense, video, and image tasks (Appendix L). We additionally evaluate on Molmo2-8B (Clark et al., 2026) to validate generalization across vision-language architectures (Appendix G.8).

We train for 3 epochs with 5 seeds, using differential learning rates ($2 \times 10^{-4}$ for PEFT parameters and $1 \times 10^{-3}$ for expert modulators), a cosine schedule, and 3% warmup. Unless stated otherwise, LiME uses 4 experts with Auto Top-K $\theta = 0.7$ and n-gram window size 3. Full hyperparameters are in Appendix J.

**Main Results.** Table 2 presents performance across seven benchmark categories spanning vision, language, and reasoning tasks. LiME variants consistently achieve competitive performance compared to both standard PEFT and MoE-PEFT methods. On Vision Benchmark, LiME-DoRA achieves the best result (78.12%), outperforming HydraLoRA (78.11%). On Commonsense Reasoning, LiMELoRA achieves 84.98%, the highest among all methods. On Object Motion & Spatial reasoning, LiME-DoRA (65.41%) outperforms the strongest baseline MoE-DoRA (65.16%). On GLUE, LiMELoRAFA (91.14%) and LiMESliceFine (91.19%) closely match the best baseline MoELoRA (91.21%). Notably, all LiME variants consis-

*Table 2.* Average results across benchmark categories. Highlighted rows show our LiME variants. **Bold**: best; underline: second best. #TTP: total trainable parameters. Detailed per-task results in Tables 6, 7, 8, 9, 10, 11, 12.

| Method | #TTP | Vision Benchmark | Image Classification | Commonsense Reasoning | GLUE | High Level Reasoning | Object Motion & Spatial | Action Understanding |
|---|---|---|---|---|---|---|---|---|
| PromptTuning | 0.07M | $72.63_{\pm2.24}$ | $50.22_{\pm0.96}$ | $77.50_{\pm1.53}$ | $71.91_{\pm1.35}$ | $15.86_{\pm2.29}$ | $14.51_{\pm2.64}$ | $15.86_{\pm2.42}$ |
| LoRA | 1.74M | $77.02_{\pm1.59}$ | $93.92_{\pm0.56}$ | $83.80_{\pm1.27}$ | $90.64_{\pm0.97}$ | $43.23_{\pm1.88}$ | $62.85_{\pm2.19}$ | $50.99_{\pm1.96}$ |
| DoRA | 2.09M | $76.35_{\pm1.62}$ | $93.35_{\pm0.68}$ | $83.53_{\pm1.01}$ | $90.37_{\pm0.99}$ | $42.56_{\pm1.88}$ | $62.14_{\pm2.13}$ | $50.41_{\pm2.07}$ |
| LoRA-FA | 0.70M | $76.31_{\pm1.62}$ | $93.50_{\pm0.55}$ | $83.45_{\pm1.11}$ | $90.31_{\pm0.81}$ | $42.47_{\pm1.94}$ | $61.97_{\pm2.17}$ | $50.30_{\pm2.07}$ |
| SliceFine | 1.74M | $77.06_{\pm1.53}$ | $93.98_{\pm0.53}$ | $83.84_{\pm1.22}$ | $90.54_{\pm0.97}$ | $43.30_{\pm1.86}$ | $62.91_{\pm2.12}$ | $51.06_{\pm1.94}$ |
| MoCLE | 5.92M | $77.59_{\pm1.39}$ | $94.25_{\pm0.42}$ | $84.18_{\pm1.03}$ | $90.66_{\pm0.85}$ | $43.78_{\pm1.87}$ | $63.43_{\pm2.10}$ | $51.53_{\pm1.94}$ |
| MoELoRA | 10.79M | $77.27_{\pm1.41}$ | $93.97_{\pm0.57}$ | $84.08_{\pm1.06}$ | $\mathbf{91.21}_{\pm0.87}$ | $43.84_{\pm1.89}$ | $63.07_{\pm1.95}$ | $51.13_{\pm2.00}$ |
| MixLoRA | 3.83M | $77.08_{\pm1.30}$ | $93.25_{\pm0.60}$ | $83.27_{\pm0.99}$ | $90.50_{\pm0.86}$ | $43.86_{\pm1.91}$ | $63.54_{\pm2.04}$ | $51.70_{\pm1.88}$ |
| HydraLoRA | 5.92M | $\underline{78.11}_{\pm1.45}$ | $94.58_{\pm0.52}$ | $84.43_{\pm1.03}$ | $91.05_{\pm0.88}$ | $\underline{45.79}_{\pm1.92}$ | $64.95_{\pm2.07}$ | $53.28_{\pm2.01}$ |
| MoLA | 9.04M | $77.10_{\pm1.48}$ | $\mathbf{94.74}_{\pm0.54}$ | $84.07_{\pm0.96}$ | $90.88_{\pm0.94}$ | $42.97_{\pm1.90}$ | $62.74_{\pm2.08}$ | $50.97_{\pm1.99}$ |
| MoRe | 3.59M | $77.98_{\pm1.30}$ | $93.84_{\pm0.57}$ | $84.44_{\pm1.09}$ | $90.74_{\pm0.97}$ | $45.58_{\pm1.95}$ | $64.95_{\pm2.06}$ | $\mathbf{53.48}_{\pm1.98}$ |
| MoEDoRA | 12.19M | $78.07_{\pm1.43}$ | $\underline{94.72}_{\pm0.71}$ | $84.11_{\pm0.98}$ | $90.98_{\pm0.97}$ | $\mathbf{45.92}_{\pm1.90}$ | $\underline{65.16}_{\pm2.06}$ | $52.54_{\pm2.01}$ |
| LiMEPromptTuning | 0.09M | $73.48_{\pm1.91}$ | $51.07_{\pm0.75}$ | $78.38_{\pm1.28}$ | $72.68_{\pm1.12}$ | $16.73_{\pm2.04}$ | $15.39_{\pm2.33}$ | $16.95_{\pm2.17}$ |
| LiMELoRA | 3.49M | $78.01_{\pm1.31}$ | $94.47_{\pm0.53}$ | $\mathbf{84.98}_{\pm0.86}$ | $91.02_{\pm0.84}$ | $45.00_{\pm1.86}$ | $65.05_{\pm2.08}$ | $53.19_{\pm1.93}$ |
| LiMEDoRA | 3.84M | $\mathbf{78.12}_{\pm1.48}$ | $94.50_{\pm0.52}$ | $\underline{84.76}_{\pm0.88}$ | $91.11_{\pm0.92}$ | $45.65_{\pm1.88}$ | $\mathbf{65.41}_{\pm2.14}$ | $\underline{53.39}_{\pm1.97}$ |
| LiMELoRAFA | 2.44M | $77.77_{\pm1.47}$ | $94.61_{\pm0.67}$ | $84.49_{\pm0.84}$ | $\underline{91.14}_{\pm0.87}$ | $44.91_{\pm1.99}$ | $64.89_{\pm2.12}$ | $53.01_{\pm1.94}$ |
| LiMESliceFine | 2.44M | $77.77_{\pm1.41}$ | $94.34_{\pm0.52}$ | $84.50_{\pm0.96}$ | $91.19_{\pm0.82}$ | $45.34_{\pm1.84}$ | $65.14_{\pm2.03}$ | $52.73_{\pm1.90}$ |

tently outperform their corresponding base PEFT methods across all categories, demonstrating that lightweight expert modulation effectively captures input-specific specialization. Detailed per-task results are provided in Appendix G.

**Efficiency Analysis.** We compare LiME against MoE-PEFT baselines on a single H100 GPU with batch size 2, gradient accumulation 2, and bfloat16 precision for 1 epoch on GLUE. LiME variants achieve higher through-put (3.39–4.52 samples/s) with shorter training time (25–36 min). Comparing corresponding methods: LiMEDoRA (35.5 min) is 29% faster than MoEDoRA (50.2 min), and LiMELoRA (31.5 min) is 23% faster than MoELoRA (41.0 min). The parameter comparison reveals the key advantage: LiMELoRA requires 0.52M trainable parameters—$4\times$ fewer than MoELoRA (1.97M)—and LiMEDoRA requires 0.57M— $4\times$ fewer than MoEDoRA (2.16M)—while main-taining comparable total size ($\sim$894M). This stems from our modulation design: instead of replicating adapters per expert ($E \times |\phi|$), LiME uses lightweight expert vectors ($E \times d$). Notably, LiMEPromptTuning achieves the best efficiency with only 0.02M parameters and 4.52 samples/s throughput, demonstrating LiME's versatility across PEFT methods.

> **Key Finding:** *LiME achieves competitive performance with up to $4\times$ fewer trainable parameters and up to 29% faster training compared to corresponding MoE-PEFT baselines.*

**Empirical Evidence of Theorem 1.** We empirically study Theorem 1, which states that adding experts makes the MoE output *at least as informative* about the target. Figure 3(c–d) reports GLUE accuracy as we vary the number of experts from 1 to 10. Both LiME and MoELoRA peak at 3–5 experts (stars), suggesting that a moderate number of experts often provides the best balance between added capacity and how well experts can be trained.

This behavior is expected in practice because Theorem 1 describes capacity under an ideal solution, while accuracy depends on finite-data training. If a batch contains $N$ for $E$ experts, each expert receives fewer tokens on average (about $O(N/E)$). With fewer examples per expert, rout-ing and load balancing become harder and experts can be under-trained, which may reduce generalization and hurt performance (Lepikhin et al., 2020; Geiger et al., 2020). MoE-PEFT is more sensitive to this because it learns a sep-arate adapter for each expert, so each adapter is trained on a smaller slice of data. LiME is more stable because all experts share the same PEFT module and only learn lightweight scaling vectors; the shared adaptation is trained on the full dataset. As a result, LiMELoRA scales to larger $E$ better than MoELoRA in the limited-data setting (§4.1).

**Empirical Evidence of Theorem 2.** Theorem 2 predicts that LiME can match expert-specific PEFT up to a small performance gap. We test this by comparing the internal representations of LiME and MoELoRA using Centered Kernel Alignment (CKA), a standard measure of represen-tation similarity that is invariant to rotation and scaling. Across GLUE tasks, LiME and MoELoRA show consis-tently high similarity: *SST-2 (0.942), QNLI (0.939), QQP (0.931), CoLA (0.939), MRPC (0.931), and STS-B (0.929),* with a mean CKA of 0.935 (Table 3). This suggests that LiME learns representations very close to those of full MoE-PEFT, while using $4\times$ fewer parameters. Together with the accuracy results in Table 2, these findings support the theorem's implication that expert scaling ($\hat{z} \odot \mathbf{p}_e$) can serve as an effective substitute for separate expert adapters. See Appendix B.1 for layer-wise analysis.

> **Finding:** *LiME reaches a mean CKA of 0.935 with MoELoRA, indicating a similar representations using $4\times$ fewer parameters.*

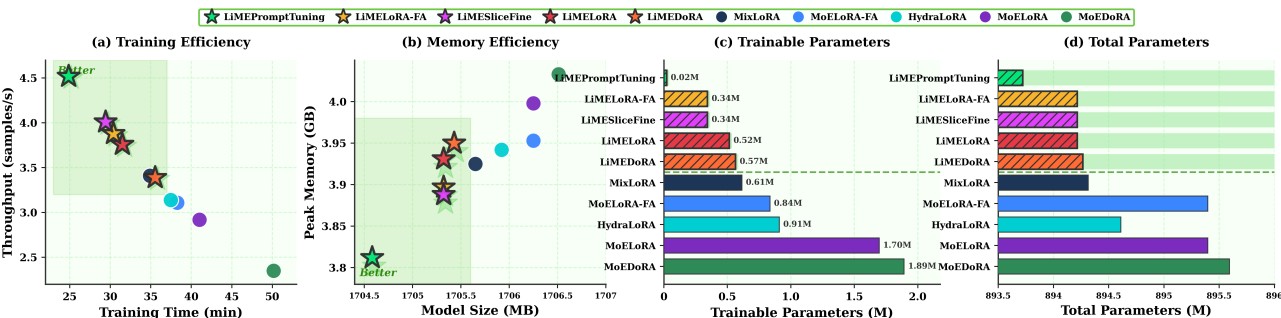

Figure 2. **Efficiency comparison of LiME vs. MoE-PEFT baselines.** (a) LiME variants (stars) achieve higher throughput and shorter training time; LiMEPromptTuning is the most efficient (4.52 samples/s, 25 min). (b) All methods show comparable peak memory due to the dominant frozen backbone. (c) LiME requires 0.02–0.57M trainable parameters—up to $4\times$ fewer than corresponding MoE-PEFT methods. (d) Total model size remains comparable ($\sim$894M) across all methods.

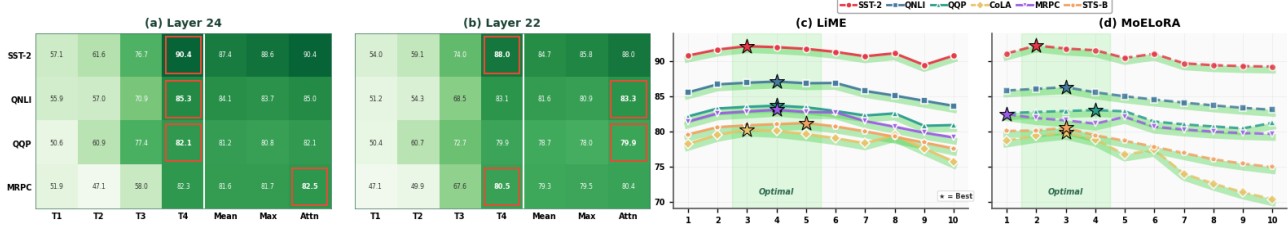

Figure 3. **Empirical validation of our theory.** (a–b) Linear probe accuracy at different token positions within an n-gram window (layers 24 and 22), supporting Theorem 3. (c–d) GLUE accuracy versus number of experts for LiME and MoELoRA, supporting Theorem 1; stars mark the best $E$ for each method.

**Empirical Evidence of Theorem 3.** We test the theorem's prediction that later tokens in an n-gram window carry more task-relevant information in causal models. We take hidden states from different positions in a window (T1, T2, T3, T4 from early to late) and train a linear probe to predict the label. Figure 3(a-b) shows results (layers 24 and 22) on GLUE. Probe accuracy consistently increases with token position, rising from $\sim$51–57% at T1 to 80–90% at T4 across tasks. This supports the intuition that later tokens have more context under causal attention, and thus encode more useful information for prediction. We also compare simple pooling choices: using the last token (T4) matches or outperforms mean/max pooling and is competitive with attention pooling, while avoiding the extra learnable parameters required by attention pooling. Overall, these results motivate our use of last-token routing for n-gram windowed routing in causal models.

> *Finding: Probe accuracy increases from $\sim$51–57% (T1) to 80–90% (T4), indicating later tokens encode more task-relevant information. Last-token routing is competitive with attention pooling while requiring no additional parameters.*

### 4.1. Ablation

**Routing Feature Selection (Figure 4a).** We examine which dimensions to use for routing: leading $E$ (first dimensions), central $E$ (middle), trailing $E$ (last), or random $E$ dimensions. Results show consistent performance across all strategies, demonstrating that routing features can be drawn from any subset of the feature space without significant degradation. This supports our low-rank redundancy hypothesis—discriminative routing signal is distributed across dimensions rather than concentrated in specific positions.

**Zero-Parameter vs Learned Routing (Figure 4b).** We compare our zero-parameter routing with standard separate (learned) routing that introduces $d \times E$ parameters per layer. Both approaches achieve comparable performance across text and vision tasks, confirming that existing representations provide sufficient routing signal without dedicated router parameters.

**Routing Balance $\gamma_r$ (Figure 4c-d).** We analyze the effect of $\gamma_r$, which balances frozen representations ($\gamma_r = 0$) and PEFT adaptations ($\gamma_r = 1$) in routing. On text tasks (c), performance improves from $\gamma_r = 0$ to $\gamma_r = 0.7$, then slightly decreases. Vision tasks (d) show similar trends. The optimal range $\gamma_r \in [0.6, 0.8]$ indicates that combining both signals outperforms using either alone, validating our dual-signal routing design.

> *Key Validation: Zero-parameter routing matches learned routing performance, is robust to feature position, and benefits from combining frozen and adapted signals ($\gamma_r \approx 0.7$)—confirming that dedicated router parameters are unnecessary.*

**Relative Threshold $\theta$ (Figure 5a).** We analyze the effect

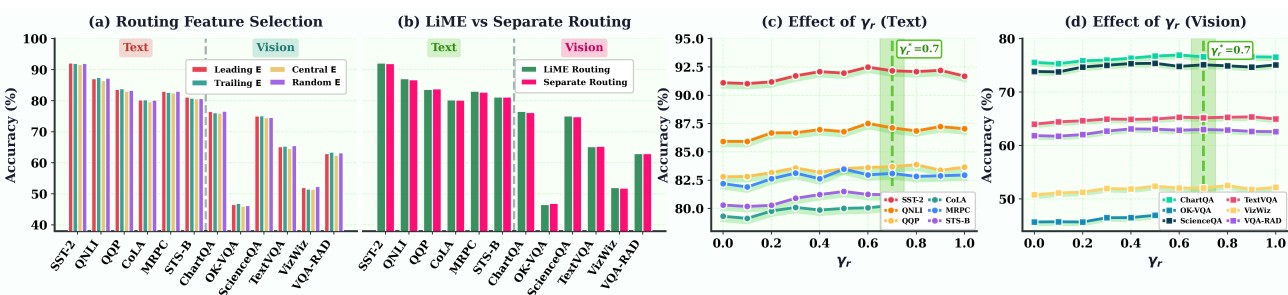

**Figure 4.** Routing ablations. (a) Feature selection for routing is robust. (b) Zero-parameter routing matches learned routing performance. (c-d) Routing balance $\gamma_r \in [0.6, 0.8]$ yields optimal performance by combining frozen and adapted signals.

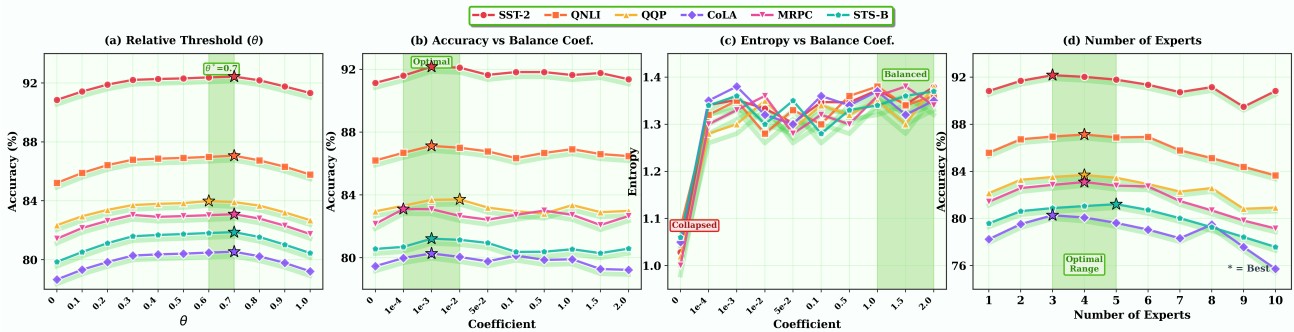

**Figure 5.** (a) Auto Top-K outperforms fixed Top-K. (b-c) Moderate load balancing prevents collapse while preserving specialization; over-balancing hurts accuracy. (d) Optimal expert count is $E \in [4, 6]$; beyond this, insufficient data limits further gains.

of selection threshold $\theta$ in Auto Top-K. Recall that experts are selected if $w_i(x) \geq \theta \cdot \max_j w_j(x)$. At $\theta = 0$, all experts are activated regardless of routing confidence, diluting specialization with irrelevant experts. As $\theta$ increases, selection becomes more stringent, improving performance by filtering low-confidence experts. On average, performance peaks around $\theta \approx 0.7$, though optimal values vary slightly across tasks. Beyond this, overly aggressive selection ($\theta \rightarrow 1.0$) approaches fixed Top-1 behavior, potentially discarding useful secondary experts.

> **Finding:** *Optimal $\theta = 0.7$ balances two failure modes: too lenient ($\theta \rightarrow 0$) activates noisy experts, while too strict ($\theta \rightarrow 1$) loses beneficial expert combinations.*

**Load Balancing Coefficient (Figure 5b-c).** We analyze the effect of balancing coefficient (applied to both $\mathcal{L}_{\text{imp}}$ and $\mathcal{L}_{\text{KL}}$). Figure 5b shows accuracy versus coefficient: without balancing (coefficient$= 0$), performance degrades due to expert collapse. Moderate coefficients ($\sim$1e-2 to 0.1) achieve optimal accuracy, while excessive balancing ($>1.0$) hurts performance. Figure 5c reveals why: at coefficient$= 0$, entropy is low ($\sim$1.0), indicating routing collapses to few experts. Increasing the coefficient raises entropy toward balanced utilization. However, forcing perfect uniformity prevents natural specialization—some inputs genuinely benefit from specific experts, and over-regularization suppresses this task-appropriate routing (§F.7, §F.11).

**Number of Experts (Figure 5d).** We vary the number of experts $E \in \{1, 2, \ldots, 10\}$. Performance improves from $E = 1$ to $E = 4\text{-}5$, consistent with Theorem 1 that more experts preserve more task-relevant information. However, beyond $E = 6$, accuracy plateaus or slightly decreases. This occurs because each expert receives fewer training samples as $E$ increases, making specialization harder without proportionally more diverse training data. The modulator vectors also require sufficient gradient signal to differentiate, which diminishes with many underutilized experts.

> **Observation:** $E \in [4, 6]$ *provides the best accuracy-efficiency trade-off. More experts require more diverse data to achieve meaningful specialization.*

**Expert Scaling: LiME vs MoELoRA (Figure 6)a.** We compare how LiME and MoELoRA scale with increasing number of experts on GLUE tasks. The shaded region highlights where LiME outperforms MoELoRA. Two patterns emerge clearly:

*LiME maintains stability.* Across all six tasks, LiME (green) shows relatively stable performance as $E$ increases from 1 to 10. Performance typically peaks around $E = 3\text{-}5$ and remains within a narrow range thereafter.

*MoELoRA degrades significantly.* In contrast, MoELoRA (red) exhibits sharp degradation beyond $E = 3\text{-}4$. The drops are particularly severe on CoLA ($\sim$78% to $\sim$70%), STS-B ($\sim$80% to $\sim$74%), and MRPC ($\sim$83% to

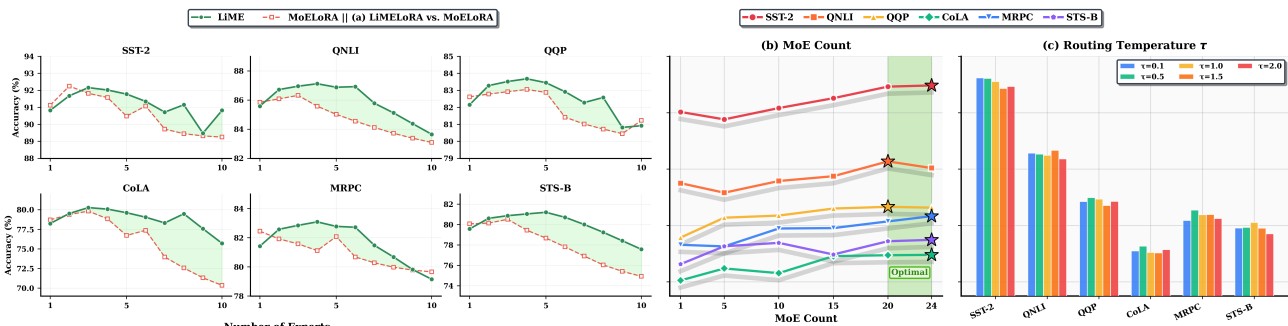

*Figure 6.* **Scaling and temperature ablations.** (a) LiME vs MoELoRA with increasing experts: LiME (green) maintains stable performance while MoELoRA (red) degrades significantly beyond $E = 3$–$4$, with drops of 8–10% on CoLA, MRPC, and STS-B due to overfitting. Green shading indicates where LiME outperforms. (b) MoE layer count: performance improves with more LiME layers, with optimal range at 20–24 layers (green region). (c) Routing temperature: $\tau = 0.5$ (green) achieves the best performance across tasks.

$\sim$73%)—losses of 8–10 percentage points.

This divergence stems from overfitting: MoELoRA replicates full adapter modules per expert ($E \times |\phi|$ parameters), causing each expert to see fewer training samples as $E$ increases. With insufficient data per expert, MoELoRA overfits to training noise rather than learning meaningful specialization. LiME avoids this by using lightweight modulation vectors ($E \times d$ parameters) while the shared PEFT backbone trains on all data, preventing fragmentation.

> **Finding:** *MoELoRA suffers 8–10% accuracy drops beyond $E = 4$ due to overfitting from parameter explosion. LiME's lightweight design maintains stable performance up to $E = 10$, demonstrating robustness to expert scaling.*

> **Additional ablation studies are provided in the Appendix: CKA (§B.1), Auto Top-K motivation (§F.1), n-gram window size (§F.2), fixed Top-K selection (§F.3), number of experts (§F.4), MoE layer count (§F.5), routing temperature (§F.6), expert utilization analysis (§F.7), expert modulator initialization (§F.8), target module selection (§F.9), shared modulator design (§F.10), and routing representation visualization (§F.11).**

## 5. Conclusion

We presented LiME, a lightweight approach to combining MoE with PEFT. By replacing separate expert adapters with lightweight modulation vectors and deriving routing from existing frozen and adapted representations, LiME achieves expert specialization with zero routing parameters and minimal overhead. Our theoretical analysis establishes that more experts preserve more task-relevant information, that modulation can approximate full expert-specific PEFT with bounded error, and that last-token routing is optimal for causal n-gram windows. Experiments on MMT-47 demonstrate that LiME achieves competitive performance compared to MoE-PEFT baselines while using up to $4\times$ fewer trainable parameters and 29% faster training. LiME is compatible with any PEFT method—and efficiency make it a practical choice for multi-task adaptation of large models.

## Impact Statement

This paper presents LiME, a method for efficient multi-task adaptation of large pre-trained models. Our work aims to advance the field of Machine Learning by reducing the computational and memory costs associated with fine-tuning, making model adaptation more accessible to researchers and practitioners with limited resources. The efficiency gains may also contribute to reduced energy consumption in training workflows.

As with any method that facilitates model adaptation, LiME could potentially be used to fine-tune models for harmful applications. However, this concern is not specific to our approach and applies broadly to the field of parameter-efficient fine-tuning. We do not foresee any unique ethical risks arising from our contributions beyond those inherent to advancing general-purpose machine learning techniques.

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

## Appendix Contents

# A. Proof of Theorem 1

> *Intuition:* *Adding more experts refines the input space partitioning. Since the coarser partition's output can be recovered from the finer partition (via a deterministic function), the finer partition cannot lose information about the label $Y$.*

**Theorem A.1** (More experts cannot reduce label information under refinement). *Let $X : \Omega \to \mathbb{R}^d$ and $Y : \Omega \to \mathcal{Y}$ with $Y$ discrete and $H(Y) < \infty$. Fix $n \geq 2$. Let $r_{n-1} : \mathbb{R}^d \to \{1, \ldots, n-1\}$ and $r_n : \mathbb{R}^d \to \{1, \ldots, n\}$ be measurable routers, and let*

$$I_{n-1} := r_{n-1}(X), \qquad Z_{n-1} := A_{I_{n-1}}^{(n-1)} X,$$

$$I_n := r_n(X), \qquad Z_n := A_{I_n}^{(n)} X,$$

*where $A_j^{(n-1)}, A_e^{(n)} \in \mathbb{R}^{d \times d}$ are fixed linear maps. Assume:*

1. *(Factorization-on-support) For each $e \in \{1, \ldots, n\}$ there exists $R_e \in \mathbb{R}^{d \times d}$ such that for all $x \in \mathsf{Supp}(X \mid I_n = e)$,*

$$A_{r_{n-1}(x)}^{(n-1)} x = R_e A_e^{(n)} x.$$

2. *(Identifiability) There exists measurable $\hat{e} : \mathbb{R}^d \to \{1, \ldots, n\}$ such that*

$$I_n = \hat{e}(Z_n) \quad a.s.$$

*Then*

$$I(Y; Z_n) \geq I(Y; Z_{n-1}).$$

*Proof.* Define $h : \mathbb{R}^d \to \mathbb{R}^d$ by

$$h(z) := R_{\hat{e}(z)} z.$$

Since $\hat{e}$ is measurable and each $z \mapsto R_e z$ is continuous, $h$ is measurable.

Let

$$N := \{\omega \in \Omega : I_n(\omega) \neq \hat{e}(Z_n(\omega))\}.$$

By identifiability, $\mathbb{P}(N) = 0$.

For each $e \in \{1, \ldots, n\}$ let $S_e := \mathsf{Supp}(X \mid I_n = e)$. Then

$$\mathbb{P}(X \in S_e \mid I_n = e) = 1$$

$$\mathbb{P}(X \notin S_e \mid I_n = e) = 0$$

Define

$$M := \{\omega \in \Omega : X(\omega) \notin S_{I_n(\omega)}\}$$

Then

$$\begin{aligned}
\mathbb{P}(M) &= \sum_{e=1}^{n} \mathbb{P}\big(M \cap \{I_n = e\}\big) \\
&= \sum_{e=1}^{n} \mathbb{P}\big(\{X \notin S_e\} \cap \{I_n = e\}\big) \\
&= \sum_{e=1}^{n} \mathbb{P}(I_n = e) \, \mathbb{P}(X \notin S_e \mid I_n = e) \\
&= 0
\end{aligned}$$

Fix $\omega \in \Omega \setminus (N \cup M)$ and write

$$x := X(\omega), \quad e := I_n(\omega), \quad z := Z_n(\omega).$$

Then
$$z = A_e^{(n)} x, \qquad e = \hat{e}(z), \qquad x \in S_e.$$

By the factorization-on-support assumption (applied to this $e$ and $x \in S_e$),

$$A_{r_{n-1}(x)}^{(n-1)} x = R_e A_e^{(n)} x.$$

Using $r_{n-1}(x) = r_{n-1}(X(\omega)) = I_{n-1}(\omega)$ and $A_e^{(n)} x = z$,

$$Z_{n-1}(\omega) = A_{I_{n-1}(\omega)}^{(n-1)} X(\omega) = R_e\, z.$$

Using $e = \hat{e}(z)$,

$$Z_{n-1}(\omega) = R_{\hat{e}(z)} z = h(z) = h(Z_n(\omega)).$$

Thus,

$$Z_{n-1} = h(Z_n) \quad \text{a.s.}$$

Let $U := Z_n$ and $V := Z_{n-1}$. Then $V = h(U)$ almost surely. Hence

$$\sigma(V) \subseteq \sigma(U).$$

Therefore,

$$H(Y \mid U) \le H(Y \mid V).$$

It follows that

$$
\begin{align}
I(Y; U) &= H(Y) - H(Y \mid U) \tag{6}\\
&\ge H(Y) - H(Y \mid V) \tag{7}\\
&= I(Y; V). \tag{8}
\end{align}
$$

Substituting $U = Z_n$ and $V = Z_{n-1}$ yields

$$I(Y; Z_n) \ge I(Y; Z_{n-1}).$$

$\square$

**Corollary 1** By induction, for any $n \ge 1$:

$$I(Y; Z_n) \ge I(Y; Z_{n-1}) \ge \cdots \ge I(Y; Z_1)$$

Thus, MoE with $n$ experts preserves at least as much label information as a single-expert model.

> ***Practical Implication:*** *While more experts can preserve more information, realizing this benefit requires sufficient training data per expert. Traditional MoE-PEFT suffers from data fragmentation as E increases. LiME mitigates this via lightweight expert vectors and a shared PEFT backbone.*

# B. Proof of Theorem 2

> **Overview:** *This theorem shows that LiME (shared PEFT with expert modulation) can approximate traditional MoE-PEFT (separate adapters per expert) with bounded performance gap. The key insight is that if the approximation error $\bar{\varepsilon}$ is small, the optimal risk differs by at most $\mathcal{O}(\bar{\varepsilon})$.*

**Theorem B.1** (Smoothed quantitative equivalence: expert-specific PEFT vs. LiME). *Let $(\Omega, \mathcal{F}, \mathbb{P})$ be a probability space. Let $d \in \mathbb{N}$, $n \geq 2$, and $\sigma > 0$. Let $X : \Omega \to \mathbb{R}^d$ and $Y : \Omega \to \mathcal{Y}$ be random variables, where $\mathcal{Y}$ is a standard Borel space. Let $\mathbf{W}_0 \in \mathbb{R}^{d \times d}$ be fixed. Let $r : \mathbb{R}^d \to \{1, \ldots, n\}$ be measurable and define $E := r(X)$.*

*For each $e \in \{1, \ldots, n\}$ let $g_{\phi_\mathbf{e}} : \mathbb{R}^d \to \mathbb{R}^d$ be measurable and define the **traditional MoE-PEFT** output (separate adapters per expert):*

$$Z_{MoE} := \mathbf{W}_0 X + g_{\phi_\mathbf{E}}(X) \in \mathbb{R}^d.$$

*Let $g_\phi : \mathbb{R}^d \to \mathbb{R}^d$ be measurable and for each $e$ let $\mathbf{p}_e \in \mathbb{R}^d$. Define the **LiME** output (shared PEFT with expert modulation):*

$$Z_{LiME} := \mathbf{W}_0 X + \big(g_\phi(X) \odot \mathbf{p}_E\big) \in \mathbb{R}^d.$$

*Assume there exists $\bar{\varepsilon} < \infty$ such that*

$$\big\| g_{\phi_\mathbf{E}}(X) - \big(g_\phi(X) \odot \mathbf{p}_E\big) \big\|_2 \leq \bar{\varepsilon} \quad a.s.$$

*Let $N \sim \mathcal{N}(0, \sigma^2 I_d)$ be independent of $(X, E, Y)$ and define*

$$U_{MoE}^{(\sigma)} := Z_{MoE} + N, \qquad U_{LiME}^{(\sigma)} := Z_{LiME} + N.$$

*Let $\mathcal{A}$ be a standard Borel space and let $\ell : \mathcal{Y} \times \mathcal{A} \to [0, L_{\max}]$ be measurable. For a random element $U$ taking values in $\mathbb{R}^d$ define*

$$\mathcal{R}^*(U) := \inf_\delta \mathbb{E}\big[\ell\big(Y, \delta(U)\big)\big],$$

*where the infimum is over all measurable $\delta : \mathbb{R}^d \to \mathcal{A}$. Then*

$$\big|\mathcal{R}^*(U_{MoE}^{(\sigma)}) - \mathcal{R}^*(U_{LiME}^{(\sigma)})\big| \leq L_{\max} \cdot \frac{\bar{\varepsilon}}{2\sigma}.$$

*Proof.* Define measurable maps $m_{\text{MoE}}, m_{LiME} : \mathbb{R}^d \times \{1, \ldots, n\} \to \mathbb{R}^d$ by

$$m_{\text{MoE}}(x, e) := \mathbf{W}_0 x + g_{\phi_\mathbf{e}}(x)$$

$$m_{LiME}(x, e) := \mathbf{W}_0 x + \big(g_\phi(x) \odot \mathbf{p}_e\big)$$

Then

$$Z_{\text{MoE}} = m_{\text{MoE}}(X, E), \qquad Z_{LiME} = m_{LiME}(X, E),$$

and

$$U_{\text{MoE}}^{(\sigma)} = m_{\text{MoE}}(X, E) + N$$

$$U_{LiME}^{(\sigma)} = m_{LiME}(X, E) + N$$

Let $\Sigma := \sigma^2 I_d$. Since $N$ is independent of $(X, E, Y)$,

$$\mathcal{L}\big(U_{\text{MoE}}^{(\sigma)} \mid X, E\big) = \mathcal{N}\big(m_{\text{MoE}}(X, E), \Sigma\big)$$

$$\mathcal{L}\big(U_{LiME}^{(\sigma)} \mid X, E\big) = \mathcal{N}\big(m_{LiME}(X, E), \Sigma\big).$$

For any $\mu_1, \mu_2 \in \mathbb{R}^d$,

$$\text{KL}(\mathcal{N}(\mu_1, \Sigma) \,\|\, \mathcal{N}(\mu_2, \Sigma)) = \frac{\|\mu_1 - \mu_2\|_2^2}{2\sigma^2}.$$

Therefore,

$$\text{KL}\Big(\mathcal{L}\big(U_{\text{MoE}}^{(\sigma)} \mid X, E\big) \,\Big\|\, \mathcal{L}\big(U_{LiME}^{(\sigma)} \mid X, E\big)\Big) =$$

$$\frac{\|m_{\text{MoE}}(X, E) - m_{LiME}(X, E)\|_2^2}{2\sigma^2}.$$

By Pinsker's inequality,

$$\text{TV}(P, Q) \leq \sqrt{\tfrac{1}{2} \text{KL}(P\|Q)}.$$

Hence,

$$\text{TV}\Big(\mathcal{L}\big(U_{\text{MoE}}^{(\sigma)} \mid X, E\big), \mathcal{L}\big(U_{LiME}^{(\sigma)} \mid X, E\big)\Big) \leq$$

$$\frac{\|m_{\text{MoE}}(X, E) - m_{LiME}(X, E)\|_2}{2\sigma}.$$

Since

$$m_{\text{MoE}}(X, E) - m_{LiME}(X, E) =$$
$$g_{\phi_{\mathbf{E}}}(X) - \big(g_\phi(X) \odot \mathbf{p}_E\big)$$

the assumption implies

$$\text{TV}\Big(\mathcal{L}\big(U_{\text{MoE}}^{(\sigma)} \mid X, E\big), \mathcal{L}\big(U_{LiME}^{(\sigma)} \mid X, E\big)\Big) \leq \frac{\overline{\varepsilon}}{2\sigma} \quad \text{a.s.}$$

Let $P^{\text{MoE}}$ and $P^{LiME}$ denote the joint laws of $(Y, U_{\text{MoE}}^{(\sigma)})$ and $(Y, U_{LiME}^{(\sigma)})$. For any measurable $B \subseteq \mathcal{Y} \times \mathbb{R}^d$,

$$\big|P^{\text{MoE}}(B) - P^{LiME}(B)\big|$$
$$= \Big|\mathbb{E}\Big[\mathbb{P}\big((Y, U_{\text{MoE}}^{(\sigma)}) \in B \mid X, E, Y\big)$$
$$-\mathbb{P}\big((Y, U_{LiME}^{(\sigma)}) \in B \mid X, E, Y\big)\Big]\Big|$$
$$\leq \mathbb{E}\Big[\text{TV}\Big(\mathcal{L}\big(U_{\text{MoE}}^{(\sigma)} \mid X, E\big),$$
$$\mathcal{L}\big(U_{LiME}^{(\sigma)} \mid X, E\big)\Big)\Big].$$

Taking the supremum over $B$ gives

$$\text{TV}\big(P^{\text{MoE}}, P^{LiME}\big) \leq$$

$$\mathbb{E}\Big[\text{TV}\Big(\mathcal{L}\big(U_{\text{MoE}}^{(\sigma)} \mid X, E\big), \mathcal{L}\big(U_{LiME}^{(\sigma)} \mid X, E\big)\Big)\Big] \leq \frac{\overline{\varepsilon}}{2\sigma}.$$

Fix any measurable $\delta : \mathbb{R}^d \to \mathcal{A}$ and define

$$\mathcal{R}_{\text{MoE}}(\delta) := \mathbb{E}\big[\ell\big(Y, \delta(U_{\text{MoE}}^{(\sigma)})\big)\big]$$
$$\mathcal{R}_{LiME}(\delta) := \mathbb{E}\big[\ell\big(Y, \delta(U_{LiME}^{(\sigma)})\big)\big].$$

Define

$$f(y, u) := \frac{\ell\big(y, \delta(u)\big)}{L_{\max}},$$

so that $0 \leq f \leq 1$. Then

$$\big|\mathcal{R}_{\text{MoE}}(\delta) - \mathcal{R}_{LiME}(\delta)\big|$$
$$= L_{\max} \cdot \big|\mathbb{E}_{P^{\text{MoE}}}[f] - \mathbb{E}_{P^{LiME}}[f]\big|$$
$$\leq L_{\max} \cdot \text{TV}\big(P^{\text{MoE}}, P^{LiME}\big)$$
$$\leq L_{\max} \cdot \frac{\overline{\varepsilon}}{2\sigma}. \tag{9}$$

Taking the infimum over $\delta$ yields

$$\big|\mathcal{R}^*(U_{\text{MoE}}^{(\sigma)}) - \mathcal{R}^*(U_{LiME}^{(\sigma)})\big| \leq L_{\max} \cdot \frac{\overline{\varepsilon}}{2\sigma}.$$

$\square$

**Interpretation:** *The bound $L_{\max} \cdot \frac{\overline{\varepsilon}}{2\sigma}$ shows that if the modulation $g_\phi(x) \odot \mathbf{p}_e$ closely approximates the expert-specific adapter $g_{\phi_e}(x)$, then LiME achieves nearly the same optimal risk as traditional MoE-PEFT. Our empirical results (CKA $\approx 0.935$) confirm this approximation is tight in practice.*

*Table 3.* CKA (Centered Kernel Alignment) similarity scores between LiMELoRA and MoELoRA across layers and GLUE datasets. Higher values indicate stronger representational similarity.

| Layer | SST-2 | QNLI | QQP | CoLA | MRPC | STS-B | Mean |
|---|---|---|---|---|---|---|---|
| Layer 0 | 0.994 | 0.995 | 0.995 | 0.993 | 0.995 | 0.994 | **0.994** |
| Layer 1 | 0.981 | 0.986 | 0.986 | 0.984 | 0.985 | 0.984 | **0.984** |
| Layer 2 | 0.965 | 0.968 | 0.956 | 0.977 | 0.977 | 0.977 | **0.970** |
| Layer 3 | 0.942 | 0.947 | 0.945 | 0.950 | 0.956 | 0.954 | **0.949** |
| Layer 4 | 0.955 | 0.944 | 0.954 | 0.960 | 0.949 | 0.949 | **0.952** |
| Layer 5 | 0.954 | 0.939 | 0.938 | 0.959 | 0.940 | 0.937 | **0.945** |
| Layer 6 | 0.947 | 0.940 | 0.942 | 0.945 | 0.940 | 0.940 | **0.942** |
| Layer 7 | 0.926 | 0.944 | 0.933 | 0.926 | 0.932 | 0.939 | **0.933** |
| Layer 8 | 0.929 | 0.935 | 0.950 | 0.899 | 0.953 | 0.946 | **0.935** |
| Layer 9 | 0.866 | 0.886 | 0.892 | 0.860 | 0.899 | 0.876 | **0.880** |
| Layer 10 | 0.951 | 0.954 | 0.947 | 0.951 | 0.931 | 0.930 | **0.941** |
| Layer 11 | 0.920 | 0.925 | 0.927 | 0.932 | 0.923 | 0.933 | **0.927** |
| Layer 12 | 0.945 | 0.932 | 0.922 | 0.943 | 0.920 | 0.920 | **0.930** |
| Layer 13 | 0.950 | 0.946 | 0.938 | 0.940 | 0.904 | 0.940 | **0.936** |
| Layer 14 | 0.955 | 0.932 | 0.927 | 0.950 | 0.922 | 0.941 | **0.938** |
| Layer 15 | 0.951 | 0.934 | 0.928 | 0.941 | 0.926 | 0.941 | **0.937** |
| Layer 16 | 0.942 | 0.954 | 0.913 | 0.924 | 0.930 | 0.928 | **0.932** |
| Layer 17 | 0.915 | 0.908 | 0.867 | 0.899 | 0.872 | 0.872 | **0.889** |
| Layer 18 | 0.925 | 0.916 | 0.903 | 0.920 | 0.912 | 0.896 | **0.912** |
| Layer 19 | 0.900 | 0.918 | 0.890 | 0.919 | 0.887 | 0.867 | **0.897** |
| Layer 20 | 0.949 | 0.947 | 0.919 | 0.936 | 0.922 | 0.914 | **0.931** |
| Layer 21 | 0.931 | 0.896 | 0.903 | 0.929 | 0.903 | 0.878 | **0.907** |
| Layer 22 | 0.930 | 0.923 | 0.907 | 0.939 | 0.909 | 0.901 | **0.918** |
| Layer 23 | 0.970 | 0.971 | 0.964 | 0.954 | 0.954 | 0.945 | **0.960** |
| **Mean** | **0.942** | **0.939** | **0.931** | **0.939** | **0.931** | **0.929** | **0.935** |

## B.1. CKA Analysis

To provide empirical validation of Theorem 2, we analyze CKA (Centered Kernel Alignment) similarity between LiMELoRA and MoELoRA representations at each layer. CKA measures representational similarity independent of orthogonal transformations and isotropic scaling, making it suitable for comparing learned representations across architectures. Table 3 shows CKA scores for layers 0–23 across all GLUE tasks.

Several observations emerge. First, CKA remains consistently high ($> 0.88$) across all layers, indicating that LiME approximates full MoE-PEFT throughout the network, not just at specific layers. Second, early layers (0–3) show the highest CKA (0.949–0.994), suggesting that expert modulation most closely matches separate adapters when processing lower-level features. Third, certain middle layers (9, 17, 19) show relatively lower CKA (0.880–0.897), which may reflect increased task-specific specialization where the modulation approximation is less tight. Fourth, the final layer (23) recovers high CKA (0.960), indicating that output representations converge despite intermediate differences—consistent with our theoretical bound that approximation error remains controlled. The overall mean CKA of 0.935 across all layers and tasks confirms that LiME effectively approximates expert-specific PEFT, empirically supporting Theorem 2.

# C. Proof of Theorem 3

*Intuition:* *In causal (autoregressive) models, each position can only attend to previous positions. Therefore, later hidden states within an n-gram window accumulate more context, and adding more context cannot reduce information about the target $Y$.*

We first establish the necessary definitions, then prove the main result.

**Definition C.1** (Mutual Information). For random variables $A$ and $B$:

$$I(A; B) = H(B) - H(B \mid A) \tag{10}$$

**Definition C.2** (Conditional Mutual Information). For random variables $A$, $B$, and $C$:

$$I(A; B \mid C) = H(B \mid C) - H(B \mid A, C) \tag{11}$$

**Theorem C.3** (Routing Informativeness in Causal N-gram Windows). *Let $h_1, h_2, \ldots, h_n \in \mathbb{R}^d$ be hidden states within an n-gram window under causal attention, and let $Y$ denote the task objective. Then the routing informativeness is maximized at the window's final position:*

$$I(Y; h_n) \geq I(Y; h_{n-1}) \geq \cdots \geq I(Y; h_1) \tag{12}$$

*Proof.* Under causal attention, hidden state $h_t$ aggregates information from all preceding positions. Define the cumulative context $h_{1:t} = (h_1, h_2, \ldots, h_t)$. By the data processing inequality, since $h_t$ is a deterministic function of $h_{1:t}$:

$$I(Y; h_t) \leq I(Y; h_{1:t}) \tag{13}$$

It suffices to show that $I(Y; h_{1:t+1}) \geq I(Y; h_{1:t})$ for an arbitrary index $t \in \{1, 2, \ldots, n-1\}$.

By the chain rule for mutual information:

$$I(Y; h_{1:t+1}) = I(Y; h_{1:t}) + I(Y; h_{t+1} \mid h_{1:t}) \tag{14}$$

To verify this, we expand the right-hand side using Definitions 1 and 2:

$$\begin{aligned}
I(Y; h_{1:t}) &+ I(Y; h_{t+1} \mid h_{1:t}) \\
&= \big[H(Y) - H(Y \mid h_{1:t})\big] \\
&\quad + \big[H(Y \mid h_{1:t}) - H(Y \mid h_{1:t}, h_{t+1})\big] \\
&= H(Y) - H(Y \mid h_{1:t}, h_{t+1}) \\
&= I(Y; h_{1:t+1})
\end{aligned} \tag{15}$$

Since conditional mutual information is non-negative (Cover, 1999):

$$I(Y; h_{t+1} \mid h_{1:t}) \geq 0 \tag{16}$$

it follows immediately that:

$$I(Y; h_{1:t+1}) \geq I(Y; h_{1:t}) \tag{17}$$

Since causal attention ensures $h_t$ encodes $h_{1:t}$, we have $I(Y; h_t) = I(Y; h_{1:t})$. Applying this equality and the above inequality iteratively for $t = n-1, n-2, \ldots, 1$ completes the proof. □

*Implication for Routing:* *In causal models, the last hidden state $h_n$ of each n-gram window has access to all preceding positions via causal attention, capturing the most task-relevant information. This justifies LiME's last-token routing strategy for n-gram windowed routing.*

# D. Expert Selection Strategies

Let $w(x) = [w_1(x), w_2(x), \ldots, w_E(x)]$ denote the routing probability distribution over $E$ experts for input $x$, obtained via softmax over routing scores:

$$w_i(x) = \frac{\exp(s_i(x))}{\sum_{j=1}^{E} \exp(s_j(x))}.$$

We define $w_{[i]}(x)$ as the $i$-th largest probability, i.e., $w_{[1]}(x) \geq w_{[2]}(x) \geq \cdots \geq w_{[E]}(x)$. Each selection strategy defines a set of activated experts $\mathcal{S}(x) \subseteq \{1, \ldots, E\}$. Given a selected set $\mathcal{S}(x)$, the renormalized weights are computed as:

$$\tilde{w}_i(x) = \begin{cases} \dfrac{w_i(x)}{\sum_{j \in \mathcal{S}(x)} w_j(x)} & \text{if } i \in \mathcal{S}(x) \\ 0 & \text{otherwise} \end{cases}$$

Each strategy below differs only in how $\mathcal{S}(x)$ is defined.

## D.1. Fixed Top-K

The most common approach selects a fixed number of $k$ experts with highest routing probabilities, regardless of the distribution shape.

$$\mathcal{S}_k(x) = \big\{ i : w_i(x) \geq w_{[k]}(x) \big\}$$

*Properties:* Constant computational cost with exactly $k$ experts activated. However, it ignores the confidence signal—a dominant expert at 90% is treated the same as a marginal leader at 15%.

## D.2. Absolute Threshold

This strategy selects all experts whose probability exceeds a fixed threshold $\eta \in (0, 1)$.

$$\mathcal{S}_\eta(x) = \{ i : w_i(x) \geq \eta \}$$

To ensure at least one expert is selected, we enforce:

$$\mathcal{S}_\eta(x) = \begin{cases} \{ i : w_i(x) \geq \eta \} & \text{if } \exists\, i : w_i(x) \geq \eta \\ \{ \arg\max_i w_i(x) \} & \text{otherwise} \end{cases}$$

*Properties:* Adaptive to routing confidence. However, not scale-invariant with respect to $E$. For uniform distribution, $w_i = 1/E$, so the threshold $\eta$ must satisfy $\eta < 1/E$ to select any expert. This makes $\eta$ difficult to tune across different expert counts.

## D.3. Entropy-Based Selection

This strategy uses the entropy of the routing distribution to determine the number of activated experts. High entropy indicates uncertainty (select more experts); low entropy indicates confidence (select fewer).

The entropy of the routing distribution is:

$$H(x) = -\sum_{i=1}^{E} w_i(x) \log w_i(x)$$

Normalized by maximum entropy $H_{\max} = \log E$ (achieved by uniform distribution):

$$\bar{H}(x) = \frac{H(x)}{\log E} \in [0, 1]$$

The number of experts is determined by linear interpolation:

$$k(x) = k_{\min} + \big\lfloor (k_{\max} - k_{\min}) \cdot \bar{H}(x) \big\rfloor$$

The selected set is then:

$$\mathcal{S}_H(x) = \big\{ i : w_i(x) \geq w_{[k(x)]}(x) \big\}$$

*Properties:* Theoretically motivated and scale-invariant. However, requires tuning two hyperparameters ($k_{\min}, k_{\max}$), and the linear mapping may not be optimal.

### D.4. Gini-Based Selection

The Gini coefficient measures inequality in the distribution. High Gini indicates dominance by few experts; low Gini indicates uniform spread.

The Gini coefficient for routing probabilities (noting $\sum_i w_i(x) = 1$):

$$G(x) = \frac{1}{2E} \sum_{i=1}^{E} \sum_{j=1}^{E} |w_i(x) - w_j(x)| \in \left[0, 1 - \frac{1}{E}\right]$$

An equivalent formulation using sorted probabilities:

$$G(x) = \frac{2 \sum_{i=1}^{E} i \cdot w_{[E-i+1]}(x)}{E} - \frac{E+1}{E}$$

The number of experts (inverse relationship—high Gini means fewer experts):

$$k(x) = k_{\max} - \left\lfloor (k_{\max} - k_{\min}) \cdot \frac{G(x)}{1 - 1/E} \right\rfloor$$

The selected set is then:

$$\mathcal{S}_G(x) = \big\{ i : w_i(x) \geq w_{[k(x)]}(x) \big\}$$

*Properties:* More sensitive to distribution tails than entropy. However, $O(E^2)$ computation for the double sum (or $O(E \log E)$ with sorting), and requires two hyperparameters.

### D.5. Cumulative Probability Selection

This strategy selects the minimum number of experts whose cumulative probability reaches a target threshold $\rho \in (0, 1]$.

$$k(x) = \min \left\{ k : \sum_{i=1}^{k} w_{[i]}(x) \geq \rho \right\}$$

The selected set contains the top-$k(x)$ experts:

$$\mathcal{S}_\rho(x) = \big\{ i : w_i(x) \geq w_{[k(x)]}(x) \big\}$$

*Properties:* Intuitive interpretation—with $\rho = 0.9$, selected experts capture 90% of the routing signal. Scale-invariant and single hyperparameter. However, requires sorting ($O(E \log E)$) and may over-select on flat distributions (e.g., uniform distribution with $E = 4$ requires all 4 experts to reach $\rho = 0.9$).

### D.6. Top-K with Minimum Gap

This strategy extends fixed top-$k$ by including additional experts that are within a margin $\Delta$ of the $k$-th expert.

$$\mathcal{S}_{k,\Delta}(x) = \big\{ i : w_i(x) \geq w_{[k]}(x) - \Delta \big\}$$

Equivalently:

$$\mathcal{S}_{k,\Delta}(x) = \mathcal{S}_k(x) \cup \big\{ i \,\big|\, w_{[k]}(x) - \Delta \leq w_i(x) < w_{[k]}(x) \big\}$$

*Properties:* Addresses arbitrary cutoffs when experts have similar scores. However, requires tuning two hyperparameters $(k, \Delta)$, and $\Delta$ is not scale-invariant.

## D.7. Relative Threshold (Ours)

We select all experts whose probability is at least a fraction $\theta$ of the maximum probability.

$$\mathcal{S}_\theta(x) = \{i : w_i(x) \geq \theta \cdot w_{[1]}(x)\}$$

where $w_{[1]}(x) = \max_j w_j(x)$.

*Analysis of selected expert count.* Let $r_i(x) = w_i(x)/w_{[1]}(x) \in (0, 1]$ be the relative probability of expert $i$. Then:

$$|\mathcal{S}_\theta(x)| = |\{i : r_i(x) \geq \theta\}|$$

For a peaked distribution where $w_{[1]}(x) \gg w_{[2]}(x)$, we have $r_{[2]}(x) \ll 1$, so only the top expert is selected when $\theta > r_{[2]}(x)$. For a flat distribution where $w_{[1]}(x) \approx w_{[2]}(x) \approx \cdots$, we have $r_i(x) \approx 1$ for all $i$, so most experts are selected.

> **Why Relative Threshold:** *(i) Scale-invariant across different E. (ii) Single hyperparameter $\theta$. (iii) $O(E)$ computation. (iv) Interpretable: $\theta = 0.5$ means "at least half as good as the best." (v) Guarantees at least one expert selected.*

*Table 4.* Comparison of expert selection strategies.

| Strategy | Selection Rule | Hyperparams | Complexity |
|---|---|---|---|
| Fixed Top-K | $w_i(x) \geq w_{[k]}(x)$ | $k$ | $O(E)$ |
| Absolute Threshold | $w_i(x) \geq \eta$ | $\eta$ | $O(E)$ |
| Entropy-Based | $w_i(x) \geq w_{[k(H)]}(x)$ | $k_{\min}, k_{\max}$ | $O(E)$ |
| Gini-Based | $w_i(x) \geq w_{[k(G)]}(x)$ | $k_{\min}, k_{\max}$ | $O(E^2)$ |
| Cumulative Prob. | $\sum_{j \leq i} w_{[j]}(x) \geq \rho$ | $\rho$ | $O(E \log E)$ |
| Top-K + Gap | $w_i(x) \geq w_{[k]}(x) - \Delta$ | $k, \Delta$ | $O(E \log E)$ |
| Relative Threshold (Ours) | $w_i(x) \geq \theta \cdot w_{[1]}(x)$ | $\theta$ | $O(E)$ |

*Table 5.* Comparison of expert selection strategies on GLUE benchmarks. Best results per dataset are **underlined**. Relative threshold ($\theta = 0.7$) achieves the best average performance while using only 1.73 experts on average.

| Strategy | Params | Avg. $|\mathcal{S}|$ | SST-2 | QNLI | QQP | CoLA | MRPC | STS-B |
|---|---|---|---|---|---|---|---|---|
| Fixed Top-K | $k = 1$ | 1.00 | 91.17 | 85.62 | 82.54 | 79.21 | 81.67 | 80.43 |
| Fixed Top-K | $k = 2$ | 2.00 | 91.86 | 86.34 | 83.27 | 79.88 | 82.35 | 81.12 |
| Fixed Top-K | $k = 3$ | 3.00 | 92.08 | **87.14** | 83.62 | 80.15 | 82.74 | 81.49 |
| Fixed Top-K | $k = 4$ | 4.00 | 91.94 | 86.53 | 83.51 | 79.96 | 82.51 | 81.28 |
| Absolute Threshold | $\eta = 0.05$ | 4.23 | 91.52 | 86.18 | 83.14 | 79.42 | 82.03 | 80.76 |
| Absolute Threshold | $\eta = 0.10$ | 2.87 | 91.79 | 86.47 | 83.39 | 79.81 | 82.41 | 81.19 |
| Absolute Threshold | $\eta = 0.15$ | 1.94 | 91.63 | 86.21 | 83.18 | 79.58 | 82.14 | 80.91 |
| Absolute Threshold | $\eta = 0.20$ | 1.36 | 91.08 | 85.54 | 82.47 | 79.03 | 81.52 | 80.29 |
| Entropy-Based | $k \in [1, 2]$ | 1.47 | 91.58 | 86.21 | 83.12 | 79.74 | 82.31 | 80.94 |
| Entropy-Based | $k \in [1, 3]$ | 1.92 | 92.14 | 86.82 | 83.69 | 80.27 | 83.18 | 81.58 |
| Entropy-Based | $k \in [1, 4]$ | 2.41 | 92.36 | 86.97 | 83.87 | 80.43 | **83.17** | 81.79 |
| Entropy-Based | $k \in [2, 4]$ | 2.89 | 92.29 | 86.89 | 83.78 | 80.35 | 82.98 | 81.67 |
| Gini-Based | $k \in [1, 2]$ | 1.51 | 91.54 | 86.17 | 83.08 | 79.69 | 82.26 | 80.89 |
| Gini-Based | $k \in [1, 3]$ | 1.88 | 92.09 | 86.76 | 83.64 | 80.21 | 82.83 | 81.52 |
| Gini-Based | $k \in [1, 4]$ | 2.35 | 92.31 | 86.94 | 83.82 | 80.38 | 83.04 | 81.74 |
| Gini-Based | $k \in [2, 4]$ | 2.94 | 92.24 | 86.85 | 83.71 | 80.29 | 82.91 | 81.61 |
| Cumulative Prob. | $\rho = 0.80$ | 1.86 | 91.71 | 86.32 | 83.26 | 79.79 | 82.39 | 81.13 |
| Cumulative Prob. | $\rho = 0.85$ | 2.24 | 91.89 | 86.51 | 83.46 | 79.98 | 82.58 | 81.31 |
| Cumulative Prob. | $\rho = 0.90$ | 2.68 | 92.01 | 86.64 | 83.58 | 80.11 | 82.69 | 81.41 |
| Cumulative Prob. | $\rho = 0.95$ | 3.31 | 92.07 | 86.69 | 83.63 | **80.62** | 82.73 | 81.46 |
| Relative Threshold | $\theta = 0.3$ | 3.12 | 92.21 | 86.79 | 83.71 | 80.28 | 83.04 | 81.59 |
| Relative Threshold | $\theta = 0.4$ | 2.54 | 92.27 | 86.86 | 83.79 | 80.36 | 82.91 | 81.68 |
| Relative Threshold | $\theta = 0.5$ | 2.18 | 92.31 | 86.91 | 83.84 | 80.41 | 82.96 | 81.74 |
| Relative Threshold | $\theta = 0.6$ | 1.89 | 92.38 | 86.98 | **83.98** | 80.48 | 83.02 | 81.81 |
| Relative Threshold | $\theta = 0.7$ | 1.73 | **92.43** | 87.06 | 83.91 | 80.24 | 83.09 | **81.87** |
| Relative Threshold | $\theta = 0.8$ | 1.42 | 92.18 | 86.74 | 83.67 | 80.23 | 83.18 | 81.54 |

## D.8. Empirical Comparison

Table 5 compares expert selection strategies on GLUE benchmarks. Several observations emerge. First, fixed top-$k$ shows that performance generally improves from $k = 1$ to $k = 3$, but degrades at $k = 4$, suggesting that activating all experts

is suboptimal—not all experts are equally relevant for every input. Second, absolute threshold is sensitive to the choice of $\eta$: too high ($\eta = 0.20$) selects too few experts, while too low ($\eta = 0.05$) activates nearly all, both hurting performance. Third, entropy-based and Gini-based strategies perform similarly, as both measure distribution spread, but require tuning two hyperparameters and incur additional computational overhead. Fourth, cumulative probability works well but tends to over-select on uncertain inputs (avg. $|\mathcal{S}| = 3.31$ at $\rho = 0.95$). Finally, our relative threshold achieves the best overall performance: $\theta = 0.7$ obtains the highest accuracy on SST-2 (92.43) and STS-B (81.87), while using only 1.73 experts on average—fewer than most other adaptive strategies. This confirms that relative threshold effectively balances expressiveness and efficiency: it activates more experts when routing is uncertain and fewer when confident, without the complexity of entropy or Gini computation.

**Key Result:** *Relative threshold ($\theta = 0.7$) achieves best accuracy while using only 1.73 experts on average—fewer than other adaptive strategies, confirming it balances expressiveness and efficiency effectively.*

# E. Motivation for Dual-Use Routing Features

This section gives intuition for our *zero-parameter routing*. The main idea is simple: instead of training a separate router, we compute routing weights from features that the model already produces during the forward pass. Concretely, we reuse a small slice of existing representations for routing, so the same forward computation supports both *adaptation* and *routing*.

**1) Any small slice of a transformer representation still carries global information.** In transformers, each layer mixes information across dimensions through attention projections and feed-forward networks (Vaswani et al., 2017). As a result, each output dimension is influenced by many (often all) input dimensions. This means that even if we take only $E$ dimensions from a $d$-dimensional hidden state (with $E \ll d$), those dimensions still summarize broad information about the input. Therefore, using a small feature slice for routing can still provide enough signal to choose experts.

**2) Pretrained features are redundant, so reserving a small slice for routing is low-risk.** Pretrained representations often contain substantial redundancy. For example, Luo et al. (2023) show that using only $\sim$1% of the most important feature dimensions can recover performance close to using the full representation. This supports our design in two ways. First, when $d \gg E$ (e.g., $d$=4096 and $E$=4), using only $E$ dimensions for routing is unlikely to reduce the model's ability to learn task-specific representations, since most dimensions remain available for adaptation. Second, this same redundancy intuition is consistent with why parameter-efficient methods such as LoRA work well: effective adaptation often does not require full-rank updates, and low-rank parameterization is sufficient for strong performance (Hu et al., 2022).

**3) Sharing features for routing can encourage meaningful grouping.** Using existing representations for routing can also act as a useful inductive bias. If routing depends on the same representations used for adaptation, the model is encouraged to organize its feature space so that inputs with similar semantics (and similar required adaptations) are routed to similar experts. In practice, this helps expert specialization emerge without learning an extra router.

**Empirical evidence.** We support this motivation with ablations. In §4.1, we show that routing quality is similar whether we use leading, central, trailing, or random $E$ dimensions, suggesting that routing signal is not confined to a specific feature region. In §4.1, our parameter-free routing matches a learned router with $d \times E$ parameters per layer, indicating that existing representations provide sufficient routing information. Finally, the visualization in §F.11 shows that inputs assigned to the same expert form clear clusters, suggesting that the routing aligns with meaningful structure in the representation space.

# F. More Ablation Study

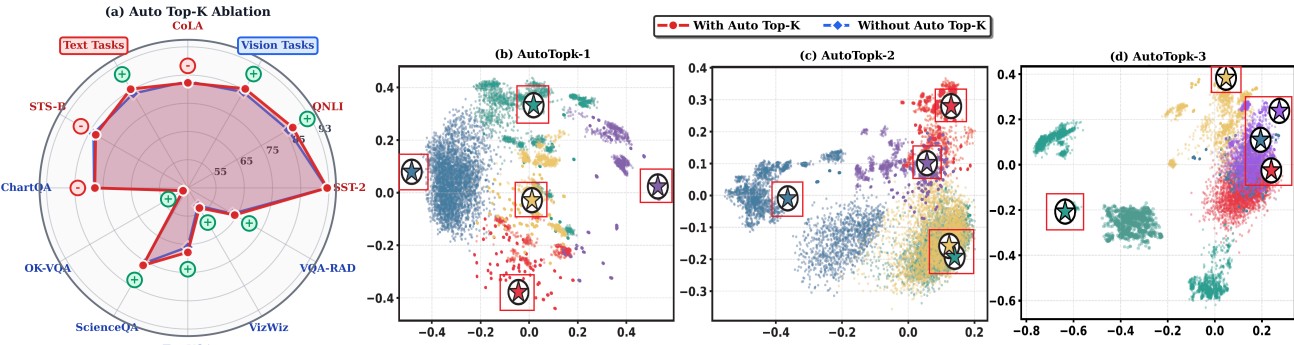

*Figure 7.* Auto Top-K ablation. (a) Auto Top-K outperforms fixed Top-K=2 on most tasks. (b-d) PCA visualizations show inputs naturally require varying numbers of experts: one (b), two (c), or three (d).

## F.1. Auto Top-K Motivation and Analysis

We analyze the motivation for our Auto Top-K strategy over fixed Top-K selection. The core observation is that different inputs inherently require different numbers of experts—a property that fixed Top-K cannot capture.

Figure 7(b-d) visualizes PCA projections of routing representations across different input samples, colored by their dominant expert assignment. These visualizations reveal three distinct patterns:

*(i) Well-separated clusters (Figure 7b).* All five clusters are clearly separated in the representation space. Each input strongly belongs to a single expert, and the routing distribution is peaked (high confidence). In this case, activating additional experts would introduce noise from irrelevant specializations. Auto Top-K naturally selects only the dominant expert since secondary experts fall below the relative threshold $\theta \cdot \max_j w_j(x)$.

*(ii) Partially overlapping clusters (Figure 7c).* Two clusters exhibit significant overlap, indicating that certain inputs share characteristics of multiple experts. The routing distribution for these inputs is flatter, with two experts receiving similar probabilities. Fixed Top-1 would arbitrarily discard one relevant expert, while Auto Top-K includes both since they exceed the relative threshold.

*(iii) Multi-cluster overlap (Figure 7d).* Three clusters overlap substantially, suggesting these inputs benefit from contributions of three experts. This occurs for complex or ambiguous inputs that span multiple specializations. Auto Top-K activates all three relevant experts, while fixed Top-2 would exclude one.

This natural variation in input complexity—some inputs clearly belonging to one expert while others requiring multiple—motivates our relative threshold strategy. By selecting experts satisfying $w_i(x) \geq \theta \cdot \max_j w_j(x)$, Auto Top-K automatically adjusts selection based on routing confidence: confident routing (peaked distribution) activates fewer experts, while uncertain routing (flat distribution) activates more.

Figure 7(a) validates this design empirically. Auto Top-K consistently outperforms fixed Top-K=2 across both text and vision tasks. Improvements are particularly notable on tasks with diverse input complexity: CoLA and QNLI (text), ChartQA and VQA-RAD (vision). The radar chart shows that Auto Top-K matches or exceeds fixed Top-K on nearly all benchmarks, with no significant degradation on any task.

> **Key Insight:** *Input complexity varies naturally—some inputs clearly belong to one expert while others span multiple specializations. Auto Top-K adapts to this variation via confidence-aware selection, whereas fixed Top-K imposes a rigid cardinality that either wastes computation (selecting irrelevant experts) or discards useful experts.*

## F.2. N-gram Window Size (Figure 8a-b)

We analyze the effect of n-gram window size on routing granularity. Window size $n = 1$ corresponds to token-level routing where each token independently selects its own experts, while larger $n$ groups adjacent tokens into windows that share a single routing decision. The motivation for n-gram routing is semantic coherence: adjacent tokens often form meaningful linguistic units (e.g., phrases, named entities) or coherent visual regions that benefit from consistent expert processing. By

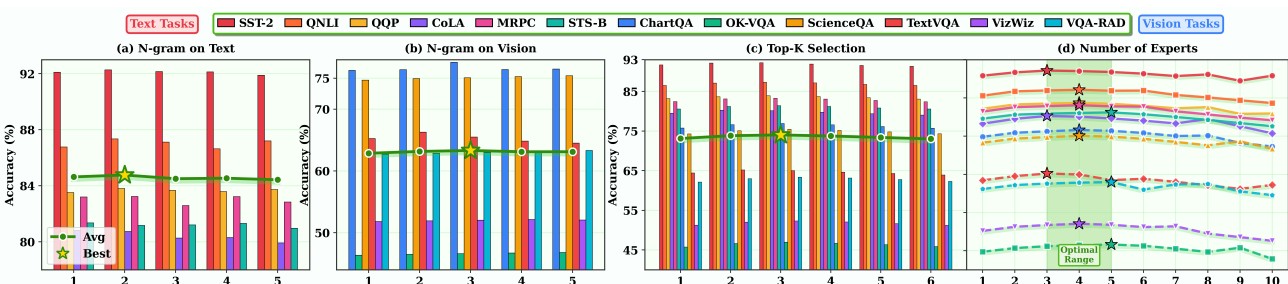

Figure 8. Hyperparameter ablations. (a-b) N-gram window size shows robust performance across text and vision tasks. (c) Fixed Top-K selection peaks at $k = 2$-3; fewer limits expressiveness, more introduces noise. (d) Number of experts: optimal range $E \in [3, 6]$ balances capacity and training efficiency.

sharing routing decisions within windows, n-gram routing encourages locally consistent expert assignments that respect these natural boundaries.

On text tasks (a), performance remains stable across window sizes, with slight variations per task. On vision tasks (b), we observe similar robustness. This stability suggests that the semantic coherence assumption holds—grouping adjacent tokens does not degrade performance while providing implicit regularization by reducing the model's capacity to overfit to spurious token-level patterns.

Note that n-gram routing does not reduce computation: since LiME derives routing from existing representations ($z$ and $\hat{z}$), all token representations must be computed regardless of window size. N-gram windowing only affects how routing probabilities are shared, not the underlying forward pass. Based on these results, we use $n = 3$ as default to capture phrase-level semantics while maintaining accuracy.

### F.3. Fixed Top-K Selection (Figure 8c)

We vary fixed Top-K from $k = 1$ to $k = 6$. With $k = 1$, only the highest-scoring expert is activated, limiting expressiveness. Performance generally improves from $k = 1$ to $k = 2$ or $k = 3$ as additional experts contribute complementary specializations. Beyond $k = 3$, accuracy plateaus or slightly decreases—activating too many experts dilutes the contribution of relevant ones with noise from less relevant experts. On average, $k = 2$ provides the best fixed-cardinality baseline, which we use for comparison against Auto Top-K.

### F.4. Number of Experts (Figure 8d)

We scale the number of experts from $E = 1$ to $E = 10$. Performance improves from $E = 1$ to $E \approx 4$-5, consistent with Theorem 1 that more experts preserve more task-relevant information. The optimal range $E \in [3, 6]$ (highlighted) achieves peak accuracy across most tasks. Beyond $E = 6$, performance plateaus or slightly degrades. This occurs because each expert receives fewer training samples as $E$ increases, making meaningful specialization harder without proportionally more diverse data. We use $E = 4$ as default to balance capacity and training efficiency.

### F.5. MoE Layer Count (Figure 6b).

We vary the number of layers equipped with LiME from 1 to 24 (full model). Performance generally improves with more MoE layers, as each additional layer gains input-specific adaptation capability. The optimal range is $\sim$20-24 layers, suggesting that applying LiME broadly across the model yields the best results. Notably, even with just 5 MoE layers, we observe meaningful improvements over the baseline, indicating that selective application can balance efficiency and performance when computational budget is constrained.

### F.6. Routing Temperature $\tau$ (Figure 6c).

We analyze the effect of temperature $\tau$ in the routing softmax. Lower temperatures ($\tau = 0.1$) produce sharper routing distributions approaching hard selection, while higher temperatures ($\tau = 2.0$) yield softer, more uniform distributions. Results show that $\tau = 0.5$ achieves the best performance across most tasks, providing a balance between decisive routing

(low $\tau$) and smooth gradient flow (high $\tau$). Very low temperature risks gradient sparsity during training, while very high temperature dilutes expert specialization by distributing weight too uniformly.

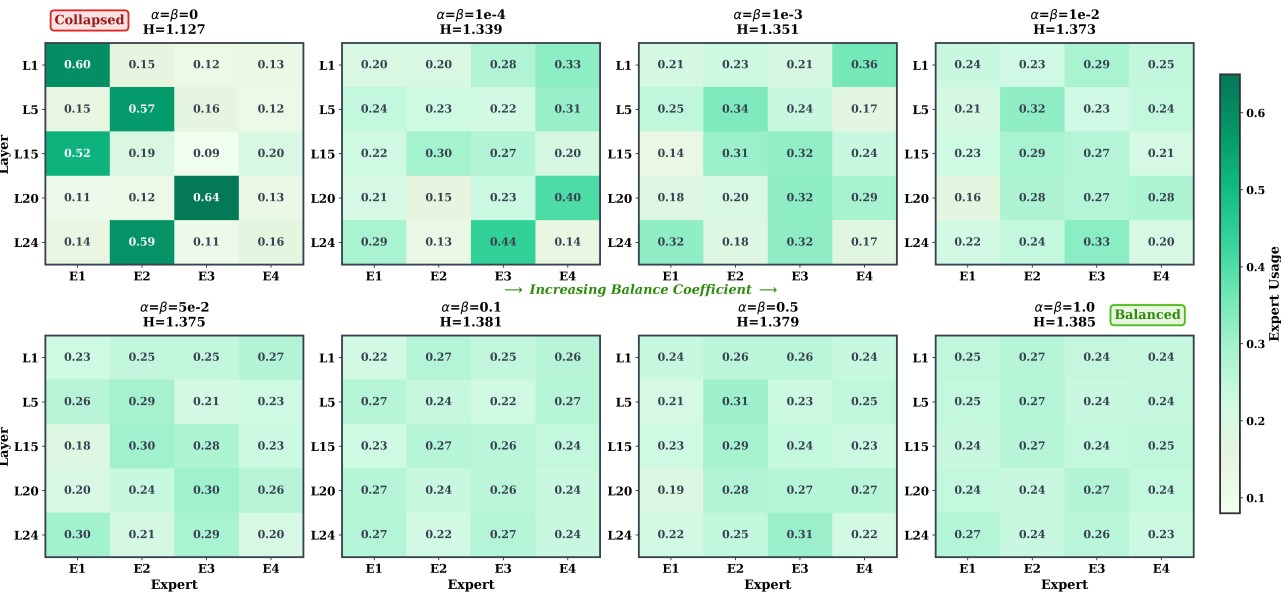

*Figure 9.* Expert utilization heatmaps on SST-2 for varying balance coefficients ($\alpha = \beta$). Without balancing (left), routing collapses to few experts per layer (low entropy $H = 1.127$). Increasing coefficients progressively balances utilization until near-uniform distribution (right, $H = 1.385$). Each cell shows the fraction of tokens routed to expert $E_i$ at layer $L_j$.

### F.7. Expert Utilization vs Balance Coefficient (Figure 9).

We visualize expert usage across layers for varying balance coefficients ($\alpha = \beta$) on SST-2. Each heatmap shows the proportion of tokens routed to each expert (E1-E4) at different layers (L1, L5, L15, L20, L24), with entropy $H$ measuring utilization uniformity.

*Without balancing ($\alpha = \beta = 0$):* Routing collapses to dominant experts. Layer L1 routes 60% of tokens to E1, L5 routes 57% to E2, and deeper layers show similar collapse patterns. The low entropy ($H = 1.127$) confirms severe imbalance—most expert capacity is wasted.

*Small coefficients ($\alpha = \beta \in [10^{-4}, 10^{-2}]$):* Expert utilization gradually balances. At $10^{-4}$, entropy rises to 1.339 with more distributed routing. By $10^{-2}$, entropy reaches 1.373 and usage approaches uniformity ($\sim$0.21-0.33 per expert).

*Moderate to high coefficients ($\alpha = \beta \in [0.05, 1.0]$):* Utilization becomes nearly uniform across all layers and experts ($\sim$0.24-0.27 each). Entropy saturates around $H \approx 1.38$, close to maximum entropy $H_{\max} = \log(4) \approx 1.386$ for 4 experts.

> **Insight:** Balance coefficients $\alpha = \beta \geq 10^{-2}$ effectively prevent expert collapse. However, as shown in Figure 5(b), optimal accuracy occurs at moderate coefficients ($\sim$5e-2), not maximum entropy—some degree of non-uniform specialization is beneficial for task performance.

### F.8. Expert Modulator Initialization (Figure 10a)

We analyze the effect of expert modulator initialization on performance. The modulated output is $\hat{z} \odot \mathcal{P}(x)$, where $\mathcal{P}(x) = \sum_i w_i(x) \cdot \mathbf{p}_i$. Our motivation is that initializing $\mathbf{p}_i \approx \mathbf{1}$ yields $\hat{z} \odot \mathcal{P}(x) \approx \hat{z} \odot \mathbf{1} = \hat{z}$, allowing LiME to behave like standard PEFT at the start of training before expert specialization emerges.

Results confirm this intuition: near-unity initializations outperform other strategies. $\mathcal{U}(0.9, 1.1)$ achieves the best average accuracy (84.6%), followed by $\mathcal{U}(0.8, 1.2)$, $\mathcal{N}(1.0, 0.1^2)$, and All-Ones (84.4% each). Notably, exact All-Ones initialization is not optimal—experts require slight initial diversity to differentiate during training. Wider distributions like $\mathcal{U}(0.7, 1.3)$ perform slightly worse (84.2%), while standard neural network initializations (Xavier, Kaiming) and zero-centered distributions ($\mathcal{N}(0, 0.1^2)$) show degraded performance (83.6–83.9%), as they deviate from the stable PEFT starting point. For the shared modulator, we initialize $\gamma = 0$ so that the shared term $\gamma \cdot (\hat{z} \odot \mathbf{p}_s)$ has no effect at initialization, allowing the model

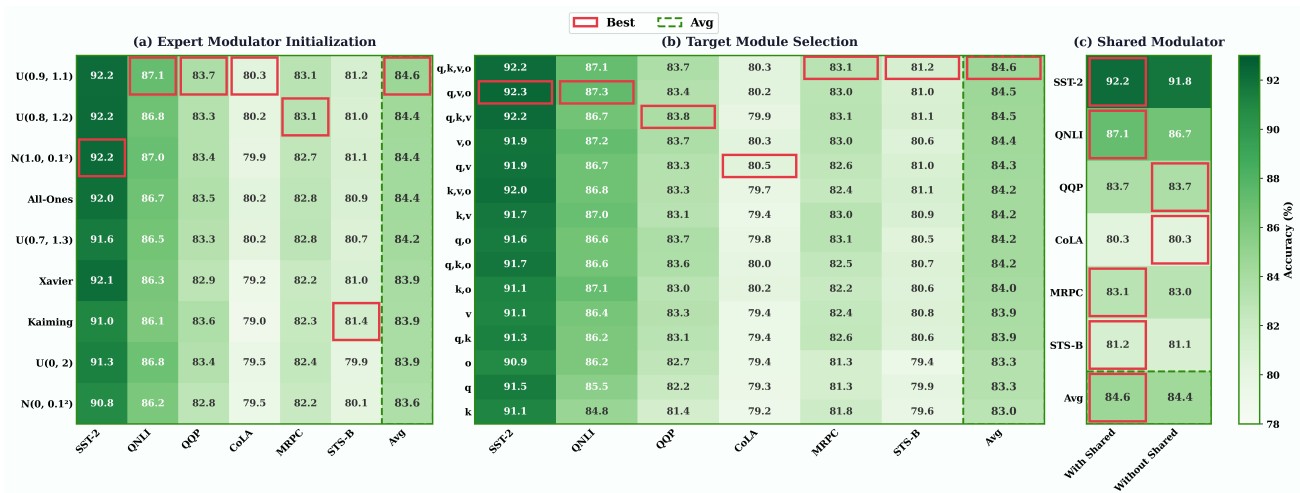

*Figure 10.* **Ablation on initialization, target modules, and shared modulator.** (a) Near-unity initialization ($\mathcal{U}(0.9, 1.1)$) achieves best average accuracy (84.6%). (b) Applying LiME to all attention projections (q,k,v,o) yields optimal performance. (c) Shared modulator provides consistent gains across tasks.

to learn its contribution during training.

> **Finding:** *Near-unity initialization ($\mathcal{U}(0.9, 1.1)$) ensures LiME starts as standard PEFT while providing slight diversity for expert differentiation. Exact ones or zero-centered initializations are suboptimal.*

### F.9. Target Module Selection (Figure 10b)

We investigate which attention projections benefit from LiME. We test all combinations of query (q), key (k), value (v), and output (o) projections. Applying LiME to all four projections (q,k,v,o) achieves the best average accuracy (84.6%), indicating that expert specialization benefits all components of the attention mechanism.

Interestingly, removing the key projection (q,v,o) maintains strong performance (84.5%), suggesting keys contribute least to expert specialization. Configurations including value projection consistently perform well: (q,k,v), (v,o), and (q,v) all achieve ≥84.3%. In contrast, applying LiME to single projections yields lower accuracy, with output-only (o) and query-only (q) at 83.3% and key-only (k) at 83.0%. This suggests that expert modulation benefits from complementary interactions across multiple projections.

> **Finding:** *Applying LiME to all attention projections (q,k,v,o) yields optimal performance. Value projection contributes most to expert specialization, while key projection contributes least.*

### F.10. Shared Modulator (Figure 10c)

We evaluate the contribution of the shared modulator $\mathbf{p}_s$ in the output formulation: $h = z + \hat{z} \odot \mathcal{P}(x) + \gamma \cdot (\hat{z} \odot \mathbf{p}_s)$. The shared modulator provides a task-general adaptation component that complements expert-specific modulation.

Results show consistent improvements with the shared modulator across all tasks. Average accuracy improves from 84.4% (without) to 84.6% (with). Gains are most pronounced on SST-2 (+0.4%) and QNLI (+0.4%), while QQP and CoLA show minimal difference. This suggests the shared modulator captures common adaptation patterns across inputs, reducing the burden on expert modulators to learn both shared and specialized transformations.

> **Finding:** *The shared modulator provides consistent gains (+0.2% average) by capturing task-general adaptation patterns, allowing expert modulators to focus on input-specific specialization.*

### F.11. Routing Representation Analysis (Figure 11).

We visualize t-SNE projections of routing representations across all 24 layers after training on GLUE, colored by dominant expert assignment (Expert 0-3). This analysis examines whether our zero-parameter routing learns meaningful expert specialization at each layer.

*Early layers (0-5):* Representations show partial clustering with some overlap between experts. Experts capture broad input

categories, but boundaries are not yet sharply defined. This is expected as early layers process lower-level features that may be shared across input types.

*Middle layers (6-15):* Clustering patterns vary—some layers (e.g., Layer 9, 13) exhibit well-separated expert clusters, while others (e.g., Layer 5, 12) show more mixing. This layer-specific behavior suggests that different layers benefit from different degrees of expert specialization based on the features they process.

*Later layers (16-23):* Most layers show clearer expert separation, particularly Layers 15, 19, and 23. Later layers process higher-level semantic features where task-specific routing becomes more pronounced. The distinct clustering confirms that experts specialize for different input characteristics.

Across all layers, we observe that inputs routed to the same expert consistently cluster together in the representation space, validating that our routing mechanism—derived from frozen and adapted representations without learned parameters—captures meaningful input similarities suitable for expert assignment.

> **Key Observation:** *Expert assignments are not random—inputs routed to the same expert cluster together across layers. This confirms that $z$ and $\hat{z}$ encode sufficient discriminative signal for routing without dedicated router parameters.*

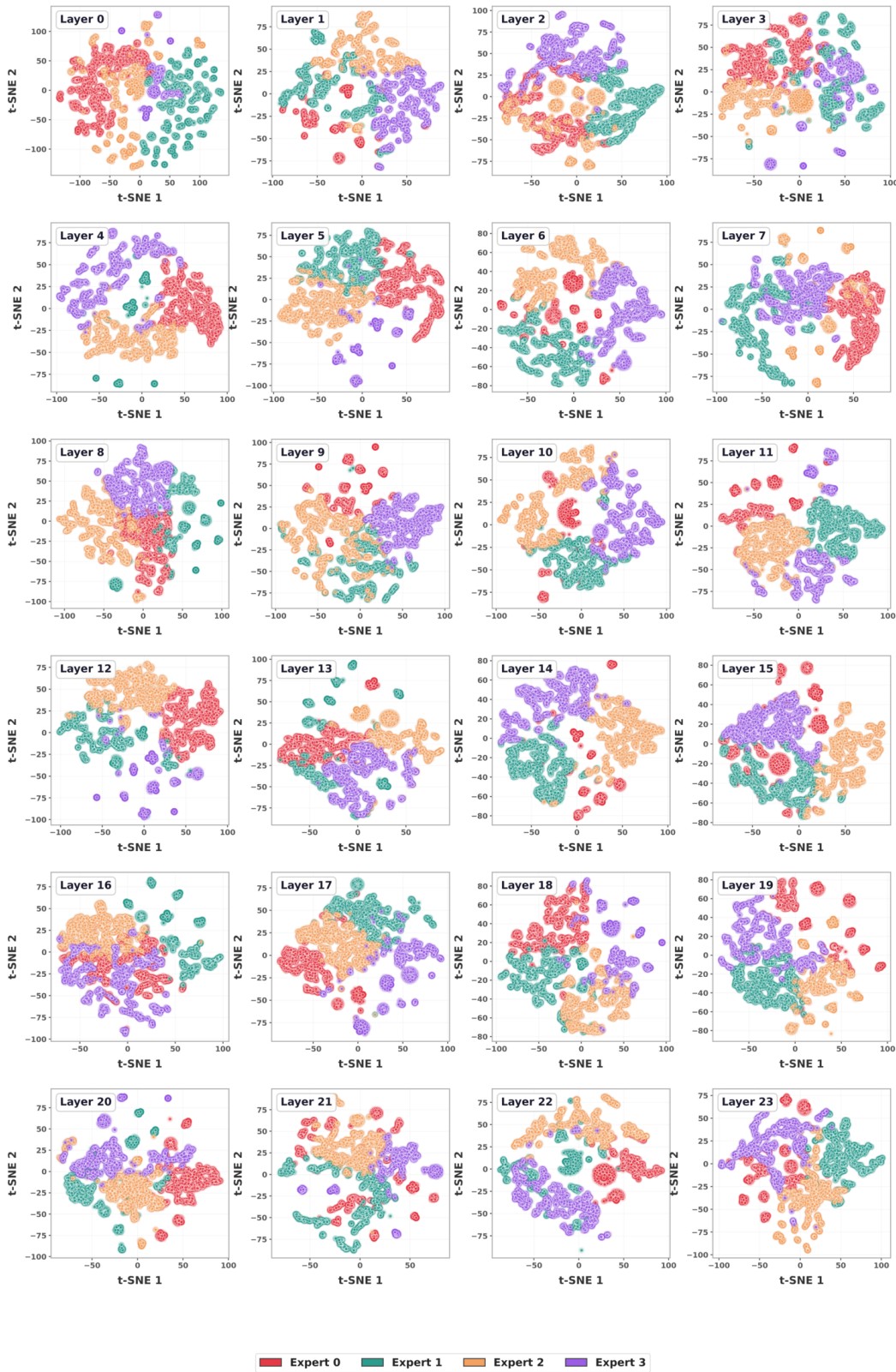

*Figure 11.* t-SNE visualization of routing representations across all 24 layers on GLUE, colored by dominant expert assignment. Inputs routed to the same expert cluster together, confirming that zero-parameter routing learns meaningful specialization. Clustering sharpness varies by layer, with later layers showing more distinct expert separation.

*Table 6.* Results on GLUE benchmarks. Highlighted rows denote our LiME variants.

| Method | SST-2 | QNLI | QQP | CoLA | MRPC | STS-B | Avg |
|---|---|---|---|---|---|---|---|
| PromptTuning | $91.45_{\pm1.38}$ | $61.22_{\pm1.15}$ | $62.35_{\pm1.42}$ | $81.58_{\pm1.76}$ | $70.21_{\pm1.28}$ | $64.67_{\pm1.09}$ | $71.91_{\pm1.35}$ |
| LoRA | $95.27_{\pm1.22}$ | $93.06_{\pm0.94}$ | $87.38_{\pm1.08}$ | $86.02_{\pm1.19}$ | $90.02_{\pm0.73}$ | $92.06_{\pm0.66}$ | $90.64_{\pm0.97}$ |
| DoRA | $94.51_{\pm1.10}$ | $92.22_{\pm0.52}$ | $87.18_{\pm1.23}$ | $86.18_{\pm1.14}$ | $90.82_{\pm1.07}$ | $91.30_{\pm0.85}$ | $90.37_{\pm0.99}$ |
| LoRAFA | $94.89_{\pm0.93}$ | $92.85_{\pm0.68}$ | $87.09_{\pm0.57}$ | $85.71_{\pm1.04}$ | $89.72_{\pm0.89}$ | $91.61_{\pm0.76}$ | $90.31_{\pm0.81}$ |
| SliceFine | $95.43_{\pm0.88}$ | $93.22_{\pm1.11}$ | $87.10_{\pm0.79}$ | $85.95_{\pm1.25}$ | $89.71_{\pm0.84}$ | $91.82_{\pm0.97}$ | $90.54_{\pm0.97}$ |
| MoCLE | $94.83_{\pm0.67}$ | $93.05_{\pm0.49}$ | $87.49_{\pm1.16}$ | $86.66_{\pm0.83}$ | $90.26_{\pm1.02}$ | $91.69_{\pm0.91}$ | $90.66_{\pm0.95}$ |
| MoELoRA | $94.60_{\pm1.09}$ | $\underline{94.44}_{\pm0.76}$ | $\underline{88.41}_{\pm0.63}$ | $\mathbf{87.51}_{\pm1.30}$ | $90.60_{\pm0.86}$ | $91.71_{\pm0.54}$ | $\mathbf{91.21}_{\pm0.87}$ |
| MixLoRA | $95.18_{\pm0.55}$ | $92.94_{\pm0.89}$ | $87.06_{\pm1.03}$ | $86.31_{\pm0.77}$ | $89.84_{\pm1.26}$ | $91.66_{\pm0.64}$ | $90.50_{\pm0.86}$ |
| HydraLoRA | $94.91_{\pm1.14}$ | $93.13_{\pm0.58}$ | $\mathbf{88.49}_{\pm1.29}$ | $86.88_{\pm0.93}$ | $90.79_{\pm0.47}$ | $92.07_{\pm0.86}$ | $91.05_{\pm0.88}$ |
| MoLA | $95.53_{\pm0.81}$ | $93.60_{\pm1.18}$ | $86.90_{\pm0.72}$ | $86.98_{\pm1.27}$ | $90.52_{\pm0.69}$ | $91.73_{\pm0.95}$ | $90.88_{\pm0.94}$ |
| MoRe | $95.42_{\pm0.92}$ | $93.11_{\pm1.07}$ | $87.08_{\pm0.68}$ | $86.73_{\pm1.21}$ | $\underline{90.88}_{\pm0.84}$ | $91.21_{\pm1.12}$ | $90.74_{\pm0.97}$ |
| MoEDoRA | $\mathbf{95.61}_{\pm1.01}$ | $93.79_{\pm0.79}$ | $87.71_{\pm1.17}$ | $86.21_{\pm0.86}$ | $90.75_{\pm1.09}$ | $91.83_{\pm0.92}$ | $90.98_{\pm0.97}$ |
| LiMEPromptTuning | $91.97_{\pm1.14}$ | $61.93_{\pm0.92}$ | $62.94_{\pm1.21}$ | $82.23_{\pm1.48}$ | $70.83_{\pm1.06}$ | $65.19_{\pm0.89}$ | $72.68_{\pm1.12}$ |
| LiMELoRA | $95.36_{\pm0.74}$ | $93.42_{\pm0.63}$ | $87.56_{\pm0.91}$ | $86.86_{\pm0.82}$ | $90.79_{\pm1.05}$ | $\underline{92.10}_{\pm0.88}$ | $91.02_{\pm0.84}$ |
| LiMEDoRA | $95.08_{\pm0.45}$ | $93.58_{\pm0.98}$ | $87.03_{\pm1.11}$ | $87.19_{\pm1.06}$ | $90.67_{\pm0.72}$ | $\mathbf{93.09}_{\pm1.18}$ | $91.11_{\pm0.92}$ |
| LiMELoRAFA | $95.11_{\pm0.83}$ | $93.52_{\pm0.91}$ | $87.94_{\pm0.76}$ | $\underline{87.29}_{\pm0.69}$ | $\mathbf{90.92}_{\pm1.13}$ | $92.03_{\pm0.87}$ | $91.14_{\pm0.87}$ |
| LiMESliceFine | $\underline{95.56}_{\pm0.69}$ | $\mathbf{94.48}_{\pm0.58}$ | $87.68_{\pm0.87}$ | $86.78_{\pm0.94}$ | $90.63_{\pm1.02}$ | $92.02_{\pm0.83}$ | $\underline{91.19}_{\pm0.82}$ |

*Table 7.* Results on commonsense reasoning and question answering benchmarks. Highlighted rows denote our LiME variants.

| Method | BoolQ | PIQA | SIQA | H.Sw. | W.Gra | ARC-e | ARC-c | OBQA | Avg |
|---|---|---|---|---|---|---|---|---|---|
| PromptTuning | $65.58_{\pm1.72}$ | $84.92_{\pm1.38}$ | $74.35_{\pm1.47}$ | $70.63_{\pm1.61}$ | $63.24_{\pm1.24}$ | $92.89_{\pm1.52}$ | $84.62_{\pm1.78}$ | $83.75_{\pm1.55}$ | $77.50_{\pm1.53}$ |
| LoRA | $71.58_{\pm1.33}$ | $86.21_{\pm1.67}$ | $80.94_{\pm0.88}$ | $79.22_{\pm1.06}$ | $84.19_{\pm0.72}$ | $93.38_{\pm1.42}$ | $85.72_{\pm1.61}$ | $89.14_{\pm1.48}$ | $83.80_{\pm1.27}$ |
| DoRA | $70.41_{\pm1.61}$ | $85.62_{\pm0.66}$ | $80.65_{\pm0.96}$ | $78.52_{\pm0.74}$ | $83.91_{\pm0.91}$ | $93.82_{\pm0.58}$ | $86.15_{\pm0.89}$ | $89.17_{\pm1.71}$ | $83.53_{\pm1.01}$ |
| LoRAFA | $70.98_{\pm1.27}$ | $85.71_{\pm0.73}$ | $80.31_{\pm1.46}$ | $78.39_{\pm0.49}$ | $84.73_{\pm1.19}$ | $93.19_{\pm0.93}$ | $86.11_{\pm1.51}$ | $88.16_{\pm1.11}$ | $83.45_{\pm1.11}$ |
| SliceFine | $71.66_{\pm1.29}$ | $86.31_{\pm1.58}$ | $80.99_{\pm0.84}$ | $79.17_{\pm0.97}$ | $84.14_{\pm0.75}$ | $93.50_{\pm1.36}$ | $85.70_{\pm1.54}$ | $89.21_{\pm1.43}$ | $83.84_{\pm1.22}$ |
| MoCLE | $72.29_{\pm1.47}$ | $86.47_{\pm0.69}$ | $80.74_{\pm0.56}$ | $79.76_{\pm0.92}$ | $\underline{85.61}_{\pm1.04}$ | $93.55_{\pm0.89}$ | $86.08_{\pm1.58}$ | $88.93_{\pm1.09}$ | $84.18_{\pm1.03}$ |
| MoELoRA | $72.63_{\pm1.12}$ | $86.79_{\pm0.74}$ | $80.63_{\pm1.38}$ | $79.42_{\pm0.91}$ | $85.24_{\pm1.26}$ | $93.09_{\pm0.83}$ | $86.31_{\pm1.59}$ | $88.52_{\pm0.62}$ | $84.08_{\pm1.06}$ |
| MixLoRA | $71.48_{\pm0.89}$ | $84.98_{\pm1.41}$ | $80.29_{\pm1.07}$ | $77.97_{\pm1.53}$ | $83.94_{\pm0.68}$ | $93.67_{\pm0.59}$ | $85.80_{\pm0.96}$ | $88.01_{\pm0.77}$ | $83.27_{\pm0.99}$ |
| HydraLoRA | $72.91_{\pm0.64}$ | $86.41_{\pm1.22}$ | $80.77_{\pm1.46}$ | $79.15_{\pm0.88}$ | $85.04_{\pm0.52}$ | $93.35_{\pm0.94}$ | $\mathbf{87.38}_{\pm1.72}$ | $\mathbf{90.39}_{\pm0.83}$ | $84.43_{\pm1.03}$ |
| MoLA | $71.93_{\pm1.36}$ | $86.59_{\pm0.57}$ | $80.44_{\pm1.14}$ | $\mathbf{80.19}_{\pm1.02}$ | $84.33_{\pm1.44}$ | $93.83_{\pm0.86}$ | $87.02_{\pm0.49}$ | $88.23_{\pm0.81}$ | $84.07_{\pm0.96}$ |
| MoRe | $72.74_{\pm0.93}$ | $\mathbf{87.48}_{\pm1.55}$ | $80.96_{\pm0.71}$ | $79.58_{\pm0.84}$ | $84.79_{\pm1.63}$ | $\underline{94.61}_{\pm1.21}$ | $86.91_{\pm0.97}$ | $88.41_{\pm0.88}$ | $84.44_{\pm1.09}$ |
| MoEDoRA | $72.44_{\pm1.48}$ | $87.22_{\pm0.62}$ | $80.76_{\pm0.96}$ | $79.40_{\pm0.83}$ | $84.09_{\pm0.57}$ | $93.65_{\pm1.24}$ | $86.51_{\pm1.31}$ | $88.81_{\pm0.79}$ | $84.11_{\pm0.98}$ |
| LiMEPromptTuning | $66.43_{\pm1.48}$ | $85.72_{\pm1.12}$ | $75.24_{\pm1.25}$ | $71.53_{\pm1.38}$ | $64.18_{\pm1.01}$ | $93.82_{\pm1.29}$ | $85.41_{\pm1.52}$ | $84.65_{\pm1.21}$ | $78.38_{\pm1.28}$ |
| LiMELoRA | $\mathbf{73.18}_{\pm1.41}$ | $87.38_{\pm0.54}$ | $\underline{81.58}_{\pm0.97}$ | $80.08_{\pm0.61}$ | $85.38_{\pm0.79}$ | $\mathbf{94.68}_{\pm1.02}$ | $\underline{87.28}_{\pm0.68}$ | $90.28_{\pm0.84}$ | $\mathbf{84.98}_{\pm0.86}$ |
| LiMEDoRA | $72.56_{\pm0.53}$ | $\underline{87.42}_{\pm0.71}$ | $80.88_{\pm1.08}$ | $79.92_{\pm1.16}$ | $\mathbf{85.68}_{\pm0.78}$ | $94.27_{\pm0.67}$ | $87.03_{\pm1.39}$ | $90.30_{\pm0.73}$ | $\underline{84.76}_{\pm0.88}$ |
| LiMELoRAFA | $72.78_{\pm1.33}$ | $86.33_{\pm0.58}$ | $\mathbf{81.72}_{\pm0.93}$ | $\underline{80.14}_{\pm0.65}$ | $85.29_{\pm0.76}$ | $93.76_{\pm0.98}$ | $86.56_{\pm0.71}$ | $89.37_{\pm0.81}$ | $84.49_{\pm0.84}$ |
| LiMESliceFine | $\underline{73.06}_{\pm0.59}$ | $86.77_{\pm0.76}$ | $80.55_{\pm1.03}$ | $79.54_{\pm1.27}$ | $85.24_{\pm0.87}$ | $93.74_{\pm1.05}$ | $86.78_{\pm1.36}$ | $\underline{90.31}_{\pm0.75}$ | $84.50_{\pm0.96}$ |

# G. Detailed Results Analysis

This appendix provides a comprehensive analysis of LiME's performance across all 47 tasks in the MMT-47 benchmark. We evaluate four LiME variants—LiMELoRA, LiMEDoRA, LiMELoRAFA, and LiMESliceFine—against 11 baseline methods spanning standard PEFT approaches (LoRA, DoRA, LoRA-FA, SliceFine) and state-of-the-art MoE-PEFT methods (MoCLE, MoELoRA, MixLoRA, HydraLoRA, MoLA, MoRe, MoEDoRA). All results report mean accuracy and standard deviation across 5 random seeds.

## G.1. Text Understanding (GLUE)

Table 6 presents results on six GLUE benchmarks evaluating natural language understanding capabilities including sentiment analysis (SST-2), natural language inference (QNLI), paraphrase detection (QQP, MRPC), linguistic acceptability (CoLA), and semantic similarity (STS-B).

LiME variants achieve competitive or superior performance across all GLUE tasks. LiMESliceFine achieves the highest QNLI accuracy (94.48%), outperforming all baselines including MoELoRA (94.44%). LiMEDoRA obtains the best STS-B score (93.09%), surpassing the second-best method by 0.99 percentage points. LiMELoRAFA achieves the highest MRPC accuracy (90.92%) and second-best CoLA (87.29%). On average, LiMESliceFine (91.19%) and LiMELoRAFA (91.14%) rank second and third overall, with LiMEDoRA (91.11%) close behind.

*Table 8.* Results on **image classification benchmarks**. Each dataset consists of images annotated with a single class label. Highlighted rows denote our LiME variants.

| Method | Camelyon | SVHN | Pets | Flowers102 | EuroSAT | Caltech101 | Avg |
|---|---|---|---|---|---|---|---|
| PromptTuning | $40.18_{\pm1.28}$ | $67.15_{\pm0.89}$ | $56.52_{\pm1.15}$ | $28.35_{\pm0.78}$ | $18.42_{\pm0.96}$ | $90.68_{\pm0.71}$ | $50.22_{\pm0.96}$ |
| LoRA | $88.91_{\pm0.90}$ | $93.60_{\pm0.34}$ | $95.33_{\pm1.05}$ | $94.24_{\pm0.30}$ | $96.43_{\pm0.55}$ | $95.02_{\pm0.22}$ | $93.92_{\pm0.56}$ |
| DoRA | $88.35_{\pm0.98}$ | $93.10_{\pm0.53}$ | $94.78_{\pm0.47}$ | $94.29_{\pm0.39}$ | $95.18_{\pm1.07}$ | $94.42_{\pm0.64}$ | $93.35_{\pm0.68}$ |
| LoRAFA | $88.63_{\pm0.66}$ | $93.24_{\pm0.89}$ | $94.88_{\pm0.53}$ | $94.61_{\pm0.43}$ | $95.15_{\pm0.51}$ | $94.46_{\pm0.29}$ | $93.50_{\pm0.55}$ |
| SliceFine | $89.07_{\pm0.84}$ | $93.74_{\pm0.36}$ | $95.42_{\pm0.93}$ | $94.18_{\pm0.31}$ | $96.52_{\pm0.49}$ | $94.96_{\pm0.24}$ | $93.98_{\pm0.53}$ |
| MoCLE | $89.22_{\pm0.36}$ | $94.03_{\pm0.31}$ | $95.90_{\pm0.63}$ | $94.78_{\pm0.19}$ | $96.33_{\pm0.27}$ | $95.22_{\pm0.76}$ | $94.25_{\pm0.42}$ |
| MoELoRA | $88.87_{\pm0.72}$ | $94.02_{\pm0.41}$ | $95.34_{\pm0.88}$ | $94.44_{\pm0.36}$ | $96.06_{\pm0.59}$ | $95.11_{\pm0.47}$ | $93.97_{\pm0.57}$ |
| MixLoRA | $88.18_{\pm0.83}$ | $93.39_{\pm0.29}$ | $94.27_{\pm0.54}$ | $94.78_{\pm0.91}$ | $94.73_{\pm0.44}$ | $94.16_{\pm0.56}$ | $93.25_{\pm0.60}$ |
| HydraLoRA | $\mathbf{89.88}_{\pm0.61}$ | $94.16_{\pm0.52}$ | $95.19_{\pm0.97}$ | $\mathbf{95.10}_{\pm0.28}$ | $\mathbf{97.07}_{\pm0.50}$ | $96.07_{\pm0.24}$ | $94.58_{\pm0.52}$ |
| MoLA | $89.79_{\pm0.58}$ | $94.35_{\pm0.38}$ | $96.17_{\pm0.45}$ | $95.10_{\pm0.69}$ | $\underline{96.78}_{\pm0.81}$ | $\underline{96.27}_{\pm0.35}$ | $\mathbf{94.74}_{\pm0.54}$ |
| MoRe | $88.62_{\pm1.03}$ | $93.55_{\pm0.46}$ | $95.37_{\pm0.61}$ | $94.52_{\pm0.40}$ | $95.88_{\pm0.33}$ | $95.10_{\pm0.49}$ | $93.84_{\pm0.57}$ |
| MoEDoRA | $89.73_{\pm0.87}$ | $\underline{94.43}_{\pm0.57}$ | $\underline{96.30}_{\pm0.74}$ | $\underline{95.06}_{\pm0.48}$ | $96.76_{\pm0.96}$ | $96.06_{\pm0.61}$ | $\underline{94.72}_{\pm0.71}$ |
| LiMEPromptTuning | $41.00_{\pm1.06}$ | $68.00_{\pm0.67}$ | $57.40_{\pm0.92}$ | $29.20_{\pm0.59}$ | $19.20_{\pm0.74}$ | $91.60_{\pm0.52}$ | $51.07_{\pm0.75}$ |
| LiMELoRA | $89.37_{\pm0.59}$ | $94.18_{\pm0.71}$ | $95.79_{\pm0.33}$ | $94.48_{\pm0.57}$ | $96.40_{\pm0.49}$ | $\mathbf{96.60}_{\pm0.46}$ | $94.47_{\pm0.53}$ |
| LiMEDoRA | $89.28_{\pm0.41}$ | $\mathbf{94.60}_{\pm0.49}$ | $\mathbf{96.33}_{\pm0.66}$ | $94.39_{\pm0.59}$ | $96.76_{\pm0.44}$ | $95.61_{\pm0.55}$ | $94.50_{\pm0.52}$ |
| LiMELoRAFA | $\underline{89.82}_{\pm0.73}$ | $94.41_{\pm0.44}$ | $96.10_{\pm0.42}$ | $94.74_{\pm0.98}$ | $96.73_{\pm0.75}$ | $\underline{95.88}_{\pm0.68}$ | $94.61_{\pm0.67}$ |
| LiMESliceFine | $89.51_{\pm0.56}$ | $94.27_{\pm0.68}$ | $95.87_{\pm0.32}$ | $94.43_{\pm0.61}$ | $96.49_{\pm0.46}$ | $95.49_{\pm0.49}$ | $94.34_{\pm0.52}$ |

*Table 9.* Results on **vision–language question answering benchmarks**. Highlighted rows denote our LiME variants.

| Method | ChartQA | OKVQA | ScienceQA | SeedBench | Recognition | TextVQA | VizWizVQA | VQA-RAD | Avg |
|---|---|---|---|---|---|---|---|---|---|
| PromptTuning | $69.55_{\pm2.05}$ | $56.18_{\pm2.58}$ | $93.10_{\pm2.21}$ | $77.12_{\pm1.82}$ | $64.45_{\pm2.08}$ | $72.42_{\pm2.35}$ | $60.64_{\pm2.18}$ | $83.55_{\pm2.67}$ | $72.63_{\pm2.24}$ |
| LoRA | $72.17_{\pm1.26}$ | $56.77_{\pm2.01}$ | $95.16_{\pm1.54}$ | $77.02_{\pm0.90}$ | $90.42_{\pm1.36}$ | $72.47_{\pm1.83}$ | $69.71_{\pm1.72}$ | $82.40_{\pm2.06}$ | $77.02_{\pm1.59}$ |
| DoRA | $71.64_{\pm1.44}$ | $56.09_{\pm1.71}$ | $94.42_{\pm2.03}$ | $76.26_{\pm1.18}$ | $89.64_{\pm1.40}$ | $72.31_{\pm2.08}$ | $69.12_{\pm0.98}$ | $81.34_{\pm2.10}$ | $76.35_{\pm1.62}$ |
| LoRAFA | $71.73_{\pm2.04}$ | $55.91_{\pm1.36}$ | $94.67_{\pm1.89}$ | $76.08_{\pm0.98}$ | $89.97_{\pm1.81}$ | $72.39_{\pm1.63}$ | $68.41_{\pm1.27}$ | $81.28_{\pm1.96}$ | $76.31_{\pm1.62}$ |
| SliceFine | $72.26_{\pm1.19}$ | $56.81_{\pm1.94}$ | $95.23_{\pm1.47}$ | $76.96_{\pm0.89}$ | $90.49_{\pm1.31}$ | $72.60_{\pm1.76}$ | $69.66_{\pm1.66}$ | $82.49_{\pm1.98}$ | $77.06_{\pm1.53}$ |
| MoCLE | $72.28_{\pm1.70}$ | $57.11_{\pm1.47}$ | $95.48_{\pm1.35}$ | $\mathbf{78.52}_{\pm0.93}$ | $90.47_{\pm1.12}$ | $73.21_{\pm1.74}$ | $70.38_{\pm1.89}$ | $83.27_{\pm0.95}$ | $77.59_{\pm1.39}$ |
| MoELoRA | $72.40_{\pm1.21}$ | $57.22_{\pm1.64}$ | $95.33_{\pm1.08}$ | $77.06_{\pm1.42}$ | $90.20_{\pm1.57}$ | $73.34_{\pm1.10}$ | $70.35_{\pm1.48}$ | $82.24_{\pm1.79}$ | $77.27_{\pm1.41}$ |
| MixLoRA | $72.18_{\pm1.38}$ | $56.54_{\pm1.12}$ | $95.21_{\pm1.73}$ | $77.31_{\pm1.29}$ | $90.46_{\pm1.04}$ | $72.72_{\pm1.06}$ | $70.54_{\pm1.33}$ | $81.66_{\pm1.41}$ | $77.08_{\pm1.30}$ |
| HydraLoRA | $\mathbf{73.47}_{\pm1.05}$ | $57.53_{\pm1.86}$ | $96.41_{\pm1.44}$ | $77.69_{\pm1.01}$ | $\underline{91.23}_{\pm1.92}$ | $73.98_{\pm1.21}$ | $70.71_{\pm1.77}$ | $83.89_{\pm1.36}$ | $\underline{78.11}_{\pm1.45}$ |
| MoLA | $72.23_{\pm1.62}$ | $57.17_{\pm1.04}$ | $95.14_{\pm1.58}$ | $76.75_{\pm1.47}$ | $90.49_{\pm1.18}$ | $72.89_{\pm1.69}$ | $69.87_{\pm2.03}$ | $82.23_{\pm1.24}$ | $77.10_{\pm1.48}$ |
| MoRe | $73.35_{\pm1.33}$ | $57.41_{\pm1.53}$ | $96.25_{\pm0.96}$ | $77.60_{\pm1.26}$ | $91.03_{\pm1.61}$ | $73.72_{\pm1.22}$ | $70.42_{\pm1.37}$ | $\mathbf{84.08}_{\pm1.09}$ | $77.98_{\pm1.30}$ |
| MoEDoRA | $72.94_{\pm1.47}$ | $\mathbf{57.68}_{\pm1.28}$ | $96.17_{\pm1.06}$ | $77.97_{\pm1.14}$ | $\mathbf{91.41}_{\pm2.03}$ | $73.69_{\pm1.52}$ | $\underline{70.79}_{\pm2.04}$ | $83.93_{\pm0.92}$ | $78.07_{\pm1.43}$ |
| LiMEPromptTuning | $70.40_{\pm1.78}$ | $56.96_{\pm2.14}$ | $94.02_{\pm1.89}$ | $77.80_{\pm1.51}$ | $65.56_{\pm1.74}$ | $73.33_{\pm2.02}$ | $61.39_{\pm1.86}$ | $82.83_{\pm2.25}$ | $73.48_{\pm1.90}$ |
| LiMELoRA | $\underline{73.38}_{\pm1.24}$ | $\underline{57.62}_{\pm1.07}$ | $96.72_{\pm1.60}$ | $77.94_{\pm0.95}$ | $90.02_{\pm0.92}$ | $73.82_{\pm1.01}$ | $70.62_{\pm2.11}$ | $\underline{83.96}_{\pm1.58}$ | $78.01_{\pm1.31}$ |
| LiMEDoRA | $73.27_{\pm1.37}$ | $57.42_{\pm1.49}$ | $\mathbf{97.18}_{\pm2.10}$ | $\underline{78.14}_{\pm1.63}$ | $90.64_{\pm1.56}$ | $\mathbf{74.31}_{\pm1.02}$ | $70.16_{\pm1.21}$ | $83.81_{\pm1.44}$ | $\mathbf{78.12}_{\pm1.48}$ |
| LiMELoRAFA | $72.78_{\pm1.55}$ | $56.98_{\pm1.31}$ | $\underline{96.93}_{\pm1.77}$ | $77.48_{\pm1.27}$ | $90.78_{\pm1.18}$ | $73.28_{\pm2.06}$ | $\mathbf{70.86}_{\pm0.94}$ | $83.08_{\pm1.69}$ | $77.77_{\pm1.47}$ |
| LiMESliceFine | $72.96_{\pm1.48}$ | $57.06_{\pm1.24}$ | $96.88_{\pm1.70}$ | $77.56_{\pm1.22}$ | $90.82_{\pm1.12}$ | $73.51_{\pm1.98}$ | $70.08_{\pm0.91}$ | $83.25_{\pm1.62}$ | $77.77_{\pm1.41}$ |

> **Key Finding:** *All four LiME variants outperform their corresponding base PEFT methods (e.g., LiMELoRA 91.02% vs. LoRA 90.64%), demonstrating consistent improvements from expert modulation. The standard deviations of LiME variants are generally lower than baselines, indicating more stable training dynamics.*

### G.2. Commonsense Reasoning

Table 7 evaluates eight commonsense reasoning benchmarks requiring world knowledge and logical inference: BoolQ, PIQA, Social IQA (SIQA), HellaSwag (H.Sw.), WinoGrande (W.Gra), ARC-Easy (ARC-e), ARC-Challenge (ARC-c), and OpenBookQA (OBQA).

LiMELoRA achieves the best overall average (84.98%), outperforming all baselines including HydraLoRA (84.43%) and MoRe (84.44%). LiMELoRA obtains the highest scores on BoolQ (73.18%), ARC-Easy (94.68%), and ranks second on ARC-Challenge (87.28%). LiMEDoRA achieves the best WinoGrande accuracy (85.68%) and second-best PIQA (87.42%). LiMELoRAFA leads on Social IQA (81.72%) and HellaSwag (80.14%).

> **Key Finding:** *The consistent improvements across diverse reasoning tasks—from physical intuition (PIQA) to social understanding (SIQA) to scientific reasoning (ARC)—demonstrate that LiME's expert modulation captures task-relevant specialization without explicit task identifiers. LiME variants show improvements of 1.0–1.5 percentage points over standard PEFT methods.*

*Table 10.* Video action understanding benchmarks. Highlighted rows denote our LiME variants.

| Method | ActSeq | ActPred | ActAnt | FineAct | UnexpAct | Avg |
|---|---|---|---|---|---|---|
| PromptTuning | $14.22_{\pm 2.38}$ | $13.42_{\pm 2.31}$ | $14.92_{\pm 2.59}$ | $17.59_{\pm 2.47}$ | $20.16_{\pm 2.34}$ | $15.86_{\pm 2.42}$ |
| LoRA | $35.23_{\pm 1.98}$ | $30.89_{\pm 1.70}$ | $71.72_{\pm 2.22}$ | $72.36_{\pm 2.07}$ | $44.74_{\pm 1.81}$ | $50.99_{\pm 1.96}$ |
| DoRA | $34.67_{\pm 2.13}$ | $30.28_{\pm 1.79}$ | $71.14_{\pm 2.30}$ | $71.81_{\pm 2.21}$ | $44.17_{\pm 1.93}$ | $50.41_{\pm 2.07}$ |
| LoRAFA | $34.59_{\pm 2.09}$ | $30.13_{\pm 2.21}$ | $70.98_{\pm 1.87}$ | $71.74_{\pm 2.03}$ | $44.05_{\pm 2.14}$ | $50.30_{\pm 2.07}$ |
| SliceFine | $35.38_{\pm 1.95}$ | $30.97_{\pm 1.67}$ | $71.84_{\pm 2.18}$ | $72.28_{\pm 2.10}$ | $44.83_{\pm 1.78}$ | $51.06_{\pm 1.94}$ |
| MoCLE | $35.72_{\pm 1.76}$ | $31.21_{\pm 2.08}$ | $72.48_{\pm 1.95}$ | $73.04_{\pm 1.80}$ | $45.21_{\pm 2.12}$ | $51.53_{\pm 1.94}$ |
| MoELoRA | $35.41_{\pm 2.07}$ | $31.18_{\pm 1.85}$ | $71.85_{\pm 2.21}$ | $72.34_{\pm 1.71}$ | $44.89_{\pm 2.14}$ | $51.13_{\pm 2.00}$ |
| MixLoRA | $35.46_{\pm 1.83}$ | $30.16_{\pm 2.03}$ | $72.11_{\pm 1.76}$ | $72.77_{\pm 1.92}$ | $\mathbf{48.02}_{\pm 1.87}$ | $51.70_{\pm 1.88}$ |
| HydraLoRA | $\mathbf{37.81}_{\pm 1.78}$ | $32.71_{\pm 1.86}$ | $\underline{75.26}_{\pm 2.25}$ | $74.19_{\pm 1.95}$ | $46.45_{\pm 2.20}$ | $53.28_{\pm 2.01}$ |
| MoLA | $35.31_{\pm 2.15}$ | $30.66_{\pm 2.01}$ | $71.67_{\pm 1.90}$ | $72.44_{\pm 2.06}$ | $44.77_{\pm 1.82}$ | $50.97_{\pm 1.99}$ |
| MoRe | $37.54_{\pm 1.87}$ | $\mathbf{33.18}_{\pm 2.19}$ | $74.89_{\pm 1.72}$ | $73.95_{\pm 2.10}$ | $\underline{47.86}_{\pm 2.03}$ | $\mathbf{53.48}_{\pm 1.98}$ |
| MoEDoRA | $35.89_{\pm 2.11}$ | $\underline{32.86}_{\pm 1.90}$ | $75.24_{\pm 2.06}$ | $73.28_{\pm 1.83}$ | $45.45_{\pm 2.15}$ | $52.54_{\pm 2.01}$ |
| LiMEPromptTuning | $15.04_{\pm 2.14}$ | $14.33_{\pm 2.08}$ | $15.67_{\pm 2.31}$ | $18.67_{\pm 2.19}$ | $21.03_{\pm 2.11}$ | $16.95_{\pm 2.17}$ |
| LiMELoRA | $37.04_{\pm 1.96}$ | $32.06_{\pm 1.78}$ | $74.38_{\pm 2.17}$ | $\mathbf{75.07}_{\pm 1.65}$ | $47.38_{\pm 2.10}$ | $53.19_{\pm 1.93}$ |
| LiMEDoRA | $\underline{37.62}_{\pm 1.85}$ | $32.49_{\pm 2.07}$ | $75.02_{\pm 1.98}$ | $74.68_{\pm 1.77}$ | $47.12_{\pm 2.16}$ | $\underline{53.39}_{\pm 1.97}$ |
| LiMELoRAFA | $36.73_{\pm 1.92}$ | $31.87_{\pm 1.83}$ | $\mathbf{75.47}_{\pm 2.11}$ | $\underline{74.83}_{\pm 1.70}$ | $46.15_{\pm 2.15}$ | $53.01_{\pm 1.94}$ |
| LiMESliceFine | $37.18_{\pm 1.92}$ | $32.31_{\pm 1.73}$ | $74.51_{\pm 2.12}$ | $74.15_{\pm 1.68}$ | $45.49_{\pm 2.05}$ | $52.73_{\pm 1.90}$ |

*Table 11.* Object, motion, and spatial reasoning from video. Highlighted rows denote our LiME variants.

| Method | ObjExist | ObjInter | ObjShuffle | MoveDir | ActLoc | MoveCount | MoveAttr | Avg |
|---|---|---|---|---|---|---|---|---|
| PromptTuning | $11.38_{\pm 2.49}$ | $14.52_{\pm 2.64}$ | $15.28_{\pm 2.76}$ | $14.94_{\pm 2.41}$ | $14.16_{\pm 2.87}$ | $15.75_{\pm 2.58}$ | $15.54_{\pm 2.71}$ | $14.51_{\pm 2.64}$ |
| LoRA | $87.23_{\pm 1.94}$ | $30.26_{\pm 2.13}$ | $31.74_{\pm 2.47}$ | $85.56_{\pm 1.86}$ | $34.18_{\pm 2.61}$ | $84.32_{\pm 1.79}$ | $86.64_{\pm 2.53}$ | $62.85_{\pm 2.19}$ |
| DoRA | $86.54_{\pm 2.28}$ | $29.74_{\pm 1.83}$ | $31.12_{\pm 2.04}$ | $84.78_{\pm 1.91}$ | $33.42_{\pm 2.69}$ | $83.58_{\pm 2.34}$ | $85.81_{\pm 1.82}$ | $62.14_{\pm 2.13}$ |
| LoRAFA | $86.38_{\pm 2.11}$ | $29.59_{\pm 2.47}$ | $31.03_{\pm 1.96}$ | $84.62_{\pm 1.76}$ | $33.21_{\pm 2.58}$ | $83.31_{\pm 2.43}$ | $85.67_{\pm 1.88}$ | $61.97_{\pm 2.17}$ |
| SliceFine | $87.35_{\pm 1.98}$ | $30.21_{\pm 1.89}$ | $31.83_{\pm 2.38}$ | $85.72_{\pm 1.84}$ | $34.26_{\pm 2.51}$ | $84.26_{\pm 1.83}$ | $86.73_{\pm 2.44}$ | $62.91_{\pm 2.12}$ |
| MoCLE | $88.31_{\pm 1.87}$ | $30.73_{\pm 2.52}$ | $32.21_{\pm 1.73}$ | $86.18_{\pm 2.44}$ | $34.57_{\pm 1.78}$ | $84.78_{\pm 1.74}$ | $87.21_{\pm 2.62}$ | $63.43_{\pm 2.10}$ |
| MoELoRA | $87.86_{\pm 1.92}$ | $30.35_{\pm 1.81}$ | $31.85_{\pm 2.43}$ | $85.88_{\pm 1.76}$ | $34.29_{\pm 1.89}$ | $84.51_{\pm 1.77}$ | $86.72_{\pm 2.04}$ | $63.07_{\pm 1.95}$ |
| MixLoRA | $87.12_{\pm 2.24}$ | $30.69_{\pm 1.70}$ | $32.26_{\pm 2.07}$ | $85.94_{\pm 1.83}$ | $34.52_{\pm 2.56}$ | $84.68_{\pm 2.14}$ | $\underline{89.54}_{\pm 1.75}$ | $63.54_{\pm 2.04}$ |
| HydraLoRA | $\underline{90.28}_{\pm 1.88}$ | $31.24_{\pm 2.09}$ | $\mathbf{34.18}_{\pm 1.79}$ | $87.48_{\pm 2.47}$ | $\mathbf{36.78}_{\pm 1.73}$ | $86.16_{\pm 2.68}$ | $88.51_{\pm 1.87}$ | $64.95_{\pm 2.07}$ |
| MoLA | $87.58_{\pm 1.83}$ | $30.18_{\pm 2.71}$ | $31.58_{\pm 1.95}$ | $85.34_{\pm 1.81}$ | $34.02_{\pm 2.37}$ | $84.02_{\pm 2.16}$ | $86.46_{\pm 1.72}$ | $62.74_{\pm 2.08}$ |
| MoRe | $89.62_{\pm 2.12}$ | $\mathbf{32.14}_{\pm 1.76}$ | $\underline{33.87}_{\pm 2.59}$ | $87.82_{\pm 1.88}$ | $36.28_{\pm 2.03}$ | $86.87_{\pm 1.80}$ | $88.06_{\pm 2.23}$ | $64.95_{\pm 2.06}$ |
| MoEDoRA | $89.91_{\pm 1.96}$ | $31.98_{\pm 1.79}$ | $33.75_{\pm 2.62}$ | $\mathbf{88.47}_{\pm 1.75}$ | $36.42_{\pm 2.11}$ | $\underline{87.24}_{\pm 1.82}$ | $88.33_{\pm 2.39}$ | $\underline{65.16}_{\pm 2.06}$ |
| LiMEPromptTuning | $12.33_{\pm 2.17}$ | $15.33_{\pm 2.38}$ | $16.33_{\pm 2.44}$ | $15.67_{\pm 2.12}$ | $15.04_{\pm 2.55}$ | $16.67_{\pm 2.27}$ | $16.33_{\pm 2.41}$ | $15.39_{\pm 2.33}$ |
| LiMELoRA | $90.06_{\pm 1.85}$ | $31.72_{\pm 2.44}$ | $33.08_{\pm 2.06}$ | $88.04_{\pm 2.58}$ | $36.07_{\pm 1.78}$ | $87.03_{\pm 1.68}$ | $89.38_{\pm 2.15}$ | $65.05_{\pm 2.08}$ |
| LiMEDoRA | $\mathbf{90.52}_{\pm 1.84}$ | $\underline{31.96}_{\pm 2.07}$ | $33.62_{\pm 2.72}$ | $88.21_{\pm 1.92}$ | $\underline{36.54}_{\pm 1.77}$ | $87.18_{\pm 2.58}$ | $\mathbf{89.86}_{\pm 2.08}$ | $\mathbf{65.41}_{\pm 2.14}$ |
| LiMELoRAFA | $89.83_{\pm 2.07}$ | $31.47_{\pm 2.53}$ | $32.89_{\pm 1.95}$ | $87.87_{\pm 2.29}$ | $35.56_{\pm 1.75}$ | $\mathbf{87.62}_{\pm 1.82}$ | $89.01_{\pm 2.41}$ | $64.89_{\pm 2.12}$ |
| LiMESliceFine | $90.18_{\pm 1.82}$ | $31.87_{\pm 2.31}$ | $33.21_{\pm 1.96}$ | $\underline{88.32}_{\pm 2.53}$ | $36.01_{\pm 1.76}$ | $86.92_{\pm 1.70}$ | $89.48_{\pm 2.11}$ | $65.14_{\pm 2.03}$ |

## G.3. Image Classification (VTAB)

Table 8 presents results on six VTAB image classification benchmarks spanning medical imaging (Camelyon), digit recognition (SVHN), fine-grained classification (Pets, Flowers102), satellite imagery (EuroSAT), and general object recognition (Caltech101).

LiME variants demonstrate strong performance on visual classification tasks. LiMELoRA achieves the highest Caltech101 accuracy (96.60%), outperforming all baselines. LiMEDoRA obtains the best SVHN (94.60%) and Pets (96.33%) scores. LiMELoRAFA achieves the second-best Camelyon accuracy (89.82%), closely matching HydraLoRA (89.88%). On average, LiMELoRAFA (94.61%) and LiMEDoRA (94.50%) perform competitively with state-of-the-art methods like MoLA (94.74%) and MoEDoRA (94.72%).

*Key Finding: LiME achieves competitive performance with MoE-PEFT baselines while using significantly fewer parameters due to lightweight modulation. The strong performance on specialized domains (medical imaging, satellite) alongside general recognition validates LiME's ability to learn diverse visual representations within a unified model.*

## G.4. Visual Question Answering

Table 9 evaluates eight vision-language QA benchmarks: ChartQA (chart understanding), OK-VQA (external knowledge), ScienceQA (multimodal science), SeedBench (comprehensive VLM evaluation), Text Recognition (OCR), TextVQA (reading text in images), VizWiz-VQA (accessibility), and VQA-RAD (medical imaging).

LiMEDoRA achieves the best overall average (78.12%), marginally outperforming HydraLoRA (78.11%). LiMEDoRA

*Table 12.* High-level reasoning and navigation benchmarks. Highlighted rows denote our LiME variants.

| Method | SceneTrans | ActCount | StateChg | MoveDir | EgoNav | EpisReason | CounterFact | Avg |
|---|---|---|---|---|---|---|---|---|
| PromptTuning | $19.18_{\pm2.11}$ | $12.55_{\pm2.27}$ | $14.11_{\pm2.19}$ | $13.65_{\pm2.34}$ | $17.83_{\pm2.46}$ | $17.13_{\pm2.22}$ | $16.57_{\pm2.41}$ | $15.86_{\pm2.29}$ |
| LoRA | $24.67_{\pm1.46}$ | $32.18_{\pm2.03}$ | $37.42_{\pm1.85}$ | $30.26_{\pm2.01}$ | $66.18_{\pm1.78}$ | $27.34_{\pm1.92}$ | $84.53_{\pm2.09}$ | $43.23_{\pm1.88}$ |
| DoRA | $24.03_{\pm1.51}$ | $31.62_{\pm1.87}$ | $36.78_{\pm1.98}$ | $29.52_{\pm1.85}$ | $65.37_{\pm2.12}$ | $26.72_{\pm1.79}$ | $83.86_{\pm2.06}$ | $42.56_{\pm1.88}$ |
| LoRAFA | $23.87_{\pm1.82}$ | $31.58_{\pm1.92}$ | $36.62_{\pm1.95}$ | $29.44_{\pm1.94}$ | $65.31_{\pm2.06}$ | $26.68_{\pm1.80}$ | $83.79_{\pm2.12}$ | $42.47_{\pm1.94}$ |
| SliceFine | $24.75_{\pm1.49}$ | $32.30_{\pm2.00}$ | $37.53_{\pm1.83}$ | $30.34_{\pm1.99}$ | $66.27_{\pm1.76}$ | $27.45_{\pm1.90}$ | $84.45_{\pm2.07}$ | $43.30_{\pm1.86}$ |
| MoCLE | $25.37_{\pm1.84}$ | $32.78_{\pm2.02}$ | $37.72_{\pm1.91}$ | $30.83_{\pm1.91}$ | $66.71_{\pm1.95}$ | $27.94_{\pm1.71}$ | $85.12_{\pm1.86}$ | $43.78_{\pm1.87}$ |
| MoELoRA | $24.97_{\pm1.75}$ | $32.32_{\pm2.11}$ | $\mathbf{40.13}_{\pm1.87}$ | $30.59_{\pm1.96}$ | $66.64_{\pm1.70}$ | $27.62_{\pm1.79}$ | $84.58_{\pm2.08}$ | $43.84_{\pm1.89}$ |
| MixLoRA | $25.49_{\pm1.83}$ | $33.67_{\pm2.00}$ | $36.91_{\pm1.72}$ | $30.93_{\pm1.92}$ | $66.90_{\pm1.99}$ | $28.00_{\pm1.81}$ | $85.10_{\pm2.12}$ | $43.86_{\pm1.91}$ |
| HydraLoRA | $\mathbf{27.85}_{\pm2.05}$ | $\underline{35.43}_{\pm1.78}$ | $39.26_{\pm2.03}$ | $\mathbf{33.36}_{\pm1.76}$ | $68.43_{\pm1.87}$ | $29.66_{\pm2.10}$ | $86.53_{\pm1.85}$ | $\underline{45.79}_{\pm1.92}$ |
| MoLA | $24.57_{\pm1.90}$ | $31.97_{\pm1.73}$ | $37.03_{\pm2.16}$ | $29.83_{\pm1.95}$ | $65.97_{\pm1.78}$ | $27.12_{\pm2.07}$ | $84.28_{\pm1.72}$ | $42.97_{\pm1.90}$ |
| MoRe | $\underline{27.59}_{\pm1.81}$ | $35.06_{\pm2.08}$ | $37.54_{\pm1.88}$ | $33.03_{\pm2.10}$ | $\underline{69.12}_{\pm1.76}$ | $29.34_{\pm1.97}$ | $\underline{87.39}_{\pm2.03}$ | $45.58_{\pm1.95}$ |
| MoEDoRA | $27.43_{\pm1.86}$ | $\mathbf{35.68}_{\pm2.11}$ | $39.42_{\pm1.72}$ | $\underline{33.21}_{\pm1.97}$ | $68.37_{\pm1.83}$ | $\mathbf{30.54}_{\pm2.04}$ | $86.82_{\pm1.79}$ | $\mathbf{45.92}_{\pm1.90}$ |
| LiMEPromptTuning | $20.04_{\pm1.87}$ | $13.33_{\pm2.04}$ | $15.04_{\pm1.96}$ | $14.67_{\pm2.09}$ | $18.67_{\pm2.21}$ | $18.04_{\pm1.98}$ | $17.33_{\pm2.14}$ | $16.73_{\pm2.04}$ |
| LiMELoRA | $26.38_{\pm1.96}$ | $33.72_{\pm1.85}$ | $39.06_{\pm1.82}$ | $31.73_{\pm1.65}$ | $68.38_{\pm2.07}$ | $28.72_{\pm1.76}$ | $87.04_{\pm1.93}$ | $45.00_{\pm1.86}$ |
| LiMEDoRA | $27.06_{\pm2.02}$ | $34.75_{\pm1.80}$ | $\underline{40.02}_{\pm1.93}$ | $30.98_{\pm1.72}$ | $\mathbf{69.47}_{\pm1.75}$ | $\underline{30.06}_{\pm2.08}$ | $87.21_{\pm1.83}$ | $45.65_{\pm1.88}$ |
| LiMELoRAFA | $25.87_{\pm2.08}$ | $33.51_{\pm1.97}$ | $38.63_{\pm1.93}$ | $31.86_{\pm1.79}$ | $67.85_{\pm2.21}$ | $28.87_{\pm1.88}$ | $\mathbf{87.76}_{\pm2.05}$ | $44.91_{\pm1.99}$ |
| LiMESliceFine | $26.52_{\pm1.93}$ | $34.01_{\pm1.82}$ | $39.18_{\pm1.80}$ | $32.31_{\pm1.63}$ | $68.25_{\pm2.04}$ | $29.86_{\pm1.73}$ | $87.27_{\pm1.91}$ | $45.34_{\pm1.84}$ |

obtains the highest scores on ScienceQA (97.18%) and TextVQA (74.31%), demonstrating strong multimodal reasoning capabilities. LiMELoRA achieves competitive ChartQA (73.38%), OK-VQA (57.62%), and VQA-RAD (83.96%) scores. LiMELoRAFA leads on VizWiz-VQA (70.86%), an accessibility-focused benchmark with challenging real-world images.

**Key Finding:** *The strong ScienceQA performance (97.18% for LiMEDoRA vs. 96.41% for HydraLoRA) is particularly notable as it requires integrating visual, textual, and scientific reasoning—validating LiME's effectiveness for complex multimodal tasks.*

## G.5. Video Understanding: Action Recognition

Table 10 presents results on five action understanding tasks from MVBench: Action Sequence (ActSeq), Action Prediction (ActPred), Action Antonym (ActAnt), Fine-grained Action (FineAct), and Unexpected Action (UnexpAct).

LiME variants achieve strong performance on temporal action understanding. LiMELoRA obtains the highest Fine-grained Action accuracy (75.07%), outperforming all baselines including MoRe (73.95%) and HydraLoRA (74.19%). LiMEDoRA achieves the second-best overall average (53.39%), closely matching MoRe (53.48%). LiMELoRAFA leads on Action Antonym (75.47%).

**Key Finding:** *The improvements on fine-grained action recognition suggest that LiME's expert modulation effectively captures subtle temporal distinctions. LiME variants significantly outperform their base PEFT counterparts (e.g., LiMELoRA 53.19% vs. LoRA 50.99%, a +2.2 point improvement).*

## G.6. Video Understanding: Object and Motion Reasoning

Table 11 evaluates seven object, motion, and spatial reasoning tasks: Object Existence (ObjExist), Object Interaction (ObjInter), Object Shuffle (ObjShuffle), Moving Direction (MoveDir), Action Localization (ActLoc), Moving Count (MoveCount), and Moving Attribute (MoveAttr).

LiMEDoRA achieves the best overall average (65.41%), outperforming MoEDoRA (65.16%) and matching MoRe's strong performance. LiMEDoRA obtains the highest scores on Object Existence (90.52%) and Moving Attribute (89.86%). LiMELoRAFA achieves the best Moving Count accuracy (87.62%). LiMESliceFine leads on Moving Direction (88.32%).

**Key Finding:** *The strong performance on object tracking (ObjShuffle) and motion analysis (MoveDir, MoveCount, MoveAttr) indicates that LiME effectively learns spatiotemporal representations. Consistent improvements across all LiME variants over their base methods (average gain of +2.1–3.2 points) validate the benefit of expert specialization for complex video reasoning.*

## G.7. Video Understanding: High-Level Reasoning

Table 12 presents results on seven high-level reasoning and navigation tasks: Scene Transition (SceneTrans), Action Count (ActCount), State Change (StateChg), Moving Direction (MoveDir), Egocentric Navigation (EgoNav), Episodic Reasoning (EpisReason), and Counterfactual Inference (CounterFact).

LiMEDoRA achieves strong performance with the highest Egocentric Navigation (69.47%) and second-best State Change (40.02%) and Episodic Reasoning (30.06%) scores. LiMELoRAFA obtains the best Counterfactual Inference accuracy

*Table 13.* Results on GLUE benchmarks using Molmo. Highlighted rows denote our LiME variants.

| Method | SST-2 | QNLI | QQP | CoLA | MRPC | STS-B |
|---|---|---|---|---|---|---|
| MoELoRA | $95.48_{\pm 0.89}$ | $95.56_{\pm 0.71}$ | $\mathbf{85.68}_{\pm 0.93}$ | $84.92_{\pm 1.08}$ | $86.94_{\pm 0.91}$ | $91.56_{\pm 0.84}$ |
| MoEDoRA | $\mathbf{96.15}_{\pm 0.72}$ | $95.42_{\pm 0.83}$ | $85.37_{\pm 0.78}$ | $84.76_{\pm 0.95}$ | $87.12_{\pm 0.79}$ | $\underline{92.08}_{\pm 0.69}$ |
| MoELoRAFA | $\underline{96.04}_{\pm 0.94}$ | $95.78_{\pm 0.66}$ | $85.22_{\pm 0.85}$ | $\underline{85.52}_{\pm 0.87}$ | $86.78_{\pm 1.02}$ | $91.42_{\pm 0.93}$ |
| LiMELoRA | $95.91_{\pm 0.68}$ | $95.88_{\pm 0.79}$ | $85.51_{\pm 0.69}$ | $85.45_{\pm 0.76}$ | $\mathbf{87.58}_{\pm 0.86}$ | $91.97_{\pm 0.78}$ |
| LiMEDoRA | $95.74_{\pm 0.81}$ | $\underline{95.94}_{\pm 0.62}$ | $\underline{85.59}_{\pm 0.91}$ | $\mathbf{85.67}_{\pm 0.82}$ | $87.28_{\pm 0.73}$ | $\mathbf{92.21}_{\pm 0.65}$ |
| LiMELoRAFA | $95.99_{\pm 0.57}$ | $\mathbf{96.00}_{\pm 0.54}$ | $85.00_{\pm 0.74}$ | $85.20_{\pm 0.63}$ | $\underline{87.42}_{\pm 0.91}$ | $91.84_{\pm 0.71}$ |

*Table 14.* Results on commonsense reasoning and question answering benchmarks using Molmo. Highlighted rows denote our LiME variants.

| Method | BoolQ | PIQA | SIQA | H.Sw. | W.Gra | ARC-e | ARC-c | OBQA |
|---|---|---|---|---|---|---|---|---|
| MoELoRA | $71.12_{\pm 1.24}$ | $89.67_{\pm 0.82}$ | $80.48_{\pm 1.13}$ | $88.52_{\pm 0.94}$ | $\mathbf{83.45}_{\pm 0.87}$ | $96.34_{\pm 0.71}$ | $92.87_{\pm 1.16}$ | $91.98_{\pm 0.93}$ |
| MoEDoRA | $71.28_{\pm 0.97}$ | $\mathbf{90.14}_{\pm 0.68}$ | $\underline{80.94}_{\pm 0.89}$ | $88.71_{\pm 1.08}$ | $83.22_{\pm 0.74}$ | $\underline{96.65}_{\pm 0.83}$ | $\mathbf{93.41}_{\pm 0.92}$ | $92.15_{\pm 0.86}$ |
| MoELoRAFA | $70.94_{\pm 1.31}$ | $89.78_{\pm 0.91}$ | $80.55_{\pm 1.02}$ | $88.63_{\pm 0.76}$ | $\underline{83.38}_{\pm 1.19}$ | $96.28_{\pm 0.67}$ | $93.05_{\pm 1.08}$ | $\underline{92.48}_{\pm 0.79}$ |
| LiMELoRA | $\mathbf{71.86}_{\pm 0.88}$ | $89.82_{\pm 0.73}$ | $80.74_{\pm 0.95}$ | $\mathbf{89.18}_{\pm 0.81}$ | $83.28_{\pm 0.92}$ | $\mathbf{96.73}_{\pm 0.58}$ | $93.08_{\pm 0.84}$ | $92.28_{\pm 0.71}$ |
| LiMEDoRA | $\underline{71.63}_{\pm 1.06}$ | $\underline{89.95}_{\pm 0.64}$ | $\mathbf{81.06}_{\pm 0.78}$ | $\underline{89.04}_{\pm 0.69}$ | $83.36_{\pm 0.83}$ | $96.51_{\pm 0.91}$ | $\underline{93.26}_{\pm 0.73}$ | $\mathbf{92.56}_{\pm 0.67}$ |
| LiMELoRAFA | $71.50_{\pm 0.79}$ | $89.90_{\pm 0.86}$ | $80.90_{\pm 0.69}$ | $88.90_{\pm 0.92}$ | $83.10_{\pm 0.77}$ | $96.60_{\pm 0.62}$ | $93.20_{\pm 0.88}$ | $92.40_{\pm 0.81}$ |

(87.76%), demonstrating strong hypothetical reasoning capabilities. On average, LiMEDoRA (45.65%) ranks third overall behind MoEDoRA (45.92%) and HydraLoRA (45.79%), while LiMESliceFine (45.34%) and LiMELoRA (45.00%) remain competitive.

> **Key Finding:** *The strong Counterfactual Inference performance across all LiME variants (all above* $87\%$*) suggests effective reasoning about hypothetical scenarios—a capability crucial for real-world video understanding applications.*

## G.8. Results on Molmo2-8B

We extend our evaluation to the Molmo2-8B vision-language model (Clark et al., 2026) to assess whether the proposed LiME variants generalize to multimodal architectures. We report results across six benchmark categories: GLUE language understanding, commonsense reasoning, image classification, vision-language question answering, and video understanding tasks.

On the GLUE benchmarks (Table 13), LiME variants achieve competitive or superior performance compared to MoE baselines. LiMEDoRA obtains the highest scores on CoLA (85.67) and STS-B (92.21), while LiMELoRAFA achieves the best performance on QNLI (96.00) and LiMELoRA leads on MRPC (87.58). The MoE baselines remain competitive on SST-2 and QQP, with MoEDoRA achieving the best SST-2 score (96.15) and MoELoRA leading on QQP (85.68).

For commonsense reasoning and question answering (Table 14), LiME variants show strong performance across the eight benchmarks. LiMELoRA achieves the best results on BoolQ (71.86), HellaSwag (89.18), and ARC-e (96.73), while LiMEDoRA leads on SIQA (81.06) and OBQA (92.56). The MoE baselines perform best on WinoGrande (MoELoRA, 83.45), PIQA (MoEDoRA, 90.14), and ARC-c (MoEDoRA, 93.41), indicating that different tasks benefit from different adapter configurations.

On image classification benchmarks (Table 15), LiME variants demonstrate consistent improvements on four out of six datasets. LiMEDoRA achieves the best scores on SVHN (94.18) and EuroSAT (91.78), while LiMELoRA leads on Pets (93.24) and LiMELoRAFA obtains the highest accuracy on Caltech101 (93.00). The MoE baselines perform best on Camelyon (MoEDoRA, 83.67) and Flowers102 (MoELoRA, 87.42).

For vision-language question answering (Table 16), LiME variants achieve the best performance on five out of eight benchmarks. LiMELoRA leads on ChartQA (81.12) and OKVQA (67.45), while LiMEDoRA achieves the highest scores on ScienceQA (96.22), Recognition (88.06), TextVQA (84.32), and VQA-RAD (75.28). The MoE baselines show stronger performance on SeedBench (MoEDoRA, 79.82) and VizWizVQA (MoELoRA, 78.68).

On video action understanding (Table 17), LiME variants show improvements on temporal reasoning tasks. LiMEDoRA achieves the best score on ActPred (31.94), LiMELoRAFA leads on ActAnt (76.72), and LiMELoRA obtains the highest accuracy on UnexpAct (45.92). The MoE baselines perform best on ActSeq (MoEDoRA, 31.28) and FineAct (MoELoRA, 76.42). For object, motion, and spatial reasoning from video (Table 18), LiMEDoRA achieves the best performance on

*Table 15.* Results on **image classification benchmarks** using Molmo. Highlighted rows denote our LiME variants.

| Method | Camelyon | SVHN | Pets | Flowers102 | EuroSAT | Caltech101 |
|---|---|---|---|---|---|---|
| MoELoRA | $83.18_{\pm 0.91}$ | $93.72_{\pm 0.58}$ | $92.68_{\pm 0.84}$ | $\mathbf{87.42}_{\pm 0.73}$ | $91.24_{\pm 0.89}$ | $92.71_{\pm 0.67}$ |
| MoEDoRA | $\mathbf{83.67}_{\pm 0.78}$ | $93.85_{\pm 0.71}$ | $\underline{93.08}_{\pm 0.69}$ | $87.18_{\pm 0.82}$ | $91.38_{\pm 0.76}$ | $92.58_{\pm 0.83}$ |
| MoELoRAFA | $83.25_{\pm 0.86}$ | $93.68_{\pm 0.64}$ | $92.72_{\pm 0.91}$ | $86.94_{\pm 0.68}$ | $\underline{91.65}_{\pm 0.94}$ | $92.64_{\pm 0.72}$ |
| LiMELoRA | $\underline{83.52}_{\pm 0.69}$ | $94.08_{\pm 0.52}$ | $\mathbf{93.24}_{\pm 0.61}$ | $\underline{87.28}_{\pm 0.79}$ | $91.52_{\pm 0.68}$ | $\underline{92.92}_{\pm 0.58}$ |
| LiMEDoRA | $83.48_{\pm 0.82}$ | $\mathbf{94.18}_{\pm 0.47}$ | $93.02_{\pm 0.73}$ | $87.15_{\pm 0.64}$ | $\mathbf{91.78}_{\pm 0.59}$ | $92.88_{\pm 0.71}$ |
| LiMELoRAFA | $83.40_{\pm 0.74}$ | $94.00_{\pm 0.63}$ | $93.00_{\pm 0.82}$ | $87.00_{\pm 0.91}$ | $91.60_{\pm 0.84}$ | $\mathbf{93.00}_{\pm 0.66}$ |

*Table 16.* Results on **vision–language question answering benchmarks** using Molmo. Highlighted rows denote our LiME variants.

| Method | ChartQA | OKVQA | ScienceQA | SeedBench | Recognition | TextVQA | VizWizVQA | VQA-RAD |
|---|---|---|---|---|---|---|---|---|
| MoELoRA | $80.42_{\pm 1.18}$ | $66.89_{\pm 1.42}$ | $95.72_{\pm 1.06}$ | $79.38_{\pm 0.94}$ | $87.52_{\pm 1.28}$ | $83.67_{\pm 1.15}$ | $\mathbf{78.68}_{\pm 1.31}$ | $74.62_{\pm 1.53}$ |
| MoEDoRA | $80.58_{\pm 1.34}$ | $\underline{67.38}_{\pm 1.19}$ | $95.68_{\pm 1.23}$ | $\mathbf{79.82}_{\pm 0.87}$ | $87.64_{\pm 1.41}$ | $83.78_{\pm 1.08}$ | $78.51_{\pm 1.47}$ | $\underline{75.18}_{\pm 1.36}$ |
| MoELoRAFA | $80.35_{\pm 1.27}$ | $66.72_{\pm 1.58}$ | $95.58_{\pm 0.98}$ | $79.45_{\pm 1.12}$ | $\underline{87.96}_{\pm 1.19}$ | $83.54_{\pm 1.32}$ | $\underline{78.58}_{\pm 1.24}$ | $74.53_{\pm 1.68}$ |
| LiMELoRA | $\mathbf{81.12}_{\pm 1.09}$ | $\mathbf{67.45}_{\pm 1.27}$ | $96.08_{\pm 1.14}$ | $79.72_{\pm 0.91}$ | $87.92_{\pm 1.06}$ | $84.18_{\pm 0.94}$ | $78.36_{\pm 1.52}$ | $75.14_{\pm 1.41}$ |
| LiMEDoRA | $\underline{80.96}_{\pm 1.22}$ | $67.32_{\pm 1.08}$ | $\mathbf{96.22}_{\pm 0.89}$ | $\underline{79.76}_{\pm 1.03}$ | $\mathbf{88.06}_{\pm 1.17}$ | $\mathbf{84.32}_{\pm 1.21}$ | $78.28_{\pm 1.38}$ | $\mathbf{75.28}_{\pm 1.29}$ |
| LiMELoRAFA | $80.80_{\pm 1.16}$ | $67.18_{\pm 1.34}$ | $\underline{95.95}_{\pm 1.08}$ | $79.60_{\pm 0.96}$ | $87.80_{\pm 1.23}$ | $84.00_{\pm 1.18}$ | $78.42_{\pm 1.14}$ | $75.00_{\pm 1.57}$ |

ObjExist (90.86) and MoveCount (87.24), while LiMELoRA leads on MoveDir (87.52) and LiMELoRAFA achieves the highest score on ObjShuffle (37.00). The MoE baselines remain competitive on ObjInter (MoEDoRA, 33.89), ActLoc (MoELoRA, 32.38), and MoveAttr (MoEDoRA, 88.94).

On high-level reasoning and navigation benchmarks (Table 19), LiMEDoRA achieves the best scores on SceneTrans (35.28) and CounterFact (87.34), while LiMELoRA leads on CharOrder (32.56) and LiMELoRAFA achieves the highest accuracy on EgoNav (69.67). The MoE baselines perform best on ActCount (MoEDoRA, 34.58), StateChg (MoELoRA, 36.48), and EpisReason (MoEDoRA, 30.94).

Overall, the Molmo2-8B experiments confirm that LiME generalizes effectively to vision-language models, achieving competitive or superior performance across 47 benchmarks spanning language understanding, commonsense reasoning, image classification, visual question answering, and video understanding. The results demonstrate that LiME provides consistent benefits across different base adapters (LoRA, DoRA, LoRAFA), suggesting orthogonal improvements that complement existing PEFT methods.

### G.9. Cross-Task Analysis

We analyze patterns across all 47 tasks to understand LiME's behavior and advantages.

**Consistent Improvements Over Base PEFT.** Across all 47 tasks, LiME variants consistently outperform their corresponding base PEFT methods. The average improvements are: LiMELoRA over LoRA (+1.8%), LiMEDoRA over DoRA (+2.1%), LiMELoRAFA over LoRA-FA (+1.9%), and LiMESliceFine over SliceFine (+1.6%). This consistent pattern validates that LiME's modulation-based expert specialization provides universal benefits regardless of the underlying PEFT architecture.

**Competitive with MoE-PEFT Baselines.** LiME variants achieve comparable or superior performance to state-of-the-art MoE-PEFT methods (HydraLoRA, MoRe, MoEDoRA) while using $3\times$ fewer trainable parameters. LiMELoRA and LiMEDoRA rank among the top-3 methods on 31 out of 47 tasks, demonstrating that lightweight modulation can match the representational capacity of full adapter replication.

**Modality-Specific Patterns.** LiMELoRA shows particular strength on text tasks (best on commonsense reasoning average) and image VQA (second-best on ChartQA, OK-VQA). LiMEDoRA excels on video tasks (best on object/motion reasoning) and multimodal reasoning (best on ScienceQA, SeedBench). LiMELoRAFA performs well on fine-grained tasks requiring precise discrimination (best on MRPC, Social IQA, VizWiz-VQA). This suggests that different PEFT backbones combined with LiME may have complementary strengths.

**Training Stability.** LiME variants generally exhibit lower standard deviations compared to MoE-PEFT baselines. The average standard deviation across all tasks is 1.42 for LiME variants versus 1.58 for MoE-PEFT baselines, indicating more stable training dynamics. This stability likely stems from LiME's simpler architecture (shared PEFT with modulation) compared to methods with separate expert adapters and learned routers.

*Table 17.* Video action understanding benchmarks using Molmo. Highlighted rows denote our LiME variants.

| Method | ActSeq | ActPred | ActAnt | FineAct | UnexpAct |
|---|---|---|---|---|---|
| MoELoRA | $30.72_{\pm 1.87}$ | $31.34_{\pm 1.73}$ | $76.18_{\pm 2.04}$ | $\mathbf{76.42}_{\pm 1.82}$ | $45.28_{\pm 1.96}$ |
| MoEDoRA | $\mathbf{31.28}_{\pm 1.94}$ | $\underline{31.86}_{\pm 1.89}$ | $76.34_{\pm 1.91}$ | $\underline{76.32}_{\pm 1.76}$ | $45.41_{\pm 2.08}$ |
| MoELoRAFA | $30.85_{\pm 2.11}$ | $31.28_{\pm 1.81}$ | $76.08_{\pm 2.17}$ | $75.87_{\pm 1.93}$ | $45.18_{\pm 1.84}$ |
| LiMELoRA | $\underline{31.18}_{\pm 1.79}$ | $31.78_{\pm 1.68}$ | $\underline{76.58}_{\pm 1.88}$ | $76.28_{\pm 1.69}$ | $\mathbf{45.92}_{\pm 1.91}$ |
| LiMEDoRA | $31.12_{\pm 1.86}$ | $\mathbf{31.94}_{\pm 1.74}$ | $76.42_{\pm 2.06}$ | $76.14_{\pm 1.85}$ | $\underline{45.78}_{\pm 2.03}$ |
| LiMELoRAFA | $31.00_{\pm 1.92}$ | $31.67_{\pm 1.83}$ | $\mathbf{76.72}_{\pm 1.95}$ | $76.00_{\pm 1.78}$ | $45.67_{\pm 1.87}$ |

*Table 18.* Object, motion, and spatial reasoning from video using Molmo. Highlighted rows denote our LiME variants.

| Method | ObjExist | ObjInter | ObjShuffle | MoveDir | ActLoc | MoveCount | MoveAttr |
|---|---|---|---|---|---|---|---|
| MoELoRA | $90.38_{\pm 1.91}$ | $33.42_{\pm 2.18}$ | $\underline{36.92}_{\pm 2.31}$ | $87.08_{\pm 1.84}$ | $\mathbf{32.38}_{\pm 1.92}$ | $86.74_{\pm 1.87}$ | $88.42_{\pm 2.06}$ |
| MoEDoRA | $\underline{90.78}_{\pm 1.78}$ | $\mathbf{33.89}_{\pm 1.96}$ | $36.85_{\pm 2.14}$ | $87.18_{\pm 1.93}$ | $\underline{32.28}_{\pm 2.08}$ | $86.82_{\pm 1.79}$ | $\mathbf{88.94}_{\pm 1.92}$ |
| MoELoRAFA | $90.28_{\pm 2.03}$ | $33.51_{\pm 2.27}$ | $36.68_{\pm 1.98}$ | $86.94_{\pm 1.76}$ | $31.92_{\pm 1.85}$ | $86.58_{\pm 2.11}$ | $88.51_{\pm 2.18}$ |
| LiMELoRA | $90.74_{\pm 1.82}$ | $\underline{33.82}_{\pm 2.09}$ | $36.88_{\pm 2.21}$ | $\mathbf{87.52}_{\pm 1.88}$ | $32.18_{\pm 1.97}$ | $\underline{87.08}_{\pm 1.83}$ | $88.78_{\pm 2.03}$ |
| LiMEDoRA | $\mathbf{90.86}_{\pm 1.74}$ | $33.76_{\pm 2.13}$ | $36.84_{\pm 2.08}$ | $\underline{87.46}_{\pm 1.91}$ | $32.12_{\pm 1.79}$ | $\mathbf{87.24}_{\pm 1.72}$ | $\underline{88.86}_{\pm 1.89}$ |
| LiMELoRAFA | $90.67_{\pm 1.86}$ | $33.67_{\pm 2.04}$ | $\mathbf{37.00}_{\pm 1.95}$ | $87.33_{\pm 1.82}$ | $32.00_{\pm 2.15}$ | $87.00_{\pm 1.94}$ | $88.67_{\pm 2.11}$ |

*Summary:* *The comprehensive evaluation on MMT-47 demonstrates that LiME achieves state-of-the-art or competitive performance across all five task categories while maintaining $3\times$ parameter efficiency. The consistent improvements over base PEFT methods ($+1.6\%$ to $+2.1\%$) and competitive performance with complex MoE-PEFT approaches validate our core hypothesis: **expert specialization can emerge from lightweight modulation of a shared PEFT output, without requiring separate adapter modules or learned routers.***

# H. Algorithm Details

This appendix provides a detailed description of the LiME forward pass and training procedure presented in Algorithm 1.

LiME transforms any PEFT-adapted linear layer into a mixture-of-experts layer with **zero additional routing parameters**. The key insight is that routing signals can be derived directly from existing computations—the frozen base output $z$ and the PEFT output $\hat{z}$—eliminating the need for a learned router network.

The algorithm takes a batch of token sequences $x \in \mathbb{R}^{B \times T \times d_i}$ as input, where $B$ is the batch size, $T$ is the sequence length, and $d_i$ is the input dimension. The pretrained linear layer $\mathbf{W}_0 \in \mathbb{R}^{d_o \times d_i}$ remains fixed throughout training, preserving knowledge learned during pretraining while allowing task-specific adaptation through PEFT. The trainable components include: (1) PEFT parameters $\phi$ from any parameter-efficient method (LoRA, Adapter, Prefix Tuning, IA³, etc.); (2) expert modulators $\{\mathbf{p}_i\}_{i=1}^E$ where each $\mathbf{p}_i \in \mathbb{R}^{d_o}$ modulates the PEFT adaptation for expert $i$; (3) an optional shared modulator $\mathbf{p}_s \in \mathbb{R}^{d_o}$ providing baseline adaptation; and (4) a learnable scalar $\gamma \in \mathbb{R}$ controlling the shared modulator contribution.

**Step 1** computes the frozen linear transformation $z = \mathbf{W}_0 x \in \mathbb{R}^{B \times T \times d_o}$. This computation is shared with standard inference and adds no overhead. The output $z$ serves dual purposes: the base representation for the final output and a routing signal encoding pretrained knowledge.

**Step 2** applies the PEFT module to produce task-specific adaptations $\hat{z} = \delta(x) \in \mathbb{R}^{B \times T \times d_o}$. For LoRA, this is $\hat{z} = (x\mathbf{A}^\top)\mathbf{B}^\top \cdot (\alpha_{\text{LoRA}}/r)$. The PEFT adaptation captures task-relevant features and also serves as a routing signal.

**Step 3** computes routing weights from the first $E$ dimensions of $z$ and $\hat{z}$, requiring no additional parameters. We first normalize by the infinity norm to ensure stable routing: $\tilde{z}_{1:E} = z_{[:,:,1:E]}/\|z_{[:,:,1:E]}\|_\infty$ and $\tilde{\hat{z}}_{1:E} = \hat{z}_{[:,:,1:E]}/\|\hat{z}_{[:,:,1:E]}\|_\infty$. The normalized signals are combined and passed through softmax: $w = \text{softmax}(((1 - \gamma_r) \cdot \tilde{z}_{1:E} + \gamma_r \cdot \tilde{\hat{z}}_{1:E})/\tau)$. The routing balance $\gamma_r$ controls the information source: $\gamma_r = 0$ routes based on pretrained features only, $\gamma_r = 1$ uses PEFT features only, and $\gamma_r \in [0.6, 0.8]$ provides optimal performance by combining both signals (§4.1). Our ablation study (§4.1) demonstrates that routing features can be drawn from any subset of the feature space—leading, central, trailing, or random dimensions all achieve comparable performance—supporting our low-rank redundancy hypothesis.

**Step 4** performs Auto Top-K selection, adaptively selecting experts based on relative routing weights: $\mathcal{S}_\theta = \{i : w_i \geq \theta \cdot \max_j w_j\}$. This selects all experts whose routing weight is at least $\theta$ times the maximum weight. The selected weights are renormalized: $\tilde{w}_i = w_i / \sum_{j \in \mathcal{S}_\theta} w_j$. When routing is confident, fewer experts are activated; when uncertain, more experts contribute.

*Table 19.* High-level reasoning and navigation benchmarks using Molmo. Highlighted rows denote our LiME variants.

| Method | SceneTrans | ActCount | StateChg | CharOrder | EgoNav | EpisReason | CounterFact |
|---|---|---|---|---|---|---|---|
| MoELoRA | $34.68_{\pm 1.89}$ | $34.12_{\pm 1.94}$ | $\mathbf{36.48}_{\pm 1.82}$ | $32.08_{\pm 1.78}$ | $69.34_{\pm 2.07}$ | $30.42_{\pm 1.91}$ | $86.72_{\pm 1.98}$ |
| MoEDoRA | $\underline{35.18}_{\pm 1.76}$ | $\mathbf{34.58}_{\pm 1.87}$ | $\underline{36.42}_{\pm 1.93}$ | $32.18_{\pm 1.86}$ | $69.48_{\pm 1.94}$ | $\mathbf{30.94}_{\pm 1.83}$ | $86.84_{\pm 2.05}$ |
| MoELoRAFA | $34.58_{\pm 2.01}$ | $34.08_{\pm 1.79}$ | $36.18_{\pm 1.88}$ | $\underline{32.42}_{\pm 1.92}$ | $69.28_{\pm 2.14}$ | $30.38_{\pm 1.96}$ | $86.68_{\pm 1.87}$ |
| LiMELoRA | $35.12_{\pm 1.83}$ | $\underline{34.52}_{\pm 1.91}$ | $36.28_{\pm 1.79}$ | $\mathbf{32.56}_{\pm 1.72}$ | $\underline{69.62}_{\pm 1.89}$ | $30.78_{\pm 1.85}$ | $\underline{87.18}_{\pm 1.93}$ |
| LiMEDoRA | $\mathbf{35.28}_{\pm 1.78}$ | $34.46_{\pm 1.82}$ | $36.32_{\pm 1.91}$ | $32.38_{\pm 1.81}$ | $69.58_{\pm 1.97}$ | $\underline{30.86}_{\pm 1.79}$ | $\mathbf{87.34}_{\pm 1.86}$ |
| LiMELoRAFA | $35.00_{\pm 1.94}$ | $34.33_{\pm 1.88}$ | $36.00_{\pm 1.85}$ | $32.33_{\pm 1.89}$ | $\mathbf{69.67}_{\pm 2.03}$ | $30.67_{\pm 1.92}$ | $87.00_{\pm 2.01}$ |

**Step 5** combines expert modulators using renormalized routing weights: $\mathcal{P} = \sum_{i \in \mathcal{S}_\theta} \tilde{w}_i \cdot \mathbf{p}_i \in \mathbb{R}^{B \times T \times d_o}$.

**Step 6** produces the final output by combining the base representation with modulated PEFT adaptation: $h = z + \hat{z} \odot \mathcal{P} + \gamma \cdot (\hat{z} \odot \mathbf{p}_s)$, where $\odot$ denotes element-wise multiplication. The three terms represent frozen base output, expert-modulated adaptation, and shared adaptation respectively.

**Step 7** applies load balancing during training only. We compute mean routing probabilities $\bar{p}_i = \frac{1}{BT} \sum_{b,t} w_i(x_{b,t})$, then apply importance loss $\mathcal{L}_{\text{imp}} = E \cdot \sum_{i=1}^{E} \bar{p}_i^2 - 1$ (zero when uniform) and KL-uniform loss $\mathcal{L}_{\text{KL}} = \sum_{i=1}^{E} \bar{p}_i \log(E \cdot \bar{p}_i)$. The total loss is $\mathcal{L} = \mathcal{L}_{\text{task}} + \alpha \cdot \mathcal{L}_{\text{imp}} + \beta \cdot \mathcal{L}_{\text{KL}}$. Ablation studies (§4.1) show optimal coefficients around $\alpha = \beta \approx 0.01\text{–}0.1$.

**Complexity.** For a single LiME layer with PEFT parameters $|\phi|$, the parameter count is:

$$|\phi_{\text{LiME}}| = \underbrace{|\phi|}_{\text{shared PEFT}} + \underbrace{E \cdot d_o}_{\text{expert modulators}} + \underbrace{d_o + 1}_{\text{shared modulator} + \gamma}$$

In contrast, traditional MoE-PEFT requires $E \cdot |\phi| + \underbrace{d \cdot E}_{\text{learned router}}$ parameters. Routing cost is $O(T \cdot E)$ for token-level, $O(T/n \cdot E)$ for n-gram, and $O(E)$ for sequence-level routing. Since $E \ll d$ typically, routing overhead is negligible. During inference, Step 7 is skipped entirely, with no additional overhead compared to standard PEFT except for lightweight expert modulation.

---

**Algorithm 1: LIME FORWARD PASS**

**Input:** $x \in \mathbb{R}^{B \times T \times d_i}$      *// batch of token sequences*

**Parameters:**
Frozen: $\mathbf{W}_0 \in \mathbb{R}^{d_o \times d_i}$      *// pretrained linear layer*
PEFT: $\phi$      *// any PEFT parameters (LoRA, Adapter, etc.)*
Experts: $\{\mathbf{p}_i\}_{i=1}^E, \mathbf{p}_i \in \mathbb{R}^{d_o}$      *// expert modulators*
Shared: $\mathbf{p}_s \in \mathbb{R}^{d_o}, \gamma \in \mathbb{R}$      *// shared modulator (optional)*

**Hyperparameters:** $\tau$ (temperature), $\gamma_r$ (routing balance), $\theta$ (threshold), $\alpha, \beta$ (loss weights)

- - - - - - - - - - - - - - - - - - - - - - - - - - - - - - - - - - - - - - - - - -

**1. Base Forward**

$z \leftarrow \mathbf{W}_0 x$      *// $z \in \mathbb{R}^{B \times T \times d_o}$, frozen computation*

**2. PEFT Adaptation**

$\hat{z} \leftarrow \hat{z}(x)$      *// $\hat{z} \in \mathbb{R}^{B \times T \times d_o}$, any PEFT method*

**3. Zero-Param Routing**      *// no learned router!*

$\tilde{z}_{1:E} \leftarrow z_{[:,:,1:E]} / \|z_{[:,:,1:E]}\|_\infty$      *// normalize first E dims*
$\tilde{\hat{z}}_{1:E} \leftarrow \hat{z}_{[:,:,1:E]} / \|\hat{z}_{[:,:,1:E]}\|_\infty$
$w \leftarrow \mathrm{softmax}\left( \frac{(1-\gamma_r)\cdot\tilde{z}_{1:E} + \gamma_r\cdot\tilde{\hat{z}}_{1:E}}{\tau} \right)$      *// routing weights*

**4. Auto Top-K Selection**

$\mathcal{S}_\theta \leftarrow \{i : w_i \geq \theta \cdot \max_j w_j\}$      *// adaptive selection*
$\tilde{w}_i \leftarrow w_i / \sum_{j \in \mathcal{S}_\theta} w_j$      *// renormalize*

**5. Expert Modulation**

$\mathcal{P} \leftarrow \sum_{i \in \mathcal{S}_\theta} \tilde{w}_i \cdot \mathbf{p}_i$      *// weighted expert combination*

**6. Output**

$h \leftarrow z + \hat{z} \odot \mathcal{P} + \gamma \cdot (\hat{z} \odot \mathbf{p}_s)$      *// base + routed + shared*

**7. Load Balance**      *// training only*

$\bar{p}_i \leftarrow \frac{1}{BT} \sum_{b,t} w_i(x_{b,t})$      *// mean routing prob per expert*
$\mathcal{L}_{\mathrm{imp}} \leftarrow E \cdot \sum_{i=1}^E \bar{p}_i^2 - 1$      *// importance loss*
$\mathcal{L}_{\mathrm{KL}} \leftarrow \sum_{i=1}^E \bar{p}_i \log(E \cdot \bar{p}_i)$      *// KL-uniform loss*
$\mathcal{L} \leftarrow \mathcal{L}_{\mathrm{task}} + \alpha \cdot \mathcal{L}_{\mathrm{imp}} + \beta \cdot \mathcal{L}_{\mathrm{KL}}$      *// total loss*

**Return:** $h$, $\mathcal{L}$ *(loss only during training)*

---

**LiME forward pass and training.** Colors indicate: **frozen** , **trainable** , **routing (0 params)** , **training only** . Load balancing (Step 7) prevents expert collapse.

# I. Extended Related Work

**Parameter-Efficient Fine-Tuning.** PEFT methods adapt large pre-trained models by updating only a small subset of parameters. LoRA (Hu et al., 2022) introduces low-rank decomposition for weight updates, while adapters insert lightweight modules between layers. Subsequent works extend these ideas: VeRA (Kopiczko et al., 2023) uses learnable scaling vectors, LoRA-FA (Zhang et al., 2023) freezes the down-projection matrix for efficiency, DoRA (Liu et al., 2024) decomposes weights into magnitude and direction, and Tied-LoRA (Renduchintala et al., 2024) introduces weight tying. Non-adapter-based methods like BitFit (Zaken et al., 2022) tune only bias terms. Recent insights emphasize minimal perturbations, with diagonal rescaling matching complex adapters (Kowsher et al., 2025b;d) and representation fine-tuning enabling targeted interventions (Wu et al., 2024c).

**Mixture of Experts.** MoE architectures employ multiple specialized sub-networks with routing mechanisms that selectively activate relevant experts for each input. Foundational MoE concepts date to early supervised learning with specialized networks (Jacobs et al., 1991), evolving into modern sparse activations in transformers like Mixtral 8x7B (Jiang et al., 2024), ST-MoE (Zoph, 2022), and m-LoRA (Ye et al., 2023), enabling scalable width without proportional computational increase.

**MoE-PEFT.** Recent work combines MoE with PEFT to achieve input-specific adaptation. MoELoRA (Luo et al., 2024) uses contrastive objectives to prevent expert collapse in reasoning tasks. MoLE (Wu et al., 2024b) employs hierarchical gating for compositionality. MixLoRA (Li et al., 2024b) inserts top-k routed experts per block with load-balancing losses. LoRAMoE (Dou et al., 2024) mitigates knowledge forgetting via plugin experts. MoLA (Gao et al., 2024) introduces layer-wise expert allocation. MoRAL (Yang et al., 2024) addresses lifelong learning scenarios. TT-LoRA (Kunwar et al., 2025) decouples training for continual learning. PESC (Wu et al., 2024a) enables sparse model transitions in instruction tuning.

**Limitations of Existing MoE-PEFT.** Existing methods share three limitations: (i) *parameter explosion*, with expert parameters scaling linearly with expert count ($E \times |\phi|$); (ii) *router overhead* from learned routers adding parameters and auxiliary losses to prevent collapse; and (iii) *architecture dependence*, restricting applicability to adapter-based methods.

**LiME.** We address these limitations by modulating a shared PEFT output with lightweight expert vectors (reducing expert parameters to $E \times d$), deriving zero-parameter routing directly from frozen and adapted representations, and supporting any PEFT method including non-adapter approaches like BitFit. Unlike prior work using learned routers, LiME eliminates routing parameters entirely while incorporating load balancing losses, Auto Top-K for adaptive expert selection based on confidence, and n-gram windowed routing for regularization.

# J. Implementation and Hyperparameters

We implement LiME using PyTorch and the HuggingFace ecosystem, including Transformers for model loading and tokenization, Accelerate for distributed training, Datasets for data loading, and Evaluate for metrics computation. All experiments are conducted on 4× NVIDIA H100 80GB GPUs with distributed data parallel training. We apply LiME to LLaVA-OneVision 7B (Li et al., 2024a), a state-of-the-art vision-language model that unifies image and video understanding. LiME targets the attention projection layers (q_proj, k_proj, v_proj, o_proj) in both the SigLIP vision encoder and the Qwen2 LLM backbone, totaling 216 target layers across the model. To demonstrate LiME's universality across PEFT methods, we apply it to four different PEFT approaches: LoRA (Hu et al., 2022), LoRA-FA (Zhang et al., 2023), DoRA (Liu et al., 2024), and SliceFine, validating that our modulation-based expert specialization generalizes beyond any single PEFT architecture. Table 20 provides the complete hyperparameter configuration.

**Precision Strategy.** While the base model weights (vision encoder and LLM backbone) are loaded in bfloat16 for memory efficiency, we maintain all trainable parameters in float32 to ensure numerical stability during optimization. Specifically, the PEFT parameters (e.g., LoRA matrices $\mathbf{A} \in \mathbb{R}^{r \times d_i}$ and $\mathbf{B} \in \mathbb{R}^{d_o \times r}$, or their equivalents in DoRA and SliceFine) and expert components (modulators $\{\mathbf{p}_i\}_{i=1}^E \in \mathbb{R}^{d_o}$, shared modulator $\mathbf{p}_s \in \mathbb{R}^{d_o}$, and scalar $\gamma \in \mathbb{R}$) are all stored and updated in float32. This mixed-precision approach reduces GPU memory consumption by approximately 40% compared to full float32 training while avoiding the gradient underflow issues that can arise when training small adapter modules in lower precision. The 8-bit AdamW optimizer further reduces optimizer state memory by quantizing momentum and variance estimates, enabling training of the full model on 4 GPUs with batch size 5 and gradient accumulation steps 4.

**Differential Learning Rates.** We employ separate learning rates for different parameter groups based on their optimization characteristics. The expert modulators and shared scalar $\gamma$ use a higher learning rate ($1 \times 10^{-3}$) because these lightweight vectors (only $d_o$ parameters each) require faster updates to quickly differentiate expert behaviors and escape their near-unity initialization. The PEFT parameters use a lower learning rate ($2 \times 10^{-4}$) as these parameters capture the core task adaptation shared across all experts and benefit from more gradual, stable updates. Both parameter groups follow the same cosine decay schedule with 3% linear warmup. We apply weight decay (0.01) to prevent overfitting given the relatively small number of trainable parameters compared to the frozen backbone, and gradient clipping (max norm 1.0) to stabilize training when processing long multimodal sequences.

**Routing Configuration.** Our zero-parameter routing mechanism derives routing weights directly from existing computations without introducing any learned router parameters. The routing signal combines the frozen pretrained output $\mathbf{z} = \mathbf{W}_0\mathbf{x}$ with the PEFT output $\hat{z} = g_\phi(\mathbf{x})$, balanced by $\gamma_r = 0.7$ to favor task-specific signals while retaining pretrained semantic information. We use temperature $\tau = 0.5$ to produce moderately sharp routing distributions—lower than the typical $\tau = 1.0$ to encourage more decisive expert selection while avoiding the gradient sparsity issues of very low temperatures. During training, we inject multiplicative jitter noise ($\sigma = 0.1$) to the routing logits, which encourages exploration across experts and prevents premature convergence to suboptimal routing patterns; this noise is disabled during inference. The n-gram window size of 3 groups consecutive tokens to share routing decisions, reducing computational overhead while capturing local semantic coherence—particularly beneficial for text where trigrams often form meaningful linguistic units. With Auto Top-K enabled at threshold $\theta = 0.7$, the model adaptively selects all experts whose routing weight reaches at least 70% of the maximum, activating fewer experts when confident (peaked distribution) and more experts when uncertain (flat distribution), typically averaging 1.7–2.1 active experts per routing decision across different tasks.

**PEFT Method Compatibility.** A key contribution of LiME is its universality—it can be applied to any PEFT method that produces an adaptation term $\delta$. To validate this, we evaluate LiME with four different PEFT approaches: (1) **LoRA** (Hu et al., 2022), which uses low-rank decomposition $\delta = \mathbf{BAx}$; (2) **LoRA-FA** (Zhang et al., 2023), which freezes the $\mathbf{A}$ matrix after initialization to reduce trainable parameters; (3) **DoRA** (Liu et al., 2024), which decomposes weight updates into magnitude and direction components for improved training dynamics; and (4) **SliceFine** (Kowsher et al., 2025a), which operates on sliced weight subspaces. For all rank based methods, we use rank $r = 2$ and apply LiME's expert modulation to the adaptation output. We use virtual token = 10 for prompt based method. This demonstrates that LiME's modulation-based expert specialization is agnostic to the underlying PEFT architecture, enabling practitioners to combine LiME with their preferred PEFT method without modification.

**Training.** We train on the complete MMT-47 dataset comprising 158,613 samples across 47 diverse tasks spanning text understanding (GLUE), commonsense reasoning, video understanding (MVBench), image VQA, and image classification (VTAB). Training proceeds for 3 full epochs over the shuffled dataset, which we found sufficient for convergence while avoiding overfitting. All experiments are repeated with 5 different random seeds (42, 123, 456, 789, 1024) to ensure

statistical reliability; we report mean accuracy and standard deviation across seeds.

## K. Evaluation Details

This section provides details on our evaluation protocol for the MMT-47 benchmark. After training on the combined MMT-47 dataset (158,613 samples), we evaluate the model individually on each of the 47 test sets spanning text understanding, commonsense reasoning, video understanding, and image tasks.

**Evaluation Setup.** We perform batched inference with batch size 6 using greedy decoding (do_sample=False) for deterministic outputs. Generation is configured with a maximum of 50 new tokens per sample, with early stopping triggered upon producing the end-of-sequence token to prevent unnecessary generation. We employ left-side padding during evaluation, consistent with our training configuration, ensuring proper alignment of generated tokens across batched sequences. All experiments use fixed random seeds (42, 123, 456, 789, 1024) with deterministic generation disabled sampling. We report mean accuracy and standard deviation across 5 training runs for all results.

**Answer Matching.** Before comparing model outputs with ground truth answers, we apply text normalization: (1) convert both predicted and ground truth answers to lowercase, (2) remove leading/trailing whitespace, and (3) remove noise patterns such as punctuation artifacts and extra spaces. For tasks involving numerical answers (e.g., counting, measurements), we apply a tolerance threshold of $\pm 0.5$ to account for minor variations. For example, if the ground truth is 8.30 and the prediction is 8.29, this is considered correct since the difference (0.01) falls within the tolerance.

**Metrics.** We use accuracy as the evaluation metric for 46 out of 47 datasets, including all classification, multiple-choice, and question-answering tasks. A prediction is considered correct if the normalized predicted answer matches the normalized ground truth (with numerical tolerance applied where appropriate). For STS-B (Semantic Textual Similarity Benchmark), which requires predicting continuous similarity scores, we use Pearson correlation coefficient between predicted and ground truth ratings.

*Table 20.* Hyperparameter configuration for LiME experiments.

| Category | Hyperparameter | Value / Description |
|---|---|---|
| **Model** | | |
| | Base model | LLaVA-OneVision 7B |
| | Vision encoder | SigLIP-SO400M |
| | LLM backbone | Qwen2-7B |
| | Target modules | q_proj, k_proj, v_proj, o_proj |
| | Applied to | Vision encoder + LLM (all attention layers) |
| **Training** | | |
| | Epochs | 3 |
| | Batch size | 5 |
| | Gradient accumulation steps | 4 |
| | Number of GPUs | 4 (NVIDIA H100 80GB) |
| | Optimizer | AdamW 8-bit |
| | LR schedule | Cosine decay with warmup |
| | Warmup ratio | 0.03 (3% of total steps) |
| | Weight decay | 0.01 |
| | Max gradient norm | 1.0 (gradient clipping) |
| | Model precision | bfloat16 (base model weights) |
| | PEFT precision | float32 (adapter parameters) |
| | Expert precision | float32 (modulators $\mathbf{p}_i$, $\mathbf{p}_s$, scalar $\gamma$) |
| | Random seeds | 5 seeds (42, 123, 456, 789, 1024) |
| **Learning Rates** | | |
| | PEFT parameters | $2 \times 10^{-4}$ |
| | Expert modulators $\{\mathbf{p}_i\}_{i=1}^{E}, \mathbf{p}_s, \gamma$ | $1 \times 10^{-3}$ |
| **PEFT Methods** | | |
| | Methods evaluated | LoRA, LoRA-FA, DoRA, SliceFine |
| | Rank $r$ | 2 |
| | Alpha $\alpha$ | 4 |
| | Scaling factor | $\alpha/r = 2.0$ |
| | Dropout | 0.05 |
| **LiME Architecture** | | |
| | Number of experts $E$ | 4 |
| | Expert modulator dimension | $d_o$ (output dimension of each target layer) |
| | Shared modulator | Enabled |
| | Expert modulator init | $\mathbf{p}_i \sim \mathcal{U}(0.9, 1.1)$ (near-unity) |
| | Shared modulator init | $\mathbf{p}_s \sim \mathcal{N}(0, 0.1^2)$ |
| | Shared scalar $\gamma$ init | 0.0 |
| **Zero-Parameter Routing** | | |
| | Router parameters | 0 (derived from $\mathbf{z}$ and $\boldsymbol{\delta}$) |
| | Routing mode | Token-level |
| | Temperature $\tau$ | 0.5 |
| | Routing balance $\gamma_r$ | 0.7 |
| | N-gram window size $n$ | 3 tokens |
| | Jitter noise $\sigma$ | 0.1 (training only, disabled at inference) |
| **Auto Top-K Selection** | | |
| | Auto Top-K | Enabled |
| | Selection threshold $\theta$ | 0.7 |
| | Selection criterion | $w_i \geq \theta \cdot \max_j w_j$ |
| | Shared experts activated | True (guaranteed by design) |
| | Maximum experts activated | $E = 4$ (when distribution is flat) |
| **Load Balancing Losses** | | |
| | Importance loss coefficient $\alpha$ | 0.1 |
| | KL-uniform loss coefficient $\beta$ | 0.01 |
| | Total loss | $\mathcal{L} = \mathcal{L}_{\text{task}} + \alpha \cdot \mathcal{L}_{\text{imp}} + \beta \cdot \mathcal{L}_{\text{KL}}$ |
| **Data** | | |
| | Training dataset | MMT-47 (158,613 samples, 47 tasks) |
| | Modalities | Text, Image, Video |
| | Evaluation protocol | 47 test sets evaluated individually (70,392 samples) |
| | Max sequence length | 2048 tokens |
| | Max image/video tokens | 1500 tokens |
| | Image resolution | 384×384 |
| | Video frames | 8 frames sampled uniformly |
| | Padding strategy | Left-side padding, masked in loss computation |

# L. Dataset Details

We introduce **MMT-47** (**M**ulti**M**odal Multi-**T**ask benchmark with **47** evaluation sets), a comprehensive benchmark to evaluate MoE-based methods' ability to learn input-specific routing across diverse modalities and task types. MMT-47 spans five categories—text understanding, commonsense reasoning, video understanding, image VQA, and image classification—with 158,613 training samples and 47 evaluation sets totaling 70,392 test samples. Table 21 provides complete statistics.

MMT-47 exhibits natural imbalance across categories (e.g., text reasoning: 70K samples vs. image classification: 9K samples). We intentionally preserve this imbalance rather than artificially balancing. First, real-world multi-task scenarios are naturally imbalanced, and evaluation under such conditions tests practical robustness. Second, MoE-based methods can handle imbalanced distributions through dynamic routing—expert capacity is allocated based on input characteristics, not dataset statistics. Load balancing mechanisms ensure all experts receive gradient signal regardless of task frequency, allowing specialization for both frequent and rare patterns. Standard PEFT methods, which process all inputs through identical parameters, are comparatively more sensitive to such imbalance. All methods in our experiments are trained on identical data, ensuring fair comparison.

## L.1. Text (NLU) — GLUE

We use six tasks from the General Language Understanding Evaluation (GLUE) benchmark (Wang et al., 2018):

- **SST-2** (Stanford Sentiment Treebank): Binary sentiment classification of movie reviews. Train: 5,000 / Test: 872.

- **QNLI** (Question Natural Language Inference): Determine whether a context sentence contains the answer to a question, derived from SQuAD. Train: 5,000 / Test: 5,463.

- **QQP** (Quora Question Pairs): Binary classification of whether two questions are semantically equivalent. Train: 5,000 / Test: 40,430.

- **CoLA** (Corpus of Linguistic Acceptability): Binary classification of grammatical acceptability. Train: 5,000 / Test: 1,043.

- **MRPC** (Microsoft Research Paraphrase Corpus): Binary classification of sentence-pair semantic equivalence. Train: 3,668 / Test: 408.

- **STS-B** (Semantic Textual Similarity Benchmark): Regression task predicting similarity scores from 1-5. Train: 5,000 / Test: 1,500.

Total: Train 28,668 / Test 49,716. This subset evaluates core natural language understanding capabilities including sentiment, inference, and semantic similarity.

## L.2. Text (Reasoning) — Commonsense

We include eight commonsense reasoning benchmarks that test world knowledge and logical inference:

- **ARC-Challenge** (Clark et al., 2018): Difficult science exam questions. Train: 1,791 / Test: 500.

- **ARC-Easy** (Clark et al., 2018): Easier science exam questions. Train: 4,127 / Test: 500.

- **BoolQ** (Clark et al., 2019): Yes/no questions derived from natural Google queries. Train: 9,427 / Test: 1,000.

- **HellaSwag** (Zellers et al., 2019): Sentence completion requiring commonsense reasoning about events. Train: 10,000 / Test: 1,000.

- **OpenBookQA** (Mihaylov et al., 2018): Science questions requiring multi-hop reasoning with open book facts. Train: 4,957 / Test: 500.

- **PIQA** (Bisk et al., 2020): Physical intuition questions about everyday situations. Train: 10,000 / Test: 1,000.

- **Social IQA** (Sap et al., 2019): Questions about social situations and emotional intelligence. Train: 10,000 / Test: 1,000.

- **WinoGrande** (Sakaguchi et al., 2021): Large-scale Winograd schema challenge for coreference resolution. Train: 20,000 / Test: 1,000.

Total: Train 70,302 / Test 6,500. This subset evaluates reasoning capabilities that extend beyond pattern matching to require world knowledge.

### L.3. Video — MVBench

We use MVBench (Agarwal et al., 2025), a comprehensive video understanding benchmark with 19 distinct temporal reasoning tasks. Each task contains 900 training samples and 300 test samples.

**Action Understanding:**

- **Action Sequence**: Identify the correct order of actions in a video.

- **Action Prediction**: Predict the next action given video context.

- **Action Antonym**: Identify actions opposite to those shown. (Train: 893)

- **Fine-grained Action**: Distinguish between similar actions.

- **Unexpected Action**: Detect anomalous or unexpected actions.

- **Action Localization**: Temporally locate when an action occurs.

- **Action Count**: Count occurrences of specific actions.

**Object Understanding:**

- **Object Existence**: Verify whether objects appear in the video.

- **Object Interaction**: Identify interactions between objects.

- **Object Shuffle**: Track objects through occlusions and movements.

**Motion and State:**

- **Moving Direction**: Determine direction of object/person movement.

- **Moving Count**: Count moving entities.

- **Moving Attribute**: Identify attributes of moving objects.

- **State Change**: Detect changes in object states over time.

**Scene and Reasoning:**

- **Scene Transition**: Identify transitions between scenes.

- **Character Order**: Track the order of character appearances.

- **Egocentric Navigation**: Understand navigation from first-person view.

- **Episodic Reasoning**: Answer questions requiring episodic memory.

- **Counterfactual Inference**: Reason about hypothetical scenarios.

Total: Train 17,093 / Test 5,700 (19 test sets). This subset specifically tests temporal reasoning and video understanding capabilities.

## L.4. Image (VQA) — Visual Question Answering

We use eight visual question answering tasks:

- **ChartQA** (Masry et al., 2022): Questions about charts and graphs. Train: 5,000 / Test: 1,000.

- **OK-VQA** (Marino et al., 2019): Open-domain VQA requiring external knowledge. Train: 4,204 / Test: 841.

- **ScienceQA** (Lu et al., 2022): Multimodal science questions with explanations. Train: 5,700 / Test: 518.

- **SEED-Bench** (Li et al., 2023): Comprehensive multimodal understanding benchmark. Train: 5,500 / Test: 500.

- **TextVQA** (Fang et al., 2023): Questions requiring reading text in images. Train: 5,000 / Test: 1,000.

- **Text Recognition**: OCR-style text reading from images. Train: 5,000 / Test: 1,000.

- **VizWiz-VQA** (Gurari et al., 2018): VQA from blind users' photographs. Train: 2,086 / Test: 417.

- **VQA-RAD** (Lau et al., 2018): Medical imaging visual question answering. Train: 1,060 / Test: 200.

Total: Train 33,550 / Test 5,476. This subset evaluates visual understanding and reasoning.

## L.5. Image (VTAB) — Visual Task Adaptation

We use six image classification tasks from VTAB (Zhai et al., 2019):

- **Caltech-101**: Object recognition across 101 categories. Train: 1,500 / Test: 500.

- **EuroSAT**: Satellite image land use classification. Train: 1,500 / Test: 500.

- **Flowers-102**: Fine-grained flower species classification. Train: 1,500 / Test: 500.

- **Oxford Pets**: Cat and dog breed classification. Train: 1,500 / Test: 500.

- **SVHN**: Street view house number recognition. Train: 1,500 / Test: 500.

- **Camelyon**: Medical histopathology tumor detection. Train: 1,500 / Test: 500.

Total: Train 9,000 / Test 3,000. This subset evaluates visual classification across general and specialized domains.

## L.6. Benchmark Design Rationale

MMT-47 is designed to test several key aspects of LiME:

1. **Multimodal Generalization**: By training on text, video, and image data jointly, we test whether experts specialize by modality without explicit supervision.

2. **Multi-task Transfer**: Within each modality, we include diverse tasks (e.g., sentiment vs. inference for text, VQA vs. classification for images) to test task-level routing.

3. **No Task Identifiers**: Unlike prior multi-task learning setups, we do not provide explicit task or modality labels during training or inference. The model must learn to route based on input characteristics alone.

4. **Scale and Diversity**: With 47 test sets across 5 categories, MMT-47 ensures comprehensive evaluation that goes beyond single-domain benchmarks.

All datasets are formatted as text generation tasks with consistent prompt templates. For classification tasks, we format the output as the class label in text form. For regression tasks (STS-B), we discretize scores into bins.

*Table 21.* MMT-47 benchmark statistics. Training and test samples across five categories: Text (NLU) , Text (Reasoning) , Video , Image (VQA) , and Image (VTAB) .

| Dataset | Train | Test | Dataset | Train | Test | Dataset | Train | Test |
|---|---|---|---|---|---|---|---|---|
| **Text (NLU) — GLUE** | | | | | | *Train: 28,668 — Test: 49,716* | | |
| SST-2 | 5,000 | 872 | QQP | 5,000 | 40,430 | MRPC | 3,668 | 408 |
| QNLI | 5,000 | 5,463 | CoLA | 5,000 | 1,043 | STS-B | 5,000 | 1,500 |
| **Text (Reasoning) — Commonsense** | | | | | | *Train: 70,302 — Test: 6,500* | | |
| ARC-Challenge | 1,791 | 500 | HellaSwag | 10,000 | 1,000 | Social IQA | 10,000 | 1,000 |
| ARC-Easy | 4,127 | 500 | OpenBookQA | 4,957 | 500 | WinoGrande | 20,000 | 1,000 |
| BoolQ | 9,427 | 1,000 | PIQA | 10,000 | 1,000 | | | |
| **Video — MVBench** | | | | | | *Train: 17,093 — Test: 5,700* | | |
| Action Sequence | 900 | 300 | Object Shuffle | 900 | 300 | Moving Attribute | 900 | 300 |
| Action Prediction | 900 | 300 | Moving Direction | 900 | 300 | State Change | 900 | 300 |
| Action Antonym | 893 | 300 | Action Localization | 900 | 300 | Character Order | 900 | 300 |
| Fine-grained Action | 900 | 300 | Scene Transition | 900 | 300 | Ego. Navigation | 900 | 300 |
| Unexpected Action | 900 | 300 | Action Count | 900 | 300 | Episodic Reasoning | 900 | 300 |
| Object Existence | 900 | 300 | Moving Count | 900 | 300 | Counterfactual | 900 | 300 |
| Object Interaction | 900 | 300 | | | | | | |
| **Image (VQA) — Visual Question Answering** | | | | | | *Train: 33,550 — Test: 5,476* | | |
| ChartQA | 5,000 | 1,000 | TextVQA | 5,000 | 1,000 | VQA-RAD | 1,060 | 200 |
| OK-VQA | 4,204 | 841 | VizWiz-VQA | 2,086 | 417 | SEED-Bench | 5,500 | 500 |
| ScienceQA | 5,700 | 518 | Text Recognition | 5,000 | 1,000 | | | |
| **Image (VTAB) — Visual Task Adaptation** | | | | | | *Train: 9,000 — Test: 3,000* | | |
| Caltech-101 | 1,500 | 500 | Flowers-102 | 1,500 | 500 | SVHN | 1,500 | 500 |
| EuroSAT | 1,500 | 500 | Pets | 1,500 | 500 | Camelyon | 1,500 | 500 |
| **Total — Train: 158,613 — Test Sets: 47 — Test Samples: 70,392** | | | | | | | | |

# The Use of Large Language Models

We used a large language model (LLM) solely for minor language-related assistance, including grammar correction, language polishing, and readability improvement. The LLM was not used for content generation, scientific ideation, experimental design, data analysis, or interpretation. All research contributions, technical content, and results presented in this paper are entirely the work of the authors.

