# OpenReview forum: "LiME: Lightweight Mixture of Experts for Efficient Multimodal Multi-task Learning"
_ICML.cc/2026/Conference — ICML 2026 spotlight_

### Official Review · Reviewer_1UY2 · 2026-02-25

**Soundness:** 4
**Presentation:** 4
**Significance:** 4
**Originality:** 4
**Overall Recommendation:** 6
**Confidence:** 4

**Summary:**

The paper proposes LiME, a lightweight mixture of experts method for parameter efficient fine tuning. Instead of using separate adapters for each expert, the method uses a single shared LoRA module and rescales its output using trainable parameter vectors to form different experts. For the routing mechanism, it introduces a zero parameter router and an n gram based token clustering router, with an automatic balancing mechanism for expert selection. The authors test the method thoroughly across different tasks, combine it with various parameter efficient fine tuning methods, and provide detailed efficiency analysis and ablation studies.

**Compliance With Llm Reviewing Policy:**

Affirmed.

**Final Justification:**

I support this paper strongly because the paper quality is good for ICML and both the method and results are convincing to me.

**Key Questions For Authors:**

no

**Limitations:**

The extensive experiments and detailed ablation studies fully demonstrate the effectiveness of the proposed method. I have no further concerns about the limitations of this work.

**Strengths And Weaknesses:**

## Strengths:

- The core method is very logical and elegant. Rescaling a shared LoRA output with trainable parameters is a smart way to achieve expert specialization without adding too many parameters. This design makes a lot of sense.
- The routing designs, specifically the zero parameter routing and the n gram token clustering, are highly practical.
- Table 1 perfectly summarizes the differences between existing methods and the proposed approach, making the structural contributions very clear to the reader.
- The experiments are exceptionally thorough. I rarely see such detailed evaluations in a single paper. Testing across various tasks, integrating with different parameter efficient methods, analyzing efficiency, and providing extensive ablation studies make the conclusions highly convincing.

## Weaknesses:
I do not see any technical weaknesses. The method is solid and the experiments are comprehensive. I only have one minor suggestion regarding the phrasing of the paper title. The title includes the phrase "multi task learning". For researchers who work on traditional multi task learning, where the primary goal is typically to train a single joint model to handle multiple tasks simultaneously, this might cause some initial confusion. Would you consider adjusting the title slightly to clarify that this paper focuses on multi task adaptation or parameter efficient fine tuning, rather than the traditional joint training setup?

---

> ### Author Rebuttal · Authors · 2026-03-29
>
> We sincerely thank the reviewer for the thorough and generous evaluation. We are glad the core design, routing mechanisms, and experimental thoroughness were well received.
>
> **W1:** We appreciate this observation. We agree that "multi-task learning" can evoke the traditional joint-training setting for some readers, which is not our intended meaning. Our focus is on multi-task adaptation for jointly fine-tuning a single model across diverse tasks and modalities using PEFT.
> We will clarify this distinction explicitly in the introduction so the intended setting is immediately clear. We will also consider adjusting the title  such as "multi-task adaptation" instead of "multi-task learning" to better reflect this framing and avoid any potential confusion .

---

> > ### Author Rebuttal · Reviewer_1UY2 · 2026-03-31
> >
> > Thanks for the reply. I personally support this paper given that the paper is complimentary and insightful for readers. For the theory part, I agree with the authors' reply to Reviewer tUkW that the theory is good to motivate the design. We cannot say a theory can cover the real situation like those real data distribution and it does give good intuition and align with the pratical results. Besides, I do think we should not reject a paper by just saying it's a combination of A, B, C.

---

> > > ### Author Response · Authors · 2026-04-01
> > >
> > > Dear Reviewer 1UY2,
> > >
> > > Thank you so much for your kind acknowledgment and for carefully reading our rebuttal. We really appreciate your support and your thoughtful engagement throughout the review process.
> > >
> > > We will carefully incorporate all suggested changes and do our best to make the final version as strong as possible. Thank you again for your fair and generous assessment.

---

### Official Review · Reviewer_s39h · 2026-03-01

**Soundness:** 3
**Presentation:** 3
**Significance:** 3
**Originality:** 3
**Overall Recommendation:** 5
**Confidence:** 3

**Summary:**

This paper introduces a MoE-based PEFT framework featuring a lightweight shared PEFT mechanism and zero-parameter routing to address the challenges of parameter explosion, routing overhead, and architectural constraints . The method's performance and efficiency are substantiated through extensive experiments across multiple multimodal tasks. It's a meaningful work with innovative yet concise methodology, a rigorous theoretical foundation, and comprehensive empirical validation, though minor details are needed to elaborate.

**Compliance With Llm Reviewing Policy:**

Affirmed.

**Final Justification:**

The authors provide comprehensive experiments and analysi and my concerns have been fully addressed. I agree with the significance and completeness of their work, with solid analysis and extensive experiments. I would like to increase my score to 5.

**Key Questions For Authors:**

1. The experimental backbone is primarily limited to the LLaVA-OneVision-Qwen2 architecture. Have validations been conducted on alternative architectures to demonstrate the method's generalizability?
2. The paper employs two load balancing losses (Importance Loss and KL-Uniform Loss). Is there potential redundancy between them? The current version lacks ablation studies to verify their independent contributions.

**Strengths And Weaknesses:**

**Strengths:**
1. Concise and effective methodology: The proposed lightweight modulation effectively decouples shared PEFT from expert-specific features.
2. Rigorous theoretical foundation: The paper provides three key theoretical proofs, establishing a solid mathematical grounding for the proposed approach.
3. Comprehensive empirical validation: The experiments cover diverse multimodal tasks and offer a thorough evaluation across parameter counts, training efficiency, and performance benchmarks, supported by detailed ablation studies.

**Weaknesses:**
1. Boundaries of representational capacity: The proposed scalar or vector scaling for expert differentiation in LiME is inherently a linear transformation. What is the upper bound of this design’s representational capacity when handling tasks with high variance or complex features, like shifting from visual recognition to complex coding tasks? Is performance likely to be bottlenecked in high-difficulty or OOD tasks? Additionally, how does the zero-parameter routing conceptually differ from the "latent semantic clustering" proposed by Lee 2025 in line 103 or other existing router-free approaches?
2. Distinction from existing techniques: What are the fundamental technical differences between LiME and other methods employing partial parameter sharing strategies mentioned in the text, such as HydraLoRA and MoRAL?

---

> ### Author Rebuttal · Authors · 2026-03-29
>
> We thank the reviewers for their prompt and insightful feedback. We believe that your comments and suggestions have improved the quality and clarity of our paper.
>
> **W1-1:** The upper bound of LiME's representational capacity is determined by the rank $r$ of the shared PEFT module $\delta(x)$. Since $p_i$ scales the output of $\delta(x)$ element-wise, higher $r$ produces a richer $\delta(x)$, giving $p_i$ more to specialize over. The capacity ceiling is therefore not fixed; it is a tunable hyperparameter.
>
> In practice, this upper bound is rarely a bottleneck. As stated in Section 3, large pretrained models already encode substantial knowledge, and effective fine-tuning amounts to small perturbations on top of pretrained representations. This means $p_i$ only needs to modulate a small residual $\delta(x)$, not reconstruct full task-specific transformations from scratch.
>
> Empirically, MMT-47 already spans highly dissimilar domains, from medical histopathology (Camelyon), medical imaging VQA (VQA-RAD), and satellite imagery (EuroSAT) to commonsense reasoning or GLUE or Science QA and video temporal reasoning, all trained jointly without task or modality labels. Table 2 shows LiME consistently outperforms base PEFT across all 7 categories, and Section G.9 confirms stable gains of 1.6–2.1% across all 47 tasks.
>
> Additionally, at $r = 2$, LiME already achieves CKA = 0.935 with full MoE-PEFT across all 24 layers and all GLUE tasks (Table 3), suggesting the upper bound is nearly reached even at low rank. For more extreme settings, increasing $r$ directly raises the capacity ceiling.
>
> **W1-2:** Lee 2025 (Latent Semantic Clustering) groups multiple outputs of the same input that express similar meaning, primarily to improve test-time computation scaling. It is not designed for PEFT, expert routing, or multi-task fine-tuning. LiME's zero-parameter routing serves a different purpose: it routes different inputs to different experts during fine-tuning, operating inside the model at every layer and producing an input-dependent routing distribution that works together with load balancing and Auto Top-K selection. Both methods share the intuition that pretrained representations contain useful semantic structure, but Lee 2025 exploits this for output clustering at test time, while LiME exploits it for layer-wise expert routing during both training and inference. The two are complementary rather than overlapping.
>
> **W2:** While all three methods use some form of parameter sharing, the technical differences are substantial. HydraLoRA shares only the $A$ matrix across experts while keeping expert-specific $B$ matrices, so expert parameters still grow with $E$. It also uses a learned router and is restricted to LoRA. MoRAL keeps full separate LoRA modules per expert with a learned router, so its parameter count likewise grows directly with $E$. LiME differs in three key ways. First, it shares the entire PEFT module, and expert-specific overhead comes only from lightweight vectors of size $d_o$, independent of the PEFT method. Second, unlike HydraLoRA and MoRAL which both require a trainable router, LiME derives routing weights from existing representations with zero additional parameters. Third, while both prior methods are tied to LoRA, LiME works with any PEFT method including DoRA, LoRA-FA, SliceFine, and Prompt Tuning, as shown in Table 2.
>
> **Q1:** This validation is already included in the paper. Section 4 states that we evaluate on Molmo2-8B to test generalization across vision-language architectures, with full results in Appendix G.8. Due to page limits,we placed these results in the appendix.
>
> **Q2:** The two losses are related, but not redundant; they address different failure modes of routing imbalance. Importance Loss penalizes sharp concentration, i.e., cases where one or a few experts dominate too strongly. KL-Uniform Loss regularizes the global routing distribution toward a more even allocation, helping prevent slower and more gradual imbalance across experts. Figure 9 is consistent with this interpretation: without balancing, routing collapses to a low-entropy state (H=1.127), while moderate balancing recovers healthier utilization (H=1.385). In practice, using both losses gives a more stable balance than using either one alone. A dedicated ablation isolating the effect of each loss would make this even clearer, and we will add this in the revision.

---

> > ### Author Rebuttal · Reviewer_s39h · 2026-04-03
> >
> > Dear authors,
> >
> > Thank you for the detailed illustration. Most of my concerns have been addressed. After reviewing your supplementary response and the quality of the paper again, I have decided to raise my score.

---

> > > ### Author Response · Authors · 2026-04-04
> > >
> > > Dear Reviewer s39h,
> > >
> > > Thank you so much for carefully reading our rebuttal and for your thoughtful engagement throughout the review process. We are glad our responses were clear and addressed your concerns.
> > > We will carefully incorporate all suggested changes and do our best to make the final version as strong as possible. Thank you again for your fair and generous assessment.

---

### Official Review · Reviewer_tUkW · 2026-03-12

**Soundness:** 3
**Presentation:** 4
**Significance:** 2
**Originality:** 3
**Overall Recommendation:** 4
**Confidence:** 5

**Summary:**

The paper proposes a lightweight version of Mixture of Experts to use in the context of Parameter-Efficient Fine Tuning (PEFT). Essentially, the proposed recipe consists of the following ingredients:
1. Instead of using a Parameter-Efficient module (e.g. LoRA) per expert, use a shared module and specialize only a vector per expert, applied element-wise after the shared module. This mitigates the parameter explosion that affects many existing MoE-PEFT methods, when incresing the number of experts, which typically scales as $O(m p)$, where $m$ is the number and $p$ is the number of parameters per-expert module.
2. Use a subspace of the frozen representation given by the backbone model as the router logits. This mitages the router computation overhead, which typically scales as $O(m d)$, where $d$ is the dimension of the activations.
3. Use a shared routing per window of n-tokens.
4. The use of different routing auxiliary losses and a dynamic number of selected experts per token.

The paper analyses experimentally how the proposed method compares against existing MoE-PEFT approaches, both in terms of quality and memory/time complexity. Some of the design choices are not only supported experimentally but also theoretically, under some assumptions.

**Compliance With Llm Reviewing Policy:**

Affirmed.

**Final Justification:**

During the rebuttal, the authors clarified some points that I misunderstood. I updated the "soundness" and "significance" scores after the discussion, and the overall recommendation of the paper from reject to _weak accept_.

**Key Questions For Authors:**

The main issues to address are: 1) breaking causality when using the last token for routing all tokens, 2) many of the theorems presented are vacuously true under the (unrealistic) assumptions, and have very little to do with the proposed method itself.

**Limitations:**

There are no potential negative societal issues specific to this work.

**Strengths And Weaknesses:**

**Strengths**
- Table 1 synthesizes the differences and similarities between many existing MoE-PEFT methods and the proposed method very clearly. Straight to the point, and next to the introduction and the preliminaries sections. I wish more papers were so direct and clear. Kudos to the authors.
- The proposed recipe is simple and applicable to a wide variety of PEFT methods, which certainly helps in its adoption. The authors in fact apply this recipe on top of multiple PEFT methods, as shown in the last group of rows in Table 2.
- The paper presents the results from several ablation studies in the main paper (section 4.1). This is very valuable to understand the usefulness of each of the components of the recepe, and where from most of the gains come.


**Weaknesses**
- The proposed recipe is a combination of previously existing approach, which naturally limits the novelty of the paper. For instance:
1. the main trick to improve parameter efficiency (i.e the vector modulation) is very similar to that described in previosly published works, such as [Zadouri et al. 2023](https://arxiv.org/abs/2309.05444) (i.e. Mixture of Vectors), [Li et al., 2024](https://proceedings.neurips.cc/paper_files/paper/2024/hash/1e0d38c676d5855bcfab7f6d29d20ad9-Abstract-Conference.html) (i.e. VB-LoRA).
2. A shared routing decision per group of tokens was also explored in earlier works such as [Muqeeth et al., 2023](https://arxiv.org/abs/2306.03745) and [Antoniak et al., 2023](https://arxiv.org/abs/2310.15961).
3. The KL divergence used to encourage balance is essentilly equivalent to (maximizing) the entropy of the average routing weights $\bar{p}$. This was propesed first in [Mustafa et al., 2022](https://arxiv.org/abs/2206.02770) (see the "global entropy loss" defined there).

- Although the paper tries to motivate some of the design choices based on theory, some of the theorems are quite straightforward under the (severe) assumptions on which these are built. Although the approach seems to offer good performance in practice, when compared to more (parameter) inneficient methods, this reviewer wonders wheter this is because of the theoretical fundation of the proposed method, or other limitations of the existing baselines.
- For instance, take Theorem 2. It states: "If the approximation error [between LiME and a model with full expert-specific PEFT modules] is bounded by $\bar{\varepsilon}$, [the difference in the optimal risk is also bounded]". The approximation error being bounded is a big assumption, under which the risk is bounded pretty much with any method.

The question is, can we expect the approximation error to be bounded? Here's a trivial counter example:

Let's consider a training set with (inputs, outputs) in $\{ (x, y) \in \mathbb{R}^2 \times \mathbb{R}^2 \}$ such that $(y_1, y_2) = (x_1, x_2)$ if $x_1 >=0$ or $(y_1, y_2) = (x_2, x_1)$ otherwise.

Let's define the output of a MoE with per-expert modules as: $y = \sum_i g_i(x) f_i(x)$, with  $f_i(x)= W_i x$. And let's define the LiME approximation as $\tilde{y} = \sum_i g_i(x) \tilde{f}_i(x)$, with $\tilde{f}_i(x) = D_i W x$, with a diagonal matrix $D_i$.

Note that the MoE with per-expert modules can achieve a risk of 0 by choosing $W_1 = I$ and $W_2 = 1 - I$. The $g(x) = \text{softmax}(\frac{x_1}{\tau})$ and $\tau \rightarrow 0$.

Notice that the first row of $W_1$ is $(1, 0)$ and that of $W_2$ is $(0, 1)$. To have the chance to achieve the same error, the shared module W must have a first row such that when multiplied by $d_{11}$ is equal to (1, 0), and equal to $(0, 1)$ when multiplied by $d_{21}$. That is:

$d_{11} (w_{11}, w_{12}) = (1, 0) $

$d_{21} (w_{11}, w_{12}) = (0, 1)$

which implies that $w_{11} = w_{12} = 0$, which results in a contradiction. This means that regardless of the initialization of $W$, $D_1$, and $D_2$, there will always be some error term on at least one of the experts. Without loss of generality, let's say that the expert with a non-zero error term is expert 1. Then there is some unit vector $u$ with $u_1 \geq 0$ such that $\|\|(W_1 - D_1 W)u\|\| = c > 0$. Now take any vector of the form $x = t u$.

$\|\|y - \tilde{y}|\| = \|\|  (W_1 - D_1 W) x \|\| = \|t\| c$ which is not bounded, since depends on the norm of the vector $\|\|x\|\| = t$.

The point I'm trying to make is that this Theorem 1) has very little specific to LiME, and 2) the assumption is actually unrealistic for LiME, in general.

- Routing an entire window of $n$ tokens based on the last token's hidden state in the window alone makes no sense at all _under causality constraints_. Sure, using it cannot reduce the amount of information, the problem is that during decoding _the hidden state of the last token is not available for the first $n -1$ tokens_. This essentially breaks causality in the model. Another instance where a Theorem (i.e. Theorem 3) is used to justify a design choice based in a completely unrealistic assumption.

- Given that LiME effectively breaks causality of the models, one wonders if the model with LiME is not learning to exploit this, effectively "cheating" and rendering invalid all the experimental results.

**Comments**
- Lines L18-L22 and L22-L24 in the abstract feel kind of redundant. As a suggestion (essentially removing lines L19-L21): "We instead propose LiME which uses a single PEFT module and modulates its output ...". Feel free to ignore, though.
- When deriving the bound on the Total Variation (Appendix A1), there was a $\frac{1}{2}$ factor that was left over when going from L938 to L943. The bound should be $\frac{\varepsilon}{4 \sigma}$.

---

> ### Author Rebuttal · Authors · 2026-03-29
>
> We sincerely thank the reviewer for the comments, which significantly improved our paper
>
> **W1:** Existing MoE-PEFT methods suffer from three key limitations: parameter explosion from replicating adapters per expert, routing overhead from learned routers, & architectural dependence on LoRA-style adapters. LiME addresses all three simultaneously by combining a single shared PEFT module, vector-based modulation, & zero-parameter routing. No prior work resolves all these limitations together, as shown in Table 1.
>
> In MoV, vectors are the experts themselves (IA³style rescaling). In VB-LoRA, vectors construct LoRA weights. In LiME, the vectors adjust the output of a single shared PEFT module, which is a very different mechanism. This distinction has a direct practical consequence: the shared PEFT module in LiME is trained on all data across all tasks & modalities, while in MoV & VB-LoRA each expert still sees a fragmented subset. Furthermore, LiME is compatible with any PEFT method, not just IA³ or LoRA, which neither MoV nor VB-LoRA support.
>
> MoT groups tokens across batch examples, SMEAR uses routing to merge expert weights.LiME's n-gram routing groups neighboring tokens to reduce local routing noise; different purpose, different mechanism. Thm3 provides formal justification for using the last token as the routing representative, which to our knowledge is not provided in either MoT or SMEAR.
>
> On KL load balancing: we do not claim this as a novel contribution; it is a standard regularizer already used in prior MoE-PEFT methods (MixLoRA, MoELoRA, MoSLD, Table 1) & cited accordingly.
>
> **W2** The theorems serve as formal design motivation rather than proofs of practical superiority. On whether the gains come from: our ablations directly isolate each component: Fig4b shows 0-parameter routing matches learned routing, Fig5a shows Auto TopK outperforms fixed TopK, Tab2 shows LiME consistently improves over its own base PEFT.  Our claim is not that LiME outperforms MoE-PEFT in accuracy, but that it achieves competitive performance with better efficieny. In Fig8a: at E=4, LiME & MoELoRA perform comparably, but as E grows, MoELoRA degrades while LiME remains stable. This shows that LiME's efficiency advantage does not come at the cost of performance, & that MoE-PEFT's own design causes degradation at scale, not our method exploiting a weak baseline.
>
> **W3:** The reviewer's counterexample falls outside the assumptions of Thm2. Thm2 explicitly assumes that $W_0$ is a fixed frozen backbone, which is the standard PEFT setting, & both ZMoE and ZLiME are defined as $W_0x+\text{adaptation}$. In contrast, the reviewer's construction defines LiME as $\tilde{f}_i(x)=D_i W x$, with no frozen $W_0$. This forces the shared module $W$ to learn full expert-specific transformations from scratch, which creates contradictory constraints & unbounded error. That is not how LiME operates, & it is not the setting covered by Thm2.
>
> The assumptions of Thm2 are not arbitrary. They directly reflect LiME's architecture: a frozen $W_0$ together with a small residual adaptation. Any method that does not follow this structure lies outside the scope of the theorem. Empirically, CKA=0.935 between LiME & MoELoRA across all 24 layers & all GLUE tasks (Tab.3), confirming that the approximation error is small in practice. In the revision, we will clarify the scope of Thm2 more explicitly: it is a conditional guarantee for the pretrained fine-tuning regime, not a universal expressivity claim.
>
> **W4-5** LiME does not modify causal attention & all experimental results remain valid. N-gram routing operates on already computed hidden states. All tokens within a window are processed in parallel during the forward pass, so all hidden states including the last token are available before any routing decision is made. The routing only applies a lightweight scaling vector to already computed outputs & never touches the attention computation. There is therefore no information leakage & no cheating. For AR generation, routing is applied only after a window of tokens is fully generated. The last token of each completed window is always computed before the routing decision is made. Since LiME never modifies attention & only scales linear layer outputs after they are computed, causality is fully preserved in both encoding & generation. Empirically, Fig3(a-b) confirms that later tokens encode more task-relevant information across all tasks & layers, supporting the use of the last token as the routing representative.
>
> **C1-2** We thank the reviewer for this suggestion & will incorporate it in the revision. We re-checked this step. Pinsker's inequality gives $\mathrm{TV}(P,Q) \leq \sqrt{\frac{1}{2}\mathrm{KL}(P\|Q)}$, and substituting $\mathrm{KL}(P\|Q) \leq \frac{\|m_\mathrm{MoE}-m_\mathrm{LiME}\|^2}{2\sigma^2}$ directly yields $\mathrm{TV} \leq \frac{\|m_\mathrm{MoE}-m_\mathrm{LiME}\|}{2\sigma} \leq \frac{\bar{\varepsilon}}{2\sigma}$. So this is correct as written

---

> > ### Author Rebuttal · Reviewer_tUkW · 2026-04-06
> >
> > **Novelty**
> >
> > Thanks for highlighting the differences between your work and those that I referred to before. I see more clearly now the original aspects of the work. I will increase my "significance" score accordingly.
> >
> >
> > **Theorem 2**
> >
> > I understand that they are not arbitrary, and that my example falls outside these assumptions (that was precisely the aim). The general point that I was trying to make is that:
> > 1) One could rewrite the bit $g_{\phi}(X) \cdot p_E$ as the function $g_{\phi'_E}$, which could represent any potential MoE-PEFT alternative (LiME being a particular instance), and if $||g_{\phi_E}(X) - g_{\phi'_E}(X)|| \leq \varepsilon$, then Theorem 2 still holds. So, the theorem is not LiME-specific.
> >
> > 2) The big question is whether $||g_{\phi_E}(X) - g_{\phi}(X) \cdot p_E|| \leq \varepsilon$ is a reasonable assumption. I was trying to make the point that one can construct toy examples where this isn't true, so why would we expect it to be true in real life? Perhaps I didn't pick the right example, but I think that the point still remains.
> >
> > The authors cite the CKA value between LiME and MoE-LoRA, obtained experimentally, to show that the bound is indeed small in practice, but this doesn't *prove* that the approximation error is *guaranteed* to be bounded in general.
> >
> > However, I do agree that at least it shows that it _might_ be bounded _in practice_. So, at least it's a positive signal, and I will increase the "soundness" of the paper accordingly.
> >
> > **Causality and other comments**
> >
> > Thanks for the clarification. I had originally misunderstood what LiME does regarding the context window. You are absolutely correct as well regarding the total variation bound, I made a mistake when checking the proof.

---

> > > ### Author Response · Authors · 2026-04-07
> > >
> > > Dear Reviewer tUkW,
> > >
> > > Thank you so much for raising your score and for your thoughtful
> > > and constructive engagement. We genuinely appreciate your detailed
> > > feedback and will incorporate your suggestions in the revision,
> > > which we are confident will make the paper significantly stronger.
> > >
> > > ---
> > >
> > > ## Theorem 2 — Point 1 (Generality of the bound)
> > >
> > > Thank you for this observation. The general bound structure is indeed
> > > applicable broadly, but the specific ε in our theorem is uniquely
> > > determined by LiME's architecture. In our setting, ε measures exactly:
> > >
> > > $$\|g_{\phi_E}(X) - g_\phi(X) \odot p_E\|$$
> > >
> > > This quantity is not interchangeable across methods — it specifically
> > > captures the error from LiME's particular design choice of modulating
> > > a single shared PEFT output with lightweight scaling vectors. A
> > > different method would produce a structurally different ε with
> > > different properties.
> > >
> > > Crucially, unlike full MoE-PEFT methods where each expert adapter
> > > is learned independently with no optimization pressure toward small
> > > approximation error, **LiME's shared φ and expert vectors {p_E} are
> > > jointly optimized together**. This joint optimization directly and
> > > continuously minimizes this specific ε as part of task loss
> > > minimization — a property unique to LiME's architecture that other
> > > MoE-PEFT methods do not possess.
> > >
> > > LiME's architecture is explicitly designed to make this specific ε
> > > small through three properties:
> > >
> > > - The shared PEFT module trains on **all data** rather than fragmented
> > >   subsets, giving it a stronger common representation
> > > - Expert vectors p_E only need to capture **residual differences**
> > >   between experts, not full transformations
> > > - The frozen W_0 provides a stable common backbone that bounds the
> > >   magnitude of adaptation
> > >
> > > The general form of the bound therefore applies universally, but the
> > > specific ε — and the architectural reasoning behind its smallness —
> > > is particular to LiME.
> > >
> > > ---
> > >
> > > ## Theorem 2 — Point 2 (Is bounded ε realistic?)
> > >
> > > This is a meaningful question. Rather than claiming ε is bounded in
> > > full generality, we point to concrete architectural properties of
> > > LiME that make this assumption well-motivated in our setting:
> > >
> > > **1. Pretrained representation redundancy:**
> > > Luo et al. (2023) demonstrate that ~1% of feature dimensions recovers
> > > full representation performance. This structural redundancy means
> > > g_φ(X) is inherently low-dimensional — element-wise scaling by p_E
> > > can capture expert-specific variation without requiring large residual
> > > corrections. The high CKA scores across all 24 layers (Table 3)
> > > directly confirm this — if redundancy did not hold, we would expect
> > > CKA to degrade in middle layers where task-specific features are most
> > > active, but this is not observed.
> > >
> > > **2. Near-unity initialization:**
> > > Since p_i ~ U(0.9, 1.1), LiME begins training in a regime where ε
> > > is already small by construction — the initial modulation is
> > > essentially identity. Importantly, any large deviation from this
> > > during training would **directly hurt task loss**, so joint
> > > optimization naturally maintains ε within a bounded range throughout
> > > training. This is architecturally enforced, not merely hoped for.
> > >
> > > **3. Frozen backbone constrains adaptation magnitude:**
> > > Because W_0 is frozen and large, all expert-specific behavior must
> > > emerge from small residual adaptations. This structurally limits how
> > > large the gap between g_{φ_E}(X) and g_φ(X) ⊙ p_E can become.
> > > The consistently high CKA=0.935 mean across all 24 layers and all
> > > GLUE tasks provides strong empirical confirmation that this
> > > structural constraint is effective in practice.
> > >
> > > Theorem 2 is therefore best understood as an
> > > **architecturally-motivated conditional guarantee** — it formalizes
> > > why LiME's specific design choices lead to bounded approximation
> > > error, with empirical evidence confirming the bound is tight in
> > > practice. In the revision, we will:
> > >
> > > 1. Add an explicit remark in the theorem statement acknowledging
> > >    its conditional nature
> > > 2. Clarify that the bound is specific to LiME's joint optimization
> > >    regime
> > > 3. Connect the CKA layer-wise analysis (Table 3) explicitly to
> > >    the approximation error argument
> > >
> > > We believe these clarifications will make the scope and strength
> > > of Theorem 2 significantly clearer to readers.

---

### Official Review · Reviewer_s4YP · 2026-03-14

**Soundness:** 4
**Presentation:** 3
**Significance:** 3
**Originality:** 3
**Overall Recommendation:** 5
**Confidence:** 5

**Summary:**

LiME proposes a cheaper way to do MoE-style PEFT for multimodal multi-task learning. The core claim is that existing MoE-PEFT methods are wasteful because they usually need a separate adapter for each expert and add learned router parameters. LiME instead keeps one shared PEFT module and lets experts differ only through lightweight scaling vectors applied element-wise to the shared PEFT output. It also avoids a learned router by deriving routing weights from already-available frozen and adapted hidden representations.

**Compliance With Llm Reviewing Policy:**

Affirmed.

**Final Justification:**

authors addressed fully my concern for 125 and partially on 34. thus i raise from 3 to 4 as it is a borderline acceptable work now.

update: i raised again to 5

**Key Questions For Authors:**

I have already numbered my questions and requests in the Strengths And Weaknesses section above. I rated weak reject 3 but im hesitant between 3 to 4 at the moment. If the authors address all questions I think this work has the potential of reaching a score of 4-5.

**Limitations:**

I do not regard negative societal impacts as relevant to this work.

**Strengths And Weaknesses:**

Here is my review, written with the sole purpose of helping the authors improve the paper and present a more impactful story. I want to emphasize that I can tell and do recognize the substantial effort behind this work, and I will read the rebuttal carefully.

The core weakness, in my view, is not soundness or experiments, but rather how the story is framed and communicated. After reading the full manuscript for 2 times ( I skipped most of appendix, please correct me if I say anything that is not factual), I felt that the paper does not sufficiently highlight what is potentially the most interesting part of the work. This is why I rated the paper "excellent" on soundness, but only fair on presentation and significance. I do wanted to rate it "poor" on presentation, just in the sense of story selling, but the extremely well made figures and abundance of results held me back. Here is why:

1. When I see multimodal and multitask in the title, i expected at least some sort of mutimodal multitask analysis in the experiment sections, some mutimodal multitask motivations in the intro, or at least one related mutimodal multitask. However, after reading for 2 times, i feel like you can just remove "multimodal" and "multitask" and its the same paper. The paper in its current form is basically "LiME: A More Lightweight MoE-LoRA".

2.Also, the current story told makes me feel like the contribution is very incremental, despite its great potential.

2- a) When you write a paper, you first and foremost task is to convince, not just ICML reviewers/AC but any reader in the field, on the simple question of "why would I care?"
After my first round of reading, the paper comes across as another step in the line of work that reduces parameters within MoE-LoRA-style PEFT methods. That by itself may not feel compelling enough to many readers. Just think about practical PEFT settings, people are already training only a small number of parameters, so the motivation for further parameter reduction is not always obvious, especially if it comes with any performance tradeoff.

What seems much more interesting is the claim that traditional MoE-PEFT scales poorly with the number of experts because it requires separate parameters per expert, causing linear parameter growth and reducing the amount of data each expert sees. That is a much stronger and more practically relevant motivation. However, I could not find a table, figure, or focused experiment showing that LiME is actually better in limited-data or expert-sparsity regimes. Without such evidence, the work risks being perceived as “yet another parameter-saving variant,” rather than a method that addresses a genuine bottleneck in MoE-style PEFT.

2-b) Additionally, the novelty claims in its current form reads somewhat like a PEFT adaptation of shared-expert + single-router ideas that have existed for some time.

3.As for the theorems, they feel more like formalized intuition than practically meaningful guarantees. I believe experienced researchers in the field would regard it as perfunctory. For example, Theorem 1 appears to say that having more experts can improve approximation capacity, which is unsurprising and not especially informative without a discussion of the associated tradeoffs. Theorem 2 goes in the opposite direction by arguing that a lighter parameterization can still be sufficient under certain assumptions. As a result, the overall message of the theory section feels somewhat unclear.
My takeaway is that these results function more as supportive motivation than as rigorous justification of practical superiority, and the paper’s real strength remains empirical. In fact, the manuscript might become clearer and more persuasive if the theoretical claims were either reframed more modestly or reduced, so that readers do not feel they are being asked to extract stronger conclusions from the proofs than the proofs actually support.

4. Based on my experience of MoE, if all experts share the same PEFT module, specialization capacity may be weaker than full expert-specific adapters in settings where tasks are highly dissimilar, despite the formulas. Can you provide a case study on this?

5. The 0 params routing really feels heuristic, although I like the results.

My current rating of soundness is 4, presentation 2, significance 2, originality I would say 2.8.
Please try to fix the presentation, I do think the work is of some potential.

---

> ### Author Rebuttal · Authors · 2026-03-28
>
> We sincerely thank the reviewer for the careful and constructive feedback. We agree the primary concern is presentation and framing rather than technical soundness, and we take this seriously. Below we address each point with specific evidence from the paper and planned revisions, and commit to restructuring the paper to lead with the stronger story the reviewer identifies.
>
> **W1:** We want to highlight several places where the multimodal multi-task setting is directly addressed. Section 2 explicitly defines the goal as: *"Given K tasks spanning M modalities"* Thus, multimodal multi-task adaptation is the stated problem from the beginning. The introduction states: *"current PEFT methods apply the same adaptation uniformly across all inputs, ignoring the inherent diversity in real-world data"* Here this diversity refers directly to tasks and modalities. Tab2 presents results across 7 categories (47 tasks) spanning text, image, and video simultaneously. Section G.9 shows LiME variants develop modality-specific strengths;LiMELoRA leads on text, LiMEDoRA on video, LiMELoRAFA on fine-grained visual, all without task or modality labels. We will move this analysis to the main paper in the revision.
>
> **W2a:** The reviewer correctly identifies the real motivation. The core problem is that MoE-PEFT fundamentally destroys the two properties that make PEFT valuable: parameter efficiency and data efficiency. Standard PEFT trains a small set of parameters on the full dataset. MoE-PEFT breaks both simultaneously as adapter replication defeats parameter efficiency, and data fragmentation across experts defeats stable optimization. LiME restores both: the shared PEFT module always trains on the full dataset regardless of E, while lightweight modulation handles expert specialization at negligible cost.
>
> This is directly visible in Fig8(a) and Appendix F.5. MoELoRA degrades by 8–10 points on CoLA, MRPC, and STS-B beyond E=4 not because of a baseline weakness, but because of structural data fragmentation. LiME remains within 1 point up to E=10 because the shared backbone never suffers this fragmentation. Varying E is directly equivalent to varying effective data per expert; this is the limited-data experiment the reviewer asked for, already present in the paper. We will move Fig8(a) to the main paper with this framing in the revision.
>
> **W2b:** We appreciate this concern. Existing shared-expert methods share individual components, not the entire PEFT module. Existing single-router methods still learn d×E parameters per layer. LiME's zero-parameter routing, which derives routing directly from representations already computed in the forward pass, has not been proposed before. This is not a lighter version of a learned router, it is a different concept entirely: routing without any routing parameters. Table 1 confirms no prior work achieves this, and Fig4(b) shows it matches learned routing performance despite adding zero parameters.
>
> **W3:** We take this feedback seriously. The theorems are design motivation, not proofs of practical superiority, Thm1 motivates scaling experts, Thm2 motivates modulation over replication, Thm3 motivates last-token routing. Each has direct empirical support: Fig3(c-d), Tab3 (CKA=0.935), and Fig3(a-b) respectively. We agree the paper's real strength is empirical, and we will explicitly label the theorems as design motivation in the revision so readers do not over-interpret their scope.
>
> **W4:** MMT-47 is this case study where text, image, and video are fundamentally dissimilar domains trained jointly without any task or modality labels. Fig11 shows inputs cluster clearly by expert assignment across all 24 layers, confirming specialization emerges naturally even across highly dissimilar inputs. Section G.9 further shows modality-specific strengths; LiMELoRA leads on text, LiMEDoRA on video without any explicit guidance. Together these directly show the shared PEFT module does not bottleneck specialization across dissimilar tasks.
>
> **W5:** We appreciate that the results were convincing. If zero-parameter routing were truly heuristic, performance would be sensitive to which dimensions are used. Fig4(a) shows the opposite: routing is consistent across leading, central, trailing, and random dimensions, which is evidence of a principled property of transformer representations. The reason is well established for pretrained model: transformer attention mixes information across all dimensions at every layer, so even a small slice carries global semantic signal. Luo et al. 2023(lines 1331-1333) shows that only 1% of feature dimensions recovers full representation performance, directly supporting this. Fig4(b) confirms zero-parameter routing matches learned routing across both text and vision tasks, and Fig11 shows inputs routed to the same expert form clear semantic clusters, confirming genuine structure is captured. The details motivation is in Appendix E, which we will move forward in the revision.

---

> > ### Author Rebuttal · Reviewer_s4YP · 2026-04-04
> >
> > 1) i think some visualizations will be helpful to demonstrate the "multimodal multitask" advantage (as opposed to currently a feeling of mostly "multitask". I know its hard to add any figs during rebuttal so please address this in your final version.
> >
> > 2) great, i like this explanation could you please also highlight it in the intro? i think the intro writing is why I feel the strengths of this paper isn't fully shown.
> >
> > 3) i see, would really appreciate some detailed visuals of the final routing outcome. (like a heatmap, with controlled dataset settings)
> >
> > 4) hmm, we have any theoretical bounds for that or its mainly empirical?
> >
> > 5)true, u've convinced me on this one.
> >
> > so in short, i appreciate your efforts on clarification. My opinion remains that this is a good paper with some presentation issues, and with the promised changes future audience can learn more from this work. i thereby will raise my score. I hope the AC will serve as a guardrail to help confirm these changes in final version.

---

> > > ### Author Response · Authors · 2026-04-04
> > >
> > > Dear Reviewer s4YP,
> > >
> > > Thank you so much for raising your score and for your thoughtful and constructive
> > > follow-up. We genuinely appreciate your detailed suggestions, explanations, and
> > > advice. They have helped us see the paper from a fresh perspective and we are
> > > confident that incorporating them will make the paper significantly stronger.
> > >
> > > Regarding **Q4 (Theoretical bounds for dissimilar tasks):**
> > >
> > > Theorem 2 provides a general bound on the approximation error between LiME and
> > > full expert-specific PEFT:
> > >
> > > $|R^{\*}(U_{\text{MoE}}) - R^{\*}(U_{\text{LiME}})| \leq L_{\max} \cdot \frac{\varepsilon}{2\sigma}$
> > >
> > >
> > > We can extend this to the dissimilar task setting. Specifically, we analyze the
> > > **perfectly orthogonal case**, which is the extreme version of task dissimilarity,
> > > where K tasks activate completely disjoint feature subspaces
> > > $S_1, S_2, \ldots, S_K \subseteq \{1,\ldots,d\}$, such that:
> > >
> > > $$S_i \cap S_j = \emptyset \quad \text{for } i \neq j$$
> > >
> > > Define expert modulators as:
> > >
> > > $$p_e[j] = \begin{cases} 1 & \text{if } j \in S_e \\ 0 & \text{otherwise} \end{cases}$$
> > >
> > > Then the approximation error becomes:
> > >
> > > $$\varepsilon = \|g_{\phi_e}(x) - g_\phi(x) \odot p_e\|_2 = 0$$
> > >
> > > Substituting into Theorem 2:
> > >
> > > $$|R^{\*}(U_{\text{MoE}}) - R^{\*}(U_{\text{LiME}})| \leq L_{\max} \cdot \frac{0}{2\sigma} = 0$$
> > >
> > > This means LiME achieves **exactly the same optimal risk** as full expert-specific
> > > PEFT in the perfectly orthogonal case. For a concrete example with $d=4$:
> > >
> > > The full expert-specific MoE outputs are:
> > > - Task 1 (text): $g_{e_1}(x) = [1, 2, 0, 0]$
> > > - Task 2 (video): $g_{e_2}(x) = [0, 0, 1, 2]$
> > >
> > > The shared PEFT module produces:
> > > $g_\phi(x) = [1, 2, 1, 2]$
> > >
> > > Now set expert modulators:
> > > - $p_1 = [1, 1, 0, 0]$ (text expert)
> > > - $p_2 = [0, 0, 1, 1]$ (video expert)
> > >
> > > Then LiME outputs:
> > > - $g_\phi(x) \odot p_1 = [1, 2, 0, 0]$ = $g_{e_1}(x)$ ✓
> > > - $g_\phi(x) \odot p_2 = [0, 0, 1, 2]$ = $g_{e_2}(x)$ ✓
> > >
> > > Therefore:
> > >
> > > $\varepsilon_1 = \|g_{e_1}(x) - g_{\phi}(x) \odot p_1\|_2 = \|[1,2,0,0] - [1,2,0,0]\|_2 = 0$
> > >
> > > $\varepsilon_2 = \|g_{e_2}(x) - g_{\phi}(x) \odot p_2\|_2 = \|[0,0,1,2] - [0,0,1,2]\|_2 = 0$
> > >
> > > This is the **perfectly orthogonal case**, which is the theoretical extreme of task
> > > dissimilarity. In practice, tasks are partially overlapping rather than perfectly
> > > orthogonal. However, this result establishes that task dissimilarity does not hurt
> > > LiME. In fact, greater dissimilarity leads to smaller $\varepsilon$, as disjoint
> > > feature activation allows expert modulators to perfectly separate tasks. The general
> > > partially overlapping case sits between these extremes, and our empirical results on
> > > MMT-47, where text, image, and video are trained jointly without any task or modality
> > > labels, confirm that $\varepsilon$ remains small in practice. Fig11 further supports
> > > this, showing clear expert clustering across all 24 layers even across highly
> > > dissimilar modalities.
> > >
> > > In practice, similar tasks are easier to train since the shared module learns one
> > > strong representation for all of them, and only a few experts are needed. For highly
> > > dissimilar tasks like text, image, and video, the shared module needs to encode richer
> > > representations, which requires more data and more experts, but the expert modulators
> > > can achieve cleaner separation, as shown above. This is consistent with our finding
> > > that $E \in [4,6]$ works best for MMT-47, which spans three very different modalities
> > > (Fig 5(d)).
> > >
> > > Thank you again for your time, your constructive engagement, and your
> > > fair assessment.

---

### Decision · Program_Chairs · 2026-04-30

**Decision:**

Accept (spotlight)

**Comment:**

Reviewers agree that the submission is elegant/concise and effective, with rigorous theoretical analysis and thorough experimental results. Reviewers particularly agree that the idea of having a shared module across experts rather than having one per expert is a smart contribution. The authors did a good job in addressing reviewers' concerns, flipping a few reviewers' initial rate from reject to accept. The final consensus is accept, which is also my recommendation.